# iN2V: Bringing Transductive Node Embeddings to Inductive Graphs

**Nicolas Lell** [1]   **Ansgar Scherp** [1]

## Abstract

Shallow node embeddings like node2vec (N2V) can be used for nodes without features or to supplement existing features with structure-based information. Embedding methods like N2V are limited in their application on new nodes, which restricts them to the transductive setting where the entire graph, including the test nodes, is available during training. We propose inductive node2vec (iN2V), which combines a post-hoc procedure to compute embeddings for nodes unseen during training and modifications to the original N2V training procedure to prepare the embeddings for this post-hoc procedure. We conduct experiments on several benchmark datasets and demonstrate that iN2V is an effective approach to bringing transductive embeddings to an inductive setting. Using iN2V embeddings improves node classification by 1 point on average, with up to 6 points of improvement depending on the dataset and the number of unseen nodes. Our iN2V is a plug-in approach to create new or enrich existing embeddings. It can also be combined with other embedding methods, making it a versatile approach for inductive node representation learning. Code to reproduce the results is available at https://github.com/Foisunt/iN2V.

## 1. Introduction

A graph neural network (GNN) may be trained without prior knowledge about the data it will encounter after deployment. This is because, in real-world graphs, new nodes and edges appear or disappear over time. For example, papers appear in citation networks, products are added to or removed from co-purchase graphs, and users join social platforms, creating new connections. These scenarios align with the *inductive setting* in graph learning, where test data is entirely unseen

during training. In contrast, the *transductive setting* allows access to the entire graph during training, while the test node labels remain hidden. From a broader perspective, the inductive setting resembles a single time step in a temporal graph learning task. In temporal GNNs, the goal is to learn representations over multiple snapshots of a graph or a graph with temporal information attached to the nodes and edges, respectively (Polleres et al., 2023; Longa et al., 2023).

Various methods exist to compute node embeddings solely from graph edges (Perozzi et al., 2014; Grover & Leskovec, 2016; Donnat et al., 2018). This is important when node features are unavailable or for enriching existing features. When using text-based embeddings such as Bag of Words, some embeddings might be missing. For example, due to out-of-vocabulary words, the Citeseer dataset (Sen et al., 2008) contains 15 nodes with empty embeddings. Other reasons for new nodes include newly created social media accounts that yet lack user-provided information.

Absent node features pose a challenge for GNNs. Message-passing GNNs such as GraphSAGE (Hamilton et al., 2017) can infer missing information from neighboring nodes. MLP-based GNNs like Graph-MLP (Hu et al., 2021) and GLNN (Zhang et al., 2022) rely solely on node features, i. e., do not have message-passing, and cannot handle such cases. A common approach to addressing missing features is Feature Propagation (Rossi et al., 2022). It propagates node features along graph edges, filling in missing features while preserving existing features.

We expand on this idea and introduce iN2V, a general and simple post-hoc approach to using trained embeddings to induce embeddings for nodes appearing in the inductive test set. We modify the training process for the popular transductive embedding model node2vec (Grover & Leskovec, 2016) to foresee future embeddings. We propose a simple but effective post-hoc procedure for propagating and updating embeddings for new nodes in the inductive setting, effectively enabling representation learning for unseen nodes. Unlike Feature Propagation (FP), our iN2V *adapts* the embeddings of training nodes during propagation, a crucial feature to enable the inductive setting. Since iN2V operates on N2V embeddings rather than raw node features provided by the datasets, it avoids relying on external node-specific information. Furthermore, it can be combined

[1]Research Group on Data Science and Big Data Analytics, Ulm University, Ulm, Germany. Correspondence to: Nicolas Lell <Nicolas.Lell@uni-ulm.de>.

*Proceedings of the $42^{nd}$ International Conference on Machine Learning*, Vancouver, Canada. PMLR 267, 2025. Copyright 2025 by the author(s).

with any existing embeddings from the datasets, enriching the node representations and improving downstream task performance. We evaluate the effectiveness of our iN2V embeddings on a range of homophilic and heterophilic datasets using the MLP and GraphSAGE models. Averaged over the other parameters, iN2V outperforms Feature Propagation by 1 point on homophilic and 0.7 points on heterophilic datasets, 1.3 points when using MLP and 0.6 points when using GraphSAGE as the classification model. When using only the extended N2V embeddings, iN2V outperforms FP by 1.3 points and by 0.6 points when using both the extended N2V embeddings and graph features. Finally, when using at most $20\%$ of the nodes for embedding generation and training, iN2V outperforms FP by 1.2 points vs 0.8 points when using at least $60\%$ of the nodes for training. In summary, our contributions are:

- Introduce iN2V, a simple and effective post-hoc method for extending trained node embeddings to unseen nodes.

- Enhance node2vec training with modifications that prepare their adaptability to inductive settings.

- Demonstrate performance gains, showing that both the inductive extension and modified training improve classification accuracy.

- Validate iN2V's robustness, showing it remains effective even when trained on only $10\%$ of nodes; in some cases outperforming using the original dataset features.

## 2. Related Work

### 2.1. Node Embeddings without Features

Many node feature generation methods are based on random walks. DeepWalk (Perozzi et al., 2014) generates random walks from a graph, treats each random walk as a sentence, and trains word2vec (Mikolov et al., 2013) embeddings on those random walks. Building on DeepWalk, node2vec (Grover & Leskovec, 2016) introduces a biased random walk generator to better balance between locality and exploration by giving distinct probabilities to return to the previous node, go to a node connected to the previous node, or visit a node not connected to the previous node. LINE (Tang et al., 2015) generates two sets of embeddings independently and concatenates them afterward. The first embedding optimizes that neighbors are similar, and the second embedding that nodes with many connections have similar embeddings. Another approach to node embeddings is subgraph2vec (Narayanan et al., 2016), which first generates rooted subgraphs for all nodes and then learns skip-gram embedding where the subgraphs of neighboring nodes are used as context for the current node. While these approaches build on the idea that neighboring nodes should have more

similar embeddings than distant nodes, struc2vec (Ribeiro et al., 2017) focuses on neighborhood degree patterns. Random walks are done based on edge weights of a fully connected graph, with edge weights calculated from the similarity of the degree distribution of the neighborhood of each node. GraphWave (Donnat et al., 2018) treats spectral graph wavelets as distributions to provide nodes with similar structural roles similar embeddings. Sub2Vec (Adhikari et al., 2018) trains subgraph-level embeddings by applying paragraph2vec (Le & Mikolov, 2014) with additional random walks to better preserve the neighborhood and structural properties of the subgraphs. There are also approaches for graph-level embeddings like graph2vec (Narayanan et al., 2017). It treats subgraphs as vocabulary and applies the doc2vec skip-gram training process. Learning graph-level representations is less related to our work, which aims to learn features for nodes unseen during training.

RDF2vec (Ristoski & Paulheim, 2016) is similar to DeepWalk but applied to Resource Description Framework (RDF) graphs. It first converts an RDF graph into sequences and then trains word2vec (Mikolov et al., 2013) on them. There are different follow-up works for RDF2vec; for example, (Hahn & Paulheim, 2024) used RDF2vec in a continual setup by sampling new walks starting from the new edges or entities. Other embedding models used for Knowledge Graphs (KGs) aim to learn not only node but also edge or relation embeddings. TransE (Bordes et al., 2013) embedded nodes and relations such that if there is a relation between two entities. The first entity's embedding added with the relation embedding is trained to be close to the second entity's embedding. Follow-up work (Wang et al., 2014; Trouillon et al., 2016; Sun et al., 2019) replaced the addition of real-valued vectors with other operations and other vector spaces, such as Hadamard product in a complex-valued vector space, and investigated regularizer and the effect of inverse relations on Knowledge Graph Completion (KGC) performance (Lacroix et al., 2018). ReFactorGNNs (Chen et al., 2022) try to combine the good KGC performance of these factorization-based models with the ease of feature integration and inductive applicability of GNNs into a single model for KGC. FedE (Chen et al., 2021) is a federated knowledge embedding framework that can use any knowledge graph embedding with multiple clients, each only having access to a part of the knowledge graph. This is not applicable to an inductive setting, as the test graph is not seen at any time during the embeddings.

The Unifying Model (Jia & Benson, 2022) fits a Markov Random Field to a graph, which can be similar to label propagation or a linear GCN depending on the attributes used. The model allows the sampling of new graphs from the training distribution, which differs from our task of providing embeddings to unseen nodes in graphs with no attributes.

## 2.2. GNNs

The most well-known graph neural network is GCN (Kipf & Welling, 2017), aggregating neighbors with weights based on their degree. GraphSAGE (Hamilton et al., 2017) modified GCN by considering the embedding of the current node separately from the neighbor aggregation and introducing sampling schemes to deal with large graphs. Other modifications of GCN used attention to assign different weights to neighbors (Velickovic et al., 2018) or make deep models easier to train by adding different kinds of skip connections (Chen et al., 2020; Xu et al., 2018; Sancak et al., 2024). For efficient models, besides reducing the number of message-passing layers (Wu et al., 2019), some works trained MLPs without using the edges for inference. Graph-MLP (Hu et al., 2021) incorporated edge information by pulling neighboring embeddings closer together with a contrastive loss. GLNN (Zhang et al., 2022) and NOSMOG (Tian et al., 2023) distill GNNs into MLPs. NOSMOG additionally increased robustness to noise with adversarial feature augmentation and concatenates Deep-Walk embeddings to the input for capturing more structured information.

Homophily is the characteristic of a graph in which neighboring nodes share the same class. Heterophilic graphs, i.e., graphs where neighbors usually belong to different classes, are an active area of research with work on how to measure homophily (Lim et al., 2021; Platonov et al., 2023a; Mironov & Prokhorenkova, 2024) and heterophilic datasets (Platonov et al., 2023b). Models that can better deal with heterophilic data use novel aggregations, consider multi-hop neighborhoods, distinguish homo- and heterophilic edges, or make the graph more homophilic using rewiring (Zhu et al., 2020; Lim et al., 2021; Kohn et al., 2024; Abu-El-Haija et al., 2019; Lell & Scherp, 2024; Chien et al., 2021; Luan et al., 2022; Bi et al., 2024).

## 3. Inductive N2V

The principal idea of our inductive N2V (iN2V) algorithm is to simply assign each test node the average embedding of its neighbors from the training set. This is repeated for multiple iterations to also deal with test nodes with longer distances to training nodes. The N2V embedding training is modified so that the embedding of training nodes are optimally prepared to induce embeddings to new nodes only seen during testing.

### 3.1. Notation and Formalization

Given a graph $G = (E, V)$ with node set $V \subset \mathbb{N}$ and edge set $E \subseteq V \times V$, with disjoint training, validation, and test sets $V_{train}, V_{val}, V_{test} \subset V$. $N(v) = \{x \in V \mid (v, x) \in E\}$ is the set of neighbors of $v$. For each node $v \in V$,

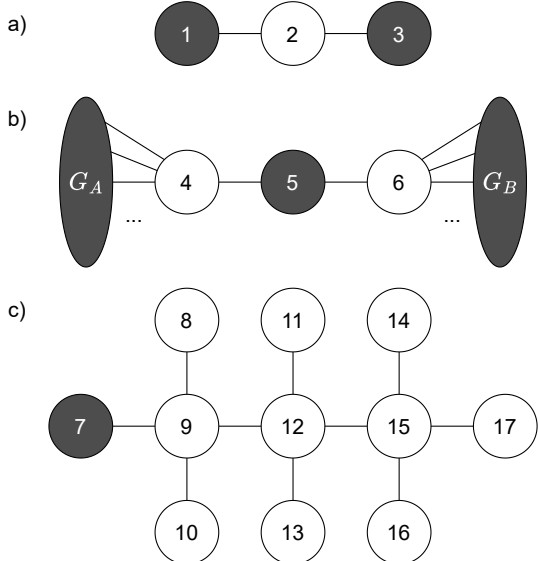

Figure 1. Three example graphs illustrate the post-hoc extensions to the white test nodes after obtaining embeddings for the gray training nodes in the inductive setup. Figure a) shows a simple example where node 2 obtains the average embedding of nodes 1 and 3. In Figure b) node 5 got a distant embedding during embedding training, but during the post-hoc extension it should be updated to move between the embeddings of the nodes from graphs $G_A$ and $G_B$. Finally, Figure c) illustrates how iN2V needs only four iterations to provide embeddings to all nodes.

we want to train an embedding $h_v \in \mathbb{R}^d$. In the inductive setting, only the subgraph induced by $V_{train}$ is available for training the embeddings. These embeddings then need to be extended to the remaining nodes $v_i \in V \setminus V_{train}$ in the validation and test set. As we use existing benchmark datasets, our nodes also have classification labels $Y \in \mathbb{N}^{|V|}$ and existing node features $X \in \mathbb{R}^{|V| \times \hat{d}}$.

### 3.2. Example

We motivate the different components of our post-hoc extension and illustrate their effect with the three example graphs a), b), and c) in Figure 1. Training nodes are shown in gray, and test nodes in white. The latter are hidden during training in the inductive case. In graph a) it is quite straightforward that $v_2$ should obtain the average of the embeddings of $v_1$ and $v_3$. In graph b), $G_A$ and $G_B$ are connected subgraphs of multiple training nodes with similar embeddings. When following the averaging idea, $v_4$ gets an embedding that is close to the average embedding in $G_A$ but skewed towards the embedding of $v_5$. Considering that the training embeddings were generated by N2V, the connected nodes in $G_A$ and $G_B$ got meaningful embeddings during training, while $v_5$ has a distant embedding as it has no neighbors in the training set and therefor only appeared as a negative sample during training. Contrary to Feature Propagation (Rossi

et al., 2022), in this case, it is useful to allow the adaption of input (training) embeddings. Following this line of thought while also maintaining some stability for the nodes with existing embeddings, each embedding should be a combination of itself and the average neighbor embedding. When doing multiple iterations of such an averaging procedure, the embedding of $v_5$ moves in between the embeddings of nodes of $G_A$ and $G_B$. However, too many iterations pose the challenge that all embeddings of individual nodes will converge to the average node embedding of that graph. Graph c) illustrates the challenge of extending embeddings into longer sequences of test nodes. When just averaging the neighborhood embedding for four iterations, the embedding of $v_{17} = v_7/4^3$ is close to zero. Feature Propagation handles this by keeping the input embeddings fixed and iterating many times until convergence. We already established the usefulness of adapting input embeddings in the example graph b). Therefore, we handle long sequences and high-degree nodes by considering only nodes that already have an embedding for averaging each iteration. That means that after the first iteration $v_9 = v_7$, after the second iteration $v_8, v_{13}, v_{10}, v_9 = v_7$, and so on. This leads to $v_{17} = v_7$ after only four iterations.

### 3.3. Generating Inductive Embeddings

We propose an iterative algorithm to extend trained embeddings to the unseen nodes. Let $h_v^{(t)}$ be the embedding of node $v$ after $t$ iterations of our algorithm. We use a lookup vector $s \in \{0,1\}^{|V|}$ with $s_v$ being the $v$-th element in the vector $s$ to keep track of which nodes already have embeddings and use $N_s(v) = \{x \in N(v) \mid s_x = 1\}$ to denote the set of neighbors which have an embedding. The mean embedding of a set of nodes $S$ is $m_S^{(t)} = \frac{1}{|S|} \sum_{v \in S} h_v^{(t)}$. For initialization, $h_v^{(0)}$ is set to the N2V embedding $h_u$ for training nodes $u \in V_{train}$ and to 0 for nodes $w \in V \setminus V_{train}$ not from the training set. The lookup vector is initialized with $s_u = 1$ and $s_w = 0$. Then $h^t$ is calculated from $h^{(t-1)}$ by:

$$
h_v^{(t)} = \begin{cases} h_v^{(t-1)} & \text{if } N_s(v) = \emptyset \quad \text{(1a)} \\ \lambda h_v^{(t-1)} + (1-\lambda)m_{N_s(v)}^{(t-1)} & \text{if } s_v = 1 \quad \text{(1b)} \\ m_{N_s(v)}^{(t-1)} & \text{else} \quad \text{(1c)} \end{cases}
$$

This means that if $v$ has no neighbor with an embedding, $h_v$ does not change (1a). If both $v$ and at least one neighbor of $v$ have an embedding, we calculate the convex combination of $h_v$ and the mean neighbor embedding $m_{N_s(v)}$ (1b). Note, for $\lambda = 1$, the embedding of a node will not change once it is set, and with $\lambda \leq 1$ all embeddings will be updated depending on their respective neighborhoods. If $v$ does not have an embedding but at least one neighbor has an

embedding, we set $v$'s embedding to the mean neighbor embedding (1c). This is done for multiple iterations. After each iteration, $s$ is updated by setting entries for nodes that got an embedding to 1. We do enough iterations such that each node with a path to at least one training node gets an embedding and additional $delay$-many iterations to update the embeddings of nodes like $v_5$ in the example Figure 1 b).

### 3.4. Boosting Inductive Performance

We propose two different approaches which modify the training to improve the generation of inductive embeddings.

**Sampling-based**   To promote embeddings that are better suited to the inductive extension of embeddings, we simulate a simple version of the post-hoc extensions during training. In each epoch, some features are replaced by their mean neighborhood embedding with probability $r$.

$$
h_v = \begin{cases} m_{N(v)} & \text{with probability } r \\ h_v & \text{else} \end{cases}
$$

**Loss-based**   In addition to the sampling-based approach, we also introduce a loss-based approach to prepare the embeddings for our inductive extension. When extending the trained embeddings to the inductive nodes during inference, we set the embedding of new nodes to their mean neighborhood embedding. The first loss promotes this relationship in the trained embeddings by pulling a node's own embedding closer to its mean neighborhood embedding:

$$
\mathcal{L}_{close}(v) = -\log(\sigma(h_v \cdot m_{N(v)})).
$$

A trivial solution to minimize this loss would be to assign identical embeddings to all neighboring nodes, so we add a second loss which promotes diversity in embeddings of the individual neighbors of each node:

$$
\mathcal{L}_{div}(v) = \frac{1}{|N(v)|^2} \sum_{u,w \in N(v)} \text{sim}(h_u, h_w),
$$

where sim is cosine similarity. The final loss for iN2V is

$$
\mathcal{L}(v) = \mathcal{L}_{n2v}(v) + \alpha \cdot \mathcal{L}_{close}(v) + \beta \cdot \mathcal{L}_{div}(v)
$$

with hyperparameters $\alpha$ and $\beta$. The N2V loss $\mathcal{L}_{n2v}$ is calculated using random walks. In each epoch, every node appears on average walks per node times walk length often in these random walks. The random walks are batched for training, but when the same node appears multiple times in a batch, the additional calculations for $\mathcal{L}_{close}$ and $\mathcal{L}_{div}$ are redundant. To reduce these redundant calculations, we

*Table 1.* Graph features, nodes, edges, classes, and adjusted homophily. Upper: homophilic, bottom: heterophilic graphs.

| Dataset | $|X|$ | $|V|$ | $|E|$ | $|C|$ | $hom_{adj}$ |
|---|---|---|---|---|---|
| Cora | 1 433 | 2 708 | 10 556 | 7 | 0.77 |
| CiteSeer | 3 703 | 3 327 | 9 104 | 6 | 0.67 |
| PubMed | 500 | 19 717 | 88 648 | 3 | 0.69 |
| Computers | 767 | 13 752 | 491 722 | 10 | 0.68 |
| Photo | 745 | 7 650 | 238 162 | 8 | 0.79 |
| WikiCS | 300 | 11 701 | 431 726 | 10 | 0.58 |
| Actor | 932 | 7 600 | 30 019 | 5 | 0.01 |
| Amazon-R. | 300 | 24 492 | 93 050 | 5 | 0.14 |
| Roman-E. | 300 | 22 662 | 32 927 | 18 | −0.05 |

sample the nodes to calculate $\mathcal{L}_{close}$ and $\mathcal{L}_{div}$ independent from the random walks such that each node is the center node for these losses once per epoch.

## 4. Experimental Apparatus

### 4.1. Datasets

We use the Cora (Sen et al., 2008), CiteSeer (Sen et al., 2008), and PubMed (Namata et al., 2012) citation graphs, the Computers (Shchur et al., 2018) and Photo (Shchur et al., 2018) co-purchase graphs, and the WikiCS Wikipedia page graph (Mernyei & Cangea, 2020). These graphs are homophilic, i.e., neighboring nodes usually share the same class. The following graphs are more heterophilic, i.e., neighboring nodes usually do not share the same class. Actor (Pei et al., 2020) is a Wikipedia co-occurrence graph, Amazon-ratings (Platonov et al., 2023b) is a co-purchase graph, and Roman-empire (Platonov et al., 2023b) is a text-based graph. Table 1 shows more details about the datasets. We report adjusted homophily as it accounts for class imbalance (Platonov et al., 2023a).

### 4.2. Procedure

We investigate how well iN2V works for differently sized data splits into training and unseen validation or test nodes. For all datasets, we use 5 splits of different sizes that always utilize the full dataset and have a validation and test set of the same size. The training set sizes are 10%, 20%, 40%, 60%, and 80%, with respective validation and test set sizes of 45%, 40%, 30%, 20%, and 10% of all nodes.

First, we prepare 10 splits from different random seeds for each of the five split sizes. Then we train N2V inductively on the training subgraphs. For the N2V hyperparameter search, we use a grid search on three of the ten splits. The embeddings from the training set are extended to the validation set using iN2V. The final N2V hyperparameters are chosen based on the validation accuracy of logistic regression on these embeddings. These hyperparameters are used to train

and store embeddings for all 10 splits. Then we evaluate the embeddings using MLP and GraphSAGE (Hamilton et al., 2017). Additionally, we investigate the concatenation of the extended N2V embeddings with the original graph features.

Regarding the extended embeddings, we compare different setups of iN2V. The first is the "frozen" setup ($\lambda = 1$), where embeddings do not change after being set. The second one is the best post-hoc setup, where $\lambda$ and $delay$ are searched as hyperparameters. The third and fourth ones combine the sampling-based and loss-based modifications to N2V with the best post-hoc setup. We compare these results with different baselines. The comparable baselines that use the same information as iN2V are plain N2V used inductively, i.e., only training nodes have embeddings, and using Feature Propagation (Rossi et al., 2022) to extend N2V embeddings to the test nodes. Using the original graph features and training N2V embeddings in a transductive setup are two more baselines. They are not directly comparable with iN2V as they use more information but are nevertheless useful for perspective. Additional experiments with other GNNs can be found in Appendix D.

### 4.3. Hyperparameter Optimization

We make all datasets undirected. For N2V, we use a context size of 10 for positive samples and 1 negative per positive sample. We use a batch size of 128 and early stopping with patience of 50 epochs. For every epoch, we sample 10 walks of length 20 per node. We do grid search over all combinations of $p$ and $q \in \{0.2, 1, 5\}$, embedding size $d \in \{64, 256\}$, and learning rate $\in \{0.1, 0.01, 0.001\}$. For the sampling based method, we try $r \in \{0.2, 0.4, 0.6, 0.8\}$. The loss weights $\alpha \in \{0, 0.1, 1, 10\}$ and $\beta \in \{0, 0.001, 0.01, 0.1\}$ are tuned separately from $r$. For Feature Propagation, we search the number of iterations in $\{10, 20, 40, 60\}$.

For MLP and GraphSAGE, we use grid search for the full 10 seeds per split. We search over all combination of number of layer $\in \{1, ... 5\}$, hidden size $\in \{64, 512\}$, learning rate $\in \{0.01, 0.001\}$, weight decay $\in \{0, 0.0001, 0.01\}$, dropout $\in \{0.2, 0.5, 0.8\}$, and whether to use jumping knowledge (Xu et al., 2018) connections.

## 5. Results and Discussion

### 5.1. Key Results

**Comparison of iN2V vs. Baselines**   Table 2 shows the performance of GraphSAGE using iN2V embeddings and compares them to normal N2V in the inductive setting (test nodes have no features) and N2V embeddings extended by Feature Propagation. We also report performance on the original graph features and transductive N2V embeddings to indicate how close the baselines and iN2V are to the

ideal case of no missing nodes or features. We can see that iN2V outperforms the comparable baselines in most cases. The notable exceptions are the Actor and Roman-empire datasets, where N2V embeddings generally perform poorly.

In some cases, using the iN2V embeddings outperforms the original features, e. g., in the $10\%$ training splits of Cora, Computers, and Amazon-Ratings. This implies that neighborhood information is more important for these datasets than the external information from the original features and that the original features do not have sufficient neighborhood information. Appendix B provides tables with the full results, including the $60\%$ train split for GraphSAGE and all MLP results. We observe for some datasets that the iN2V embeddings outperform the original features for more splits when using MLP as a model, as it can not generate neighborhood information internally as GraphSAGE does.

**Homophilic vs Heterophilic Datasets**   While we generally see quite good performance of models using N2V embeddings on homophilic datasets, this is not the case for heterophilic ones. This is expected since N2V provides similar embeddings for neighbors, which matches the homophilic label structure. On Actor, the models using N2V embeddings only learn to predict the largest class ($25.86\%$). On Roman-empire, the models only predict the largest class ($13.96\%$) when using a few training nodes and become slightly better when using most of the data during training. Roman-empire is the dataset with the lowest average degree in our comparison. This means the graph is prone to splitting into many disconnected subgraphs in the inductive setting. This makes this dataset challenging for training any random-walk-based embeddings. It also explains the rise in performance for the largest training split. Amazon-ratings is an interesting exception to this trend because even though it has a low adjusted homophily of $0.14$, N2V embeddings seem to work as well as on the homophilic datasets and even outperform using the original features for small training splits. Table 5 paints a similar picture when using MLP instead of GraphSAGE, with the difference that even for $80\%$ training data, the MLP does not perform better than just predicting the largest class for the Roman-empire dataset.

**Influence of GNN Choice and N2V Modifications**   Table 3 shows the performance aggregated over all datasets and train splits. The first two columns show the performance impact of the loss-based and sampling-based modifications to N2V when using only those embeddings as input for the GNNs. Our post-hoc extension, which can adapt input embeddings, outperforms Feature Propagation and the frozen post-hoc variant. This shows that adapting existing features is an important capability of iN2V. The loss-based and sampling-based modifications to N2V training provide

*Table 2.* Accuracy of best iN2V variant vs baseline embeddings. The underlying model is GraphSAGE. Gray numbers are not directly comparable as they use additional information, i. e., original features, or the transductive setup.

| Dataset | Percentage of training data | | | |
| --- | --- | --- | --- | --- |
| | 10% | 20% | 40% | 80% |
| Cora | | | | |
| N2V (inductive) | $42.18_{3.52}$ | $59.91_{4.15}$ | $75.07_{1.92}$ | $\mathbf{84.50}_{1.44}$ |
| Feature Propagation | $77.91_{2.62}$ | $79.48_{2.19}$ | $81.03_{1.85}$ | $84.13_{2.35}$ |
| iN2V (own) | $\mathbf{78.88}_{1.45}$ | $\mathbf{80.94}_{1.58}$ | $\mathbf{83.30}_{1.09}$ | $84.46_{2.08}$ |
| Original features | $75.27_{2.63}$ | $83.37_{1.17}$ | $86.23_{1.77}$ | $87.05_{1.20}$ |
| N2V (transductive) | $79.25_{1.45}$ | $81.66_{1.29}$ | $83.81_{0.95}$ | $86.01_{1.99}$ |
| Citeseer | | | | |
| N2V (inductive) | $34.17_{3.43}$ | $42.76_{2.15}$ | $56.12_{3.24}$ | $68.89_{2.61}$ |
| Feature Propagation | $56.85_{1.84}$ | $60.53_{2.03}$ | $\mathbf{63.03}_{2.21}$ | $\mathbf{69.76}_{1.82}$ |
| iN2V (own) | $\mathbf{57.88}_{0.91}$ | $\mathbf{60.78}_{1.78}$ | $63.02_{1.88}$ | $68.92_{2.07}$ |
| Original features | $69.85_{1.42}$ | $72.86_{0.98}$ | $74.93_{1.56}$ | $76.82_{1.77}$ |
| N2V (transductive) | $57.14_{1.51}$ | $61.33_{1.09}$ | $66.45_{1.60}$ | $72.76_{2.54}$ |
| Pubmed | | | | |
| N2V (inductive) | $66.02_{5.57}$ | $74.73_{2.36}$ | $79.84_{1.78}$ | $\mathbf{82.61}_{0.52}$ |
| Feature Propagation | $76.37_{0.62}$ | $77.72_{0.50}$ | $80.74_{0.72}$ | $82.43_{0.86}$ |
| iN2V (own) | $\mathbf{79.93}_{0.50}$ | $\mathbf{80.80}_{0.46}$ | $\mathbf{82.14}_{0.43}$ | $82.59_{0.63}$ |
| Original features | $85.95_{0.47}$ | $86.99_{0.28}$ | $88.32_{0.48}$ | $89.85_{0.56}$ |
| N2V (transductive) | $81.36_{0.49}$ | $82.20_{0.50}$ | $83.22_{0.34}$ | $83.66_{0.65}$ |
| Computers | | | | |
| N2V (inductive) | $77.64_{2.81}$ | $84.44_{0.84}$ | $87.18_{0.77}$ | $89.35_{0.70}$ |
| Feature Propagation | $82.79_{0.64}$ | $86.43_{0.63}$ | $89.40_{0.45}$ | $90.87_{0.64}$ |
| iN2V (own) | $\mathbf{88.36}_{0.58}$ | $\mathbf{89.67}_{0.40}$ | $\mathbf{90.84}_{0.37}$ | $\mathbf{91.38}_{0.51}$ |
| Original features | $87.52_{0.48}$ | $89.76_{0.40}$ | $91.12_{0.20}$ | $91.50_{0.48}$ |
| N2V (transductive) | $89.18_{0.38}$ | $90.16_{0.44}$ | $90.77_{0.39}$ | $91.16_{0.56}$ |
| Photo | | | | |
| N2V (inductive) | $85.48_{1.28}$ | $87.73_{1.40}$ | $90.98_{0.71}$ | $92.21_{0.96}$ |
| Feature Propagation | $87.43_{1.09}$ | $90.14_{0.34}$ | $91.57_{0.42}$ | $92.95_{0.79}$ |
| iN2V (own) | $\mathbf{90.51}_{0.72}$ | $\mathbf{91.70}_{0.44}$ | $\mathbf{92.37}_{0.46}$ | $\mathbf{93.08}_{0.77}$ |
| Original features | $93.74_{0.42}$ | $94.59_{0.37}$ | $95.27_{0.38}$ | $95.59_{0.77}$ |
| N2V (transductive) | $91.29_{0.41}$ | $92.29_{0.30}$ | $92.90_{0.45}$ | $93.33_{0.71}$ |
| WikiCS | | | | |
| N2V (inductive) | $67.78_{2.68}$ | $74.22_{1.65}$ | $78.21_{0.69}$ | $81.62_{0.86}$ |
| Feature Propagation | $74.77_{2.27}$ | $78.04_{1.00}$ | $80.17_{0.68}$ | $81.93_{1.08}$ |
| iN2V (own) | $\mathbf{78.91}_{0.61}$ | $\mathbf{80.19}_{0.70}$ | $\mathbf{81.28}_{0.61}$ | $\mathbf{82.37}_{1.01}$ |
| Original features | $80.75_{0.64}$ | $82.56_{0.81}$ | $84.28_{0.55}$ | $85.88_{0.70}$ |
| N2V (transductive) | $79.75_{0.41}$ | $80.93_{0.64}$ | $81.88_{0.55}$ | $82.81_{0.68}$ |
| Actor | | | | |
| N2V (inductive) | $25.14_{1.09}$ | $\mathbf{25.56}_{0.90}$ | $\mathbf{25.54}_{1.15}$ | $25.68_{1.42}$ |
| Feature Propagation | $\mathbf{25.22}_{1.29}$ | $25.39_{1.06}$ | $25.14_{0.59}$ | $25.05_{1.34}$ |
| iN2V (own) | $25.18_{0.97}$ | $25.50_{0.70}$ | $25.40_{0.99}$ | $\mathbf{25.79}_{2.27}$ |
| Original features | $31.77_{0.71}$ | $33.84_{0.91}$ | $36.48_{0.54}$ | $36.71_{1.23}$ |
| N2V (transductive) | $25.50_{0.76}$ | $25.56_{0.95}$ | $25.41_{0.74}$ | $24.55_{1.70}$ |
| Amazon-ratings | | | | |
| N2V (inductive) | $37.47_{0.47}$ | $40.69_{0.63}$ | $44.71_{0.85}$ | $49.47_{1.07}$ |
| Feature Propagation | $38.97_{0.76}$ | $41.68_{0.76}$ | $\mathbf{45.72}_{0.75}$ | $50.04_{1.42}$ |
| iN2V (own) | $\mathbf{40.02}_{0.79}$ | $\mathbf{42.01}_{0.48}$ | $45.48_{0.57}$ | $\mathbf{50.38}_{2.22}$ |
| Original features | $39.20_{1.06}$ | $41.66_{0.70}$ | $48.07_{0.58}$ | $57.34_{0.97}$ |
| N2V (transductive) | $41.82_{0.66}$ | $43.69_{0.52}$ | $46.31_{0.78}$ | $49.80_{0.94}$ |
| Roman-empire | | | | |
| N2V (inductive) | $\mathbf{13.96}_{0.35}$ | $\mathbf{14.10}_{0.32}$ | $\mathbf{15.77}_{0.76}$ | $16.81_{2.94}$ |
| Feature Propagation | $13.23_{1.44}$ | $13.89_{0.44}$ | $15.49_{0.67}$ | $\mathbf{21.97}_{1.25}$ |
| iN2V (own) | $13.79_{0.39}$ | $13.86_{0.29}$ | $14.48_{0.63}$ | $18.55_{1.23}$ |
| Original features | $66.09_{0.75}$ | $70.28_{0.63}$ | $74.41_{0.47}$ | $82.90_{1.09}$ |
| N2V (transductive) | $13.82_{0.22}$ | $13.86_{0.37}$ | $15.35_{1.53}$ | $27.46_{1.36}$ |

*Table 3.* Effect of model and input, test accuracy averaged over all datasets and splits.

| Input | embed | | embed ‖ feat. | |
| Model | MLP | SAGE | MLP | SAGE |
|---|---|---|---|---|
| N2V (inductive) | 27.91 | 59.45 | 68.72 | 73.51 |
| Feature Propagation | 62.97 | 63.20 | 73.17 | 74.09 |
| frozen ($\lambda = 1$) | 61.74 | 62.88 | 72.40 | 73.46 |
| post-hoc | 64.38 | 63.75 | 73.99 | 74.24 |
| p-h w losses | 64.57 | 63.95 | 74.04 | 74.32 |
| p-h w sampling | 64.42 | 63.83 | 73.95 | 74.02 |
| Original features | 71.01 | 74.73 | 71.01 | 74.73 |
| N2V(transductive) | 63.89 | 65.07 | 74.15 | 75.57 |

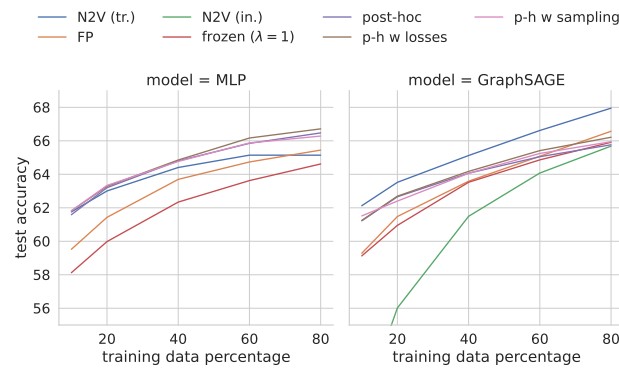

*Figure 2.* Influence of amount of training data. Results are averaged over all datasets.

a small boost to performance. When we compare the results for MLP vs GraphSAGE, we see that the differences are bigger when using an MLP. Interestingly, MLP outperforms GraphSAGE when using post-hoc embeddings. This is the case because GraphSAGE's main advantage over MLP is its ability to aggregate neighborhood information internally. N2V already encodes this information in the embeddings, and additionally, our post-hoc extension is similar to the aggregation performed by models like GraphSAGE. This effect might also be reinforced by the fact that we use logistic regression on the embeddings for the N2V hyperparameter selection, which is closer to an MLP than it is to Graph-SAGE. Detailed results per dataset and split are shown in Appendix B.

**Combining N2V embeddings and Original Features** For datasets that already have features, N2V embeddings can be used to provide additional structural information to GNNs. The third and fourth columns of Table 3 show our results in an aggregated way, while per dataset results can be found in Appendix C. Using both the N2V embeddings and original features as input for the models increases overall performance by about 10 points compared to just using N2V embeddings. The increase is bigger for the N2V (inductive) baseline, as the test nodes do not have N2V embeddings in that scenario. When just using N2V embeddings, MLP has a slightly higher average performance than GraphSAGE in most setups; this switches to GraphSAGE having a slight lead over MLP when using both as input.

Compared to the original features baseline, GraphSAGE actually loses an average of .5 performance points when also using the N2V as input, whereas MLP gains 3 points. This again shows that MLP can benefit from more structural information while GraphSAGE is already capable of aggregating this information by itself. The increase heavily depends on the dataset; see detailed per-dataset results in Appendix C. Both models gain over 50 points on Roman-empire and have only smaller gains on the other datasets.

These dataset-specific differences are explained by the usefulness of N2V embeddings vs the usefulness of the original features. On Roman-empire, the graph is close to a sequence of words with few additional edges; the neighborhood information encoded by N2V embeddings does not bring much useful information.

**Influence of Amount of Training Data** Figure 2 visualizes the effect of the training set size averaged over all datasets. In general, performance increases with more training data. The only exception is N2V in the inductive setting when using MLP, as the test nodes do not have embeddings, and performance stays at a random guessing level. Graph-SAGE can compensate for the missing test embeddings. While starting with low performance for little training data, it is close to the methods that actually extend embeddings to test nodes for $80\%$ training data. The MLP model with Feature Propagation or frozen post-hoc ($\lambda = 1$) consistently performs below the other post-hoc variants. GraphSAGE with Feature Propagation and frozen post-hoc catch up and outperform some other variants when using $80\%$ training data. Another interesting observation is that for MLP, the transductive N2V starts in line with the post-hoc variants but rises less when the training data increases. For GraphSAGE, transductive N2V already starts higher than the post-hoc variants and even widens the lead when the amount of training data increases.

**Applying of Loss and Sampling Modification with Feature Propagation** While our modifications to the N2V training procedure are motivated by our post-hoc extension, these two parts act independently. The loss-based and sampling-based modifications change the generated N2V embeddings, and the post-hoc algorithm extends these independent of their exact values. This means that instead of using our post-hoc algorithm, we can also use Feature Propagation to extend the modified N2V embeddings. Ta-

*Table 4.* Applying our loss- and sampling-based N2V modifications when using Feature Propagation to extend embeddings.

| model | MLP | GraphSAGE |
|---|---|---|
| FP | 62.97 | 63.20 |
| FP w loss | 63.04 | 63.22 |
| FP w sampling | 62.98 | 62.98 |

ble 4 shows the results of this experiment averaged over all datasets. Feature Propagation gains around .05 points with the loss modification of the embeddings for both models. The sampling-based modification does not change the MLP performance, but it reduces the GraphSAGE performance by .2 points. This is lower than the average of .2 points gained with post-hoc with the losses and .06 gained by the sampling-based modification in Table 3 and shows the synergy of our post-hoc methods with the N2V modifications.

### 5.2. Ablation Study on the Post-hoc Method

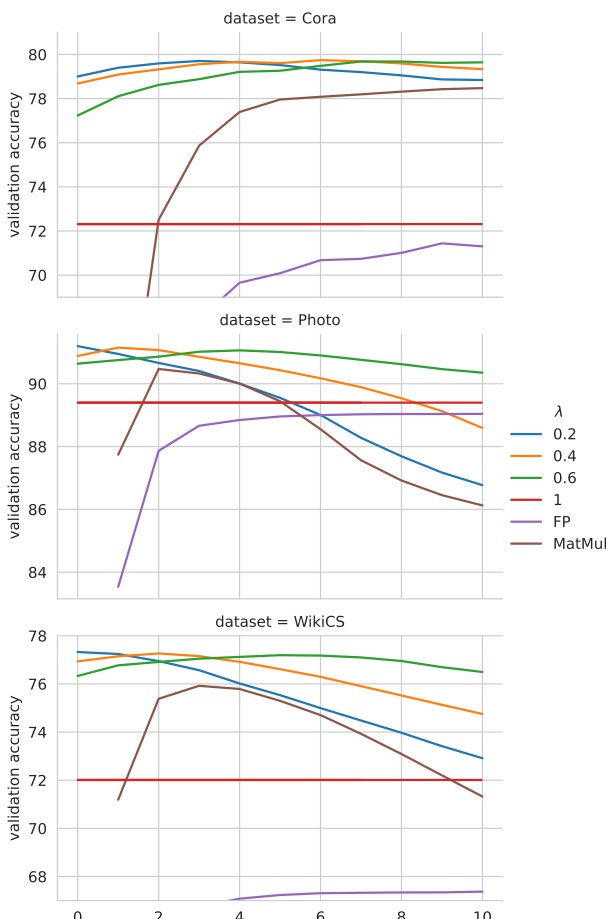

*Figure 3.* Ablation of the effect of $\lambda$ for different delays vs Feature Propagation and MatMul.

We perform ablation studies on our post-hoc method and the loss modification. For this, we use the $40\%$ training split, set the N2V hyperparameters $p$ and $q$ to 1, embeddings size to 256, and learning rate to 0.01. For a sensitivity analysis of these hyperparameters, see Appendix A.

Figure 3 shows the effect of $\lambda$ and $delay$ in our post-hoc extension, Feature Propagation, as well as a MatMul baseline that multiplies the embeddings matrix $iter$-many times with the adjacency matrix. When increasing $iter$ or $delay$, the post-hoc methods and MatMul increase to a maximum, which depends on $\lambda$, and then drop off again. This nicely shows the trade-off we discuss in Section 3. More iterations allow an adaption to new paths from the test split, but too many iterations lead to a convergence of all embeddings to a graph average. For $\lambda = 1$, this is not the case as embeddings do not change once they are set, which means that the embedding and performance are fixed for $delay \geq 0$. As Feature Propagation keeps the training embeddings fixed, its performance increases with more iterations. Overall, we can observe that adapting training embeddings is important, as post-hoc with $\lambda < 1$ and MatMul outperform both Feature Propagation and post-hoc with $\lambda = 1$.

Figure 4 shows the effect of the loss weights $\alpha$ and $\beta$ on logistic regression validation accuracy. As we already saw in our main results, the loss-based N2V modification gives a small performance improvement. The figure suggests that $\alpha$ has a bigger influence on the performance as long as $\beta$ is not too high.

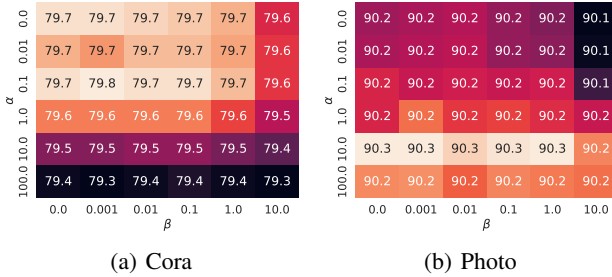

(a) Cora          (b) Photo

*Figure 4.* Validation accuracy for different weights for $\alpha$ and $\beta$.

## 6. Limitations and Future Work

Our iN2V algorithm uses N2V and shares some of the limitations. N2V embeddings do not add any information for datasets where neighborhood structures alone are irrelevant. We have observed this case with the Actor and Roman-Empire datasets, where the performance when using only N2V embeddings as model input was close to predicting the largest class. Many molecule datasets consist of thousands of individual subgraphs, where the train and test split is done on a per-graph level. In cases where there is no path between the training and test nodes, our post-hoc extensions

(like Feature Propagation) cannot provide embeddings for test nodes in an inductive setting. Our method generally can only provide embeddings to test nodes with a path to at least one train node. If this affects only a few nodes, then iN2V still shows good performance. We have shown this with Cora and Citeseer, which have $78$ and $438$ components, respectively. Some of these components have no nodes in the training set when using the small training splits.

As our post-hoc extension is flexible and not tied to N2V, other shallow embedding methods can be used for datasets where the neighborhood structure does not provide helpful information. For example, one could use struc2vec, which focuses on the similarity of neighborhood degree distributions, to obtain better embeddings for heterophilic datasets. In this case, it might be better to utilize the neighborhood-similarity-based graph struc2vec builds for the post-hoc algorithm. Another limitation and avenue for future work is that we used random sampling to create the dataset splits. Using random splits is important for a fair evaluation of different embeddings and models (Shchur et al., 2018), but in the inductive case, this leads to some unconnected training nodes in the splits with few training nodes. Especially for those splits, it might be beneficial to use a biased sampler that prefers nodes with edges to already sampled nodes to obtain better-connected training sets.

Our iN2V is not limited to simple graphs; it can already deal with multi-edges and self-loops. Self-loops increase the influence of $h_v$ in Equation 1b to more than $\lambda$ as it also appears in $m_{N_s(v)}$. Edge weights could be incorporated into the mean neighbor embedding $m_{N_s(v)}$ by replacing the mean with a weighted sum using normalized edge weights. For KGs, iN2V could be used as-is or by replacing N2V with a KG-focused embedding like TransE that embeds vertices and relations. When doing so, Equations 1b and 1c have to be adapted to incorporate the relation embeddings.

## 7. Conclusion

We introduced iN2V, a general post-hoc extension to induce embeddings to unseen nodes in the inductive setup. We modified the training algorithm of N2V to obtain embeddings better suited to this induction. Our extensive experiments on different datasets, training splits, and using different GNNs on the embeddings showed that iN2V works well and beats the comparable baselines. For some datasets and splits, iN2V even outperforms using the original graph features. In our detailed discussion and ablation, we have shown that our post-hoc extensions perform remarkably well. At the same time, the modifications of the N2V training have a smaller influence on the final performance. Our experiments also showed general limitations of N2V-based approaches for some of the heterophilic datasets, where the performance remained close to random.

## Acknowledgements

This work was performed on the computational resource bwUniCluster funded by the Ministry of Science, Research and the Arts Baden-Württemberg and the Universities of the State of Baden-Württemberg, Germany, within the framework program bwHPC

## Impact Statement

This paper aims to transfer shallow embedding methods like N2V to the inductive setup, where only the graph induced by training nodes is known during training. Our work can be used to apply GNNs to new nodes in the inductive setup, for example new users in social networks or new products in co-purchase graphs, and outperforms existing methods. While our work has implications, particularly in improving the handling of nodes with missing features or in dealing with feature-less graphs, we believe that no specific societal consequences require immediate emphasis in this context.

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

## Supplemental Materials

## A. Hyperparameter Sensitivity

This section follows up on Section 5.2. Figure 5 shows the sensitivity of validation accuracy to the iN2V learning rate and embedding dimension while all other hyperparameters (including $\lambda$, $\alpha$, $\beta$, and $r$) are aggregated in the densities. We can see that the embedding dimension has the biggest influence on the performance, especially for the Cora dataset. For Figure 6, which shows the sensitivity to $p$ and $q$, we therefore fix the embedding dimension to 256.

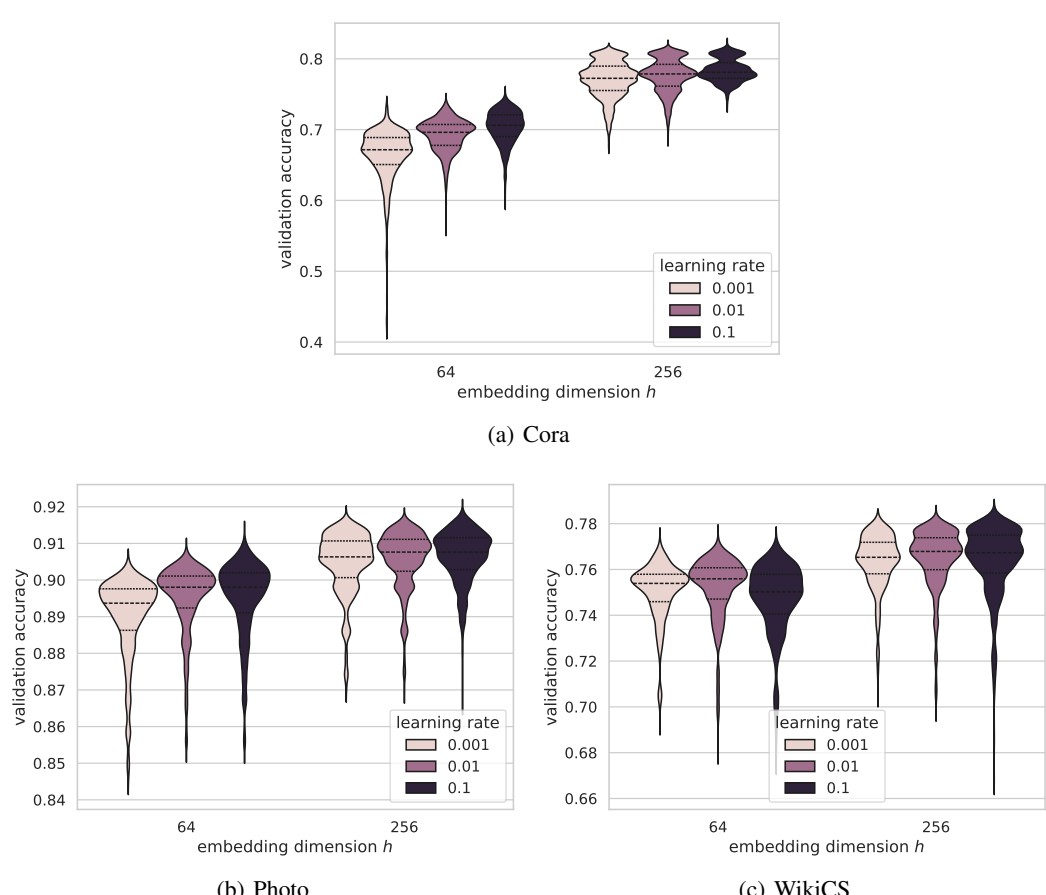

(a) Cora

(b) Photo

(c) WikiCS

*Figure 5.* Sensitivity to learning rate and embedding dimension.

## B. Complete Result Tables

Tables 5 and 6 show the comparison of MLP and GraphSAGE on the iN2V embeddings vs baselines. N2V, applied in the inductive setting, and Feature Propagation are comparable because they have access to the same information during training, while original features and N2V in the transductive setting utilize more information (the original features and all nodes during training). Tables 7 and 8 show the MLP and GraphSAGE results when using different iN2V settings.

## C. Combining Original Graph Features with Trained N2V Embeddings

As shown by related work, shallow embeddings like DeepWalk or N2V can be combined with the original graph features to improve the performance of GNNs. This is especially helpful for MLPs who do not have access to structure information when only using the default graph features (Tian et al., 2023). Tables 9 and 10 compare the iN2V embeddings vs baselines with the original graph features concatenated to the input embeddings. Tables 11 and 12 compare the different iN2V setups when concatenating the embeddings with the original graph features.

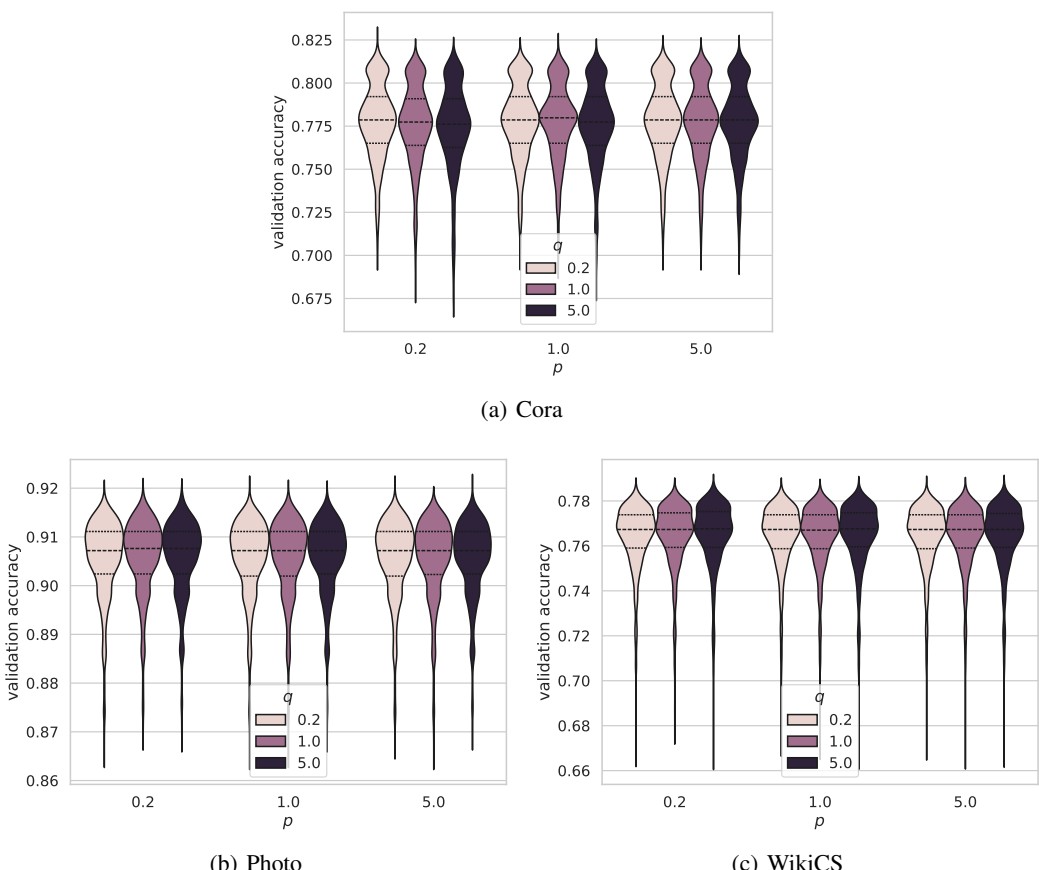

(a) Cora

(b) Photo

(c) WikiCS

*Figure 6.* Sensitivity to the N2V $p$ and $q$ hyperparameters.

## D. Using other GNNs

The effectiveness of iN2V is not limited by the chosen GNN. To demonstrate this, we additionally performed all experiments with GAT (Velickovic et al., 2018), see Tables 13, 15, 17, and 19 and with GIN (Xu et al., 2019), see Tables 14, 16, 18, and 20. For GAT, we set the number of attention heads to 8. Otherwise, we used the same hyperparameter tuning procedure for both GAT and GIN as for GraphSAGE.

The results for GIN and GAT are in line with the GraphSAGE results. iN2V outperforms all baselines for most splits on all datasets except Actor, where all N2V-based results amount to guessing the largest class. Averaging the iN2V results over all datasets, splits, and whether to concatenate the iN2V embedding with the original graph features, MLP remains the best model with an average accuracy of 69.37 points. GraphSAGE reaches an average of 69.25 points, GIN of 69.08 points, and GAT of 68.64 points. This shows that an MLP can outperform classical GNNs without distillation or contrastive learning if enough structure information is provided with the features.

When considering all four GNNs, iN2V outperforms FP by 0.8 points on homophilic and 0.3 points on heterophilic datasets. iN2V outperforms FP by 1.3 points when using MLP as a classification model and by 0.4 points when using message-passing GNNs. iN2V outperforms FP by 0.9 points when using only trained embeddings and by 0.3 points when using both trained embeddings and the original graph features as input. Finally, for the 10% and 20% training splits, iN2V leads over FP by 0.9 points in contrast to the 0.4 point lead for the 60% and 80% training splits.

*Table 5.* Comparison of best iN2V variant vs baselines; MLP accuracy. Gray numbers are not directly comparable as they use additional information (graph features/transductive setup).

| Dataset | Percentage of training data | | | | |
|---|---|---|---|---|---|
| | 10% | 20% | 40% | 60% | 80% |
| **Cora** | | | | | |
| N2V (inductive) | $30.04_{1.06}$ | $30.07_{1.15}$ | $30.12_{1.05}$ | $29.67_{1.04}$ | $30.70_{2.14}$ |
| Feature Propagation | $78.54_{1.42}$ | $81.11_{1.17}$ | $82.38_{1.01}$ | $83.76_{1.00}$ | $85.13_{2.27}$ |
| **iN2V** (own) | $\mathbf{79.84}_{1.27}$ | $\mathbf{81.70}_{1.19}$ | $\mathbf{83.93}_{1.16}$ | $\mathbf{84.69}_{0.95}$ | $\mathbf{85.57}_{2.22}$ |
| Original features | $66.98_{1.70}$ | $71.34_{1.79}$ | $75.94_{1.38}$ | $78.78_{1.11}$ | $79.23_{1.95}$ |
| N2V (transductive) | $78.44_{1.15}$ | $80.84_{1.49}$ | $82.82_{1.10}$ | $83.75_{1.20}$ | $83.10_{1.86}$ |
| **Citeseer** | | | | | |
| N2V (inductive) | $19.81_{1.75}$ | $21.43_{1.00}$ | $21.11_{1.27}$ | $21.10_{1.46}$ | $19.82_{3.05}$ |
| Feature Propagation | $56.76_{2.21}$ | $60.14_{1.58}$ | $65.73_{1.37}$ | $68.54_{1.53}$ | $72.16_{2.65}$ |
| **iN2V** (own) | $\mathbf{57.88}_{1.35}$ | $\mathbf{61.67}_{1.69}$ | $\mathbf{66.47}_{1.91}$ | $\mathbf{70.62}_{1.37}$ | $\mathbf{73.27}_{1.73}$ |
| Original features | $66.60_{1.56}$ | $70.53_{1.04}$ | $73.23_{1.17}$ | $74.56_{1.04}$ | $75.20_{2.10}$ |
| N2V (transductive) | $56.43_{2.01}$ | $60.65_{1.81}$ | $64.77_{1.77}$ | $68.80_{1.69}$ | $70.09_{1.32}$ |
| **Pubmed** | | | | | |
| N2V (inductive) | $39.45_{0.69}$ | $39.69_{0.77}$ | $40.15_{0.42}$ | $40.01_{0.71}$ | $39.30_{1.06}$ |
| Feature Propagation | $78.44_{0.44}$ | $78.99_{0.49}$ | $81.02_{0.73}$ | $81.91_{0.64}$ | $82.56_{0.95}$ |
| **iN2V** (own) | $\mathbf{80.42}_{0.44}$ | $\mathbf{81.77}_{0.40}$ | $\mathbf{82.78}_{0.58}$ | $\mathbf{83.19}_{0.44}$ | $\mathbf{83.26}_{0.87}$ |
| Original features | $84.11_{0.21}$ | $86.05_{0.40}$ | $87.72_{0.40}$ | $88.63_{0.60}$ | $89.10_{0.63}$ |
| N2V (transductive) | $81.03_{0.51}$ | $81.76_{0.52}$ | $82.72_{0.39}$ | $82.96_{0.51}$ | $83.17_{0.76}$ |
| **Computers** | | | | | |
| N2V (inductive) | $37.39_{0.32}$ | $37.33_{0.28}$ | $37.31_{0.66}$ | $37.51_{0.75}$ | $37.72_{0.87}$ |
| Feature Propagation | $81.90_{0.93}$ | $86.16_{0.66}$ | $89.20_{0.35}$ | $90.27_{0.60}$ | $90.71_{0.61}$ |
| **iN2V** (own) | $\mathbf{88.70}_{0.47}$ | $\mathbf{90.04}_{0.51}$ | $\mathbf{91.10}_{0.47}$ | $\mathbf{91.51}_{0.48}$ | $\mathbf{91.55}_{0.52}$ |
| Original features | $82.15_{0.63}$ | $84.08_{0.65}$ | $85.57_{0.53}$ | $86.00_{0.77}$ | $86.56_{0.58}$ |
| N2V (transductive) | $88.58_{0.54}$ | $89.63_{0.39}$ | $90.18_{0.41}$ | $90.51_{0.35}$ | $90.39_{0.57}$ |
| **Photo** | | | | | |
| N2V (inductive) | $25.51_{0.46}$ | $24.86_{1.30}$ | $23.66_{1.99}$ | $22.96_{1.95}$ | $23.15_{2.24}$ |
| Feature Propagation | $87.37_{0.85}$ | $89.38_{0.58}$ | $91.49_{0.51}$ | $92.25_{0.53}$ | $92.64_{0.77}$ |
| **iN2V** (own) | $\mathbf{90.70}_{0.50}$ | $\mathbf{91.74}_{0.36}$ | $\mathbf{92.60}_{0.55}$ | $\mathbf{92.86}_{0.72}$ | $\mathbf{93.07}_{0.68}$ |
| Original features | $89.32_{0.59}$ | $90.85_{0.55}$ | $91.96_{0.55}$ | $92.41_{0.60}$ | $92.92_{0.87}$ |
| N2V (transductive) | $91.23_{0.44}$ | $91.88_{0.47}$ | $92.67_{0.50}$ | $92.73_{0.61}$ | $92.44_{0.67}$ |
| **WikiCS** | | | | | |
| N2V (inductive) | $22.99_{0.41}$ | $22.96_{0.49}$ | $23.05_{0.51}$ | $23.12_{0.76}$ | $22.82_{1.26}$ |
| Feature Propagation | $73.59_{1.36}$ | $75.83_{1.53}$ | $78.83_{0.82}$ | $80.58_{0.57}$ | $80.97_{0.87}$ |
| **iN2V** (own) | $\mathbf{79.21}_{0.59}$ | $\mathbf{80.37}_{0.80}$ | $\mathbf{81.47}_{0.52}$ | $\mathbf{82.12}_{0.68}$ | $\mathbf{82.76}_{0.83}$ |
| Original features | $76.85_{0.64}$ | $78.76_{0.66}$ | $80.43_{0.64}$ | $81.56_{0.78}$ | $82.38_{1.06}$ |
| N2V (transductive) | $79.23_{0.68}$ | $80.51_{0.59}$ | $81.47_{0.56}$ | $81.86_{0.53}$ | $81.92_{0.66}$ |
| **Actor** | | | | | |
| N2V (inductive) | $25.41_{1.39}$ | $\mathbf{25.55}_{1.16}$ | $25.69_{1.08}$ | $\mathbf{26.05}_{0.51}$ | $\mathbf{26.57}_{0.79}$ |
| Feature Propagation | $25.56_{1.04}$ | $25.15_{0.94}$ | $\mathbf{25.83}_{0.65}$ | $25.18_{0.53}$ | $25.00_{1.05}$ |
| **iN2V** (own) | $\mathbf{25.78}_{0.76}$ | $25.45_{0.82}$ | $25.00_{1.35}$ | $25.14_{0.76}$ | $25.51_{2.00}$ |
| Original features | $35.13_{0.51}$ | $36.20_{0.67}$ | $37.79_{0.62}$ | $38.75_{0.86}$ | $37.87_{1.53}$ |
| N2V (transductive) | $25.08_{1.04}$ | $25.34_{0.99}$ | $25.51_{1.01}$ | $25.12_{0.89}$ | $23.92_{1.12}$ |
| **Amazon-ratings** | | | | | |
| N2V (inductive) | $36.91_{0.30}$ | $36.93_{0.35}$ | $36.74_{0.37}$ | $36.78_{0.52}$ | $36.54_{0.89}$ |
| Feature Propagation | $39.70_{0.60}$ | $42.18_{0.81}$ | $44.90_{0.62}$ | $46.24_{0.76}$ | $46.32_{0.96}$ |
| **iN2V** (own) | $\mathbf{41.41}_{0.87}$ | $\mathbf{43.75}_{0.54}$ | $\mathbf{46.52}_{0.77}$ | $\mathbf{51.66}_{0.31}$ | $\mathbf{51.93}_{0.78}$ |
| Original features | $37.80_{0.51}$ | $41.77_{0.61}$ | $47.20_{0.56}$ | $50.82_{0.77}$ | $54.00_{0.69}$ |
| N2V (transductive) | $41.97_{0.50}$ | $42.71_{0.48}$ | $45.57_{0.47}$ | $46.58_{0.60}$ | $47.97_{0.89}$ |
| **Roman-empire** | | | | | |
| N2V (inductive) | $13.34_{1.25}$ | $13.55_{0.97}$ | $\mathbf{14.00}_{0.28}$ | $13.99_{0.40}$ | $13.27_{1.67}$ |
| Feature Propagation | $13.83_{0.25}$ | $\mathbf{13.93}_{0.26}$ | $13.90_{0.29}$ | $13.88_{0.52}$ | $13.53_{0.78}$ |
| **iN2V** (own) | $\mathbf{13.89}_{0.26}$ | $13.90_{0.27}$ | $13.79_{0.24}$ | $\mathbf{14.23}_{0.69}$ | $\mathbf{14.27}_{0.94}$ |
| Original features | $63.46_{0.43}$ | $65.05_{0.31}$ | $66.47_{0.41}$ | $66.83_{0.71}$ | $66.55_{1.02}$ |
| N2V (transductive) | $13.80_{0.26}$ | $13.74_{0.39}$ | $13.95_{0.25}$ | $13.98_{0.49}$ | $13.27_{0.90}$ |

*Table 6.* Comparison of best iN2V variant vs baselines; GraphSAGE accuracy. Gray numbers are not directly comparable as they use additional information (graph features/transductive setup).

| Dataset | Percentage of training data | | | | |
|---|---|---|---|---|---|
| | 10% | 20% | 40% | 60% | 80% |
| **Cora** | | | | | |
| N2V (inductive) | $42.18_{3.52}$ | $59.91_{4.15}$ | $75.07_{1.92}$ | $81.31_{1.56}$ | $\mathbf{84.50}_{1.44}$ |
| Feature Propagation | $77.91_{2.62}$ | $79.48_{2.19}$ | $81.03_{1.85}$ | $\mathbf{84.00}_{1.00}$ | $84.13_{2.35}$ |
| **iN2V** (own) | $\mathbf{78.88}_{1.45}$ | $\mathbf{80.94}_{1.58}$ | $\mathbf{83.30}_{1.09}$ | $83.73_{1.13}$ | $84.46_{2.08}$ |
| Original features | $75.27_{2.63}$ | $83.37_{1.17}$ | $86.23_{1.77}$ | $86.99_{0.86}$ | $87.05_{1.20}$ |
| N2V (transductive) | $79.25_{1.45}$ | $81.66_{1.29}$ | $83.81_{0.95}$ | $85.24_{0.83}$ | $86.01_{1.99}$ |
| **Citeseer** | | | | | |
| N2V (inductive) | $34.17_{3.43}$ | $42.76_{2.15}$ | $56.12_{3.24}$ | $63.61_{2.57}$ | $68.89_{2.61}$ |
| Feature Propagation | $56.85_{1.84}$ | $60.53_{2.03}$ | $\mathbf{63.03}_{2.21}$ | $66.51_{2.18}$ | $\mathbf{69.76}_{1.82}$ |
| **iN2V** (own) | $\mathbf{57.88}_{0.91}$ | $\mathbf{60.78}_{1.78}$ | $63.02_{1.88}$ | $\mathbf{66.81}_{1.42}$ | $68.92_{2.07}$ |
| Original features | $69.85_{1.42}$ | $72.86_{0.98}$ | $74.93_{1.56}$ | $76.39_{1.05}$ | $76.82_{1.77}$ |
| N2V (transductive) | $57.14_{1.51}$ | $61.33_{1.09}$ | $66.45_{1.60}$ | $70.14_{1.26}$ | $72.76_{2.54}$ |
| **Pubmed** | | | | | |
| N2V (inductive) | $66.02_{5.57}$ | $74.73_{2.36}$ | $79.84_{1.78}$ | $81.85_{0.61}$ | $\mathbf{82.61}_{0.52}$ |
| Feature Propagation | $76.37_{0.62}$ | $77.72_{0.50}$ | $80.74_{0.72}$ | $82.00_{0.53}$ | $82.43_{0.86}$ |
| **iN2V** (own) | $\mathbf{79.93}_{0.50}$ | $\mathbf{80.80}_{0.46}$ | $\mathbf{82.14}_{0.43}$ | $\mathbf{82.81}_{0.46}$ | $82.59_{0.63}$ |
| Original features | $85.95_{0.47}$ | $86.99_{0.28}$ | $88.32_{0.48}$ | $89.25_{0.44}$ | $89.85_{0.56}$ |
| N2V (transductive) | $81.36_{0.49}$ | $82.20_{0.50}$ | $83.22_{0.34}$ | $83.57_{0.50}$ | $83.66_{0.65}$ |
| **Computers** | | | | | |
| N2V (inductive) | $77.64_{2.81}$ | $84.44_{0.84}$ | $87.18_{0.77}$ | $89.21_{0.42}$ | $89.35_{0.70}$ |
| Feature Propagation | $82.79_{0.64}$ | $86.43_{0.63}$ | $89.40_{0.45}$ | $90.43_{0.40}$ | $90.87_{0.64}$ |
| **iN2V** (own) | $\mathbf{88.36}_{0.58}$ | $\mathbf{89.67}_{0.40}$ | $\mathbf{90.84}_{0.37}$ | $\mathbf{91.08}_{0.36}$ | $\mathbf{91.38}_{0.51}$ |
| Original features | $87.52_{0.48}$ | $89.76_{0.40}$ | $91.12_{0.20}$ | $91.49_{0.58}$ | $91.50_{0.48}$ |
| N2V (transductive) | $89.18_{0.38}$ | $90.16_{0.44}$ | $90.77_{0.39}$ | $91.19_{0.39}$ | $91.16_{0.56}$ |
| **Photo** | | | | | |
| N2V (inductive) | $85.48_{1.28}$ | $87.73_{1.40}$ | $90.98_{0.71}$ | $91.92_{0.63}$ | $92.21_{0.96}$ |
| Feature Propagation | $87.43_{1.09}$ | $90.14_{0.34}$ | $91.57_{0.42}$ | $92.48_{0.51}$ | $92.95_{0.79}$ |
| **iN2V** (own) | $\mathbf{90.51}_{0.72}$ | $\mathbf{91.70}_{0.44}$ | $\mathbf{92.37}_{0.46}$ | $\mathbf{92.82}_{0.63}$ | $\mathbf{93.08}_{0.77}$ |
| Original features | $93.74_{0.42}$ | $94.59_{0.37}$ | $95.27_{0.38}$ | $95.52_{0.52}$ | $95.59_{0.77}$ |
| N2V (transductive) | $91.29_{0.41}$ | $92.29_{0.30}$ | $92.90_{0.45}$ | $93.01_{0.67}$ | $93.33_{0.71}$ |
| **WikiCS** | | | | | |
| N2V (inductive) | $67.78_{2.68}$ | $74.22_{1.65}$ | $78.21_{0.69}$ | $80.40_{0.45}$ | $81.62_{0.86}$ |
| Feature Propagation | $74.77_{2.27}$ | $78.04_{1.00}$ | $80.17_{0.68}$ | $81.03_{0.68}$ | $81.93_{1.08}$ |
| **iN2V** (own) | $\mathbf{78.91}_{0.61}$ | $\mathbf{80.19}_{0.70}$ | $\mathbf{81.28}_{0.61}$ | $\mathbf{81.73}_{0.76}$ | $\mathbf{82.37}_{1.01}$ |
| Original features | $80.75_{0.64}$ | $82.56_{0.81}$ | $84.28_{0.55}$ | $85.21_{0.63}$ | $85.88_{0.70}$ |
| N2V (transductive) | $79.75_{0.41}$ | $80.93_{0.64}$ | $81.88_{0.55}$ | $82.30_{0.65}$ | $82.81_{0.68}$ |
| **Actor** | | | | | |
| N2V (inductive) | $25.14_{1.09}$ | $\mathbf{25.56}_{0.90}$ | $\mathbf{25.54}_{1.15}$ | $\mathbf{25.42}_{1.03}$ | $25.68_{1.42}$ |
| Feature Propagation | $\mathbf{25.22}_{1.29}$ | $25.39_{1.06}$ | $25.14_{0.59}$ | $25.09_{1.42}$ | $25.05_{1.34}$ |
| **iN2V** (own) | $25.18_{0.97}$ | $25.50_{0.76}$ | $25.40_{0.99}$ | $25.16_{0.92}$ | $\mathbf{25.79}_{2.27}$ |
| Original features | $31.77_{0.71}$ | $33.84_{0.91}$ | $36.48_{0.54}$ | $37.38_{0.73}$ | $36.71_{1.23}$ |
| N2V (transductive) | $25.50_{0.76}$ | $25.56_{0.95}$ | $25.41_{0.74}$ | $24.26_{1.18}$ | $24.55_{1.70}$ |
| **Amazon-ratings** | | | | | |
| N2V (inductive) | $37.47_{0.47}$ | $40.69_{0.63}$ | $44.71_{0.85}$ | $46.98_{0.68}$ | $49.47_{1.07}$ |
| Feature Propagation | $38.97_{0.76}$ | $41.68_{0.76}$ | $\mathbf{45.72}_{0.75}$ | $48.03_{0.67}$ | $50.04_{1.42}$ |
| **iN2V** (own) | $\mathbf{40.02}_{0.79}$ | $\mathbf{42.01}_{0.48}$ | $45.48_{0.57}$ | $\mathbf{50.39}_{0.48}$ | $\mathbf{50.38}_{2.22}$ |
| Original features | $39.20_{1.06}$ | $41.66_{0.70}$ | $48.07_{0.58}$ | $53.09_{0.69}$ | $57.34_{0.97}$ |
| N2V (transductive) | $41.82_{0.66}$ | $43.69_{0.52}$ | $46.31_{0.78}$ | $48.40_{0.55}$ | $49.80_{0.94}$ |
| **Roman-empire** | | | | | |
| N2V (inductive) | $\mathbf{13.96}_{0.35}$ | $\mathbf{14.10}_{0.32}$ | $\mathbf{15.77}_{0.76}$ | $15.98_{1.55}$ | $16.81_{2.94}$ |
| Feature Propagation | $13.23_{1.44}$ | $13.89_{0.44}$ | $15.49_{0.67}$ | $\mathbf{16.13}_{1.04}$ | $\mathbf{21.97}_{1.25}$ |
| **iN2V** (own) | $13.79_{0.39}$ | $13.86_{0.29}$ | $14.48_{0.63}$ | $15.33_{1.12}$ | $18.55_{1.23}$ |
| Original features | $66.09_{0.75}$ | $70.28_{0.63}$ | $74.41_{0.47}$ | $78.56_{0.50}$ | $82.90_{1.09}$ |
| N2V (transductive) | $13.82_{0.22}$ | $13.86_{0.37}$ | $15.35_{1.53}$ | $21.46_{0.46}$ | $27.46_{1.36}$ |

*Table 7.* Comparison of N2V vs different iN2V setups, MLP accuracy.

| Dataset | Percentage of training data | | | | |
|---|---|---|---|---|---|
| | 10% | 20% | 40% | 60% | 80% |
| **Cora** | | | | | |
| N2V (inductive) | $30.04_{1.06}$ | $30.07_{1.15}$ | $30.12_{1.05}$ | $29.67_{1.04}$ | $30.70_{2.14}$ |
| frozen ($\lambda=1$) | $74.64_{1.75}$ | $76.71_{1.60}$ | $79.57_{1.24}$ | $81.75_{1.63}$ | $83.51_{1.79}$ |
| post-hoc | $\mathbf{79.84}_{1.27}$ | $80.28_{1.80}$ | $83.93_{1.16}$ | $84.00_{1.03}$ | $\mathbf{85.57}_{2.46}$ |
| p-h w losses | $79.38_{1.42}$ | $81.07_{1.58}$ | $\mathbf{83.99}_{1.06}$ | $83.75_{1.51}$ | $85.24_{2.33}$ |
| p-h w sampling | $79.57_{1.34}$ | $\mathbf{81.70}_{1.19}$ | $82.71_{0.84}$ | $\mathbf{84.69}_{0.95}$ | $85.57_{2.22}$ |
| **Citeseer** | | | | | |
| N2V (inductive) | $19.81_{1.75}$ | $21.43_{1.00}$ | $21.11_{1.27}$ | $21.10_{1.46}$ | $19.82_{3.05}$ |
| frozen ($\lambda=1$) | $54.19_{2.79}$ | $59.75_{1.63}$ | $65.06_{1.69}$ | $68.95_{1.27}$ | $71.59_{1.90}$ |
| post-hoc | $55.91_{2.26}$ | $61.51_{1.37}$ | $66.47_{1.91}$ | $70.62_{1.37}$ | $72.55_{1.77}$ |
| p-h w losses | $\mathbf{58.02}_{1.29}$ | $61.67_{1.69}$ | $66.56_{1.90}$ | $\mathbf{70.69}_{1.30}$ | $73.09_{2.32}$ |
| p-h w sampling | $57.88_{1.35}$ | $61.40_{1.21}$ | $\mathbf{66.62}_{1.78}$ | $70.21_{1.59}$ | $\mathbf{73.27}_{1.73}$ |
| **Pubmed** | | | | | |
| N2V (inductive) | $39.45_{0.69}$ | $39.69_{0.77}$ | $40.15_{0.42}$ | $40.01_{0.71}$ | $39.30_{1.06}$ |
| frozen ($\lambda=1$) | $76.61_{0.74}$ | $78.43_{0.75}$ | $80.49_{0.68}$ | $81.45_{0.69}$ | $82.08_{0.79}$ |
| post-hoc | $\mathbf{80.47}_{0.35}$ | $\mathbf{81.77}_{0.40}$ | $82.69_{0.45}$ | $83.42_{0.56}$ | $83.30_{0.60}$ |
| p-h w losses | $80.42_{0.44}$ | $81.68_{0.39}$ | $82.74_{0.47}$ | $\mathbf{83.45}_{0.49}$ | $83.26_{0.87}$ |
| p-h w sampling | $80.42_{0.46}$ | $81.57_{0.38}$ | $\mathbf{82.78}_{0.58}$ | $83.19_{0.44}$ | $\mathbf{83.35}_{0.59}$ |
| **Computers** | | | | | |
| N2V (inductive) | $37.39_{0.32}$ | $37.33_{0.28}$ | $37.31_{0.66}$ | $37.51_{0.75}$ | $37.72_{0.87}$ |
| frozen ($\lambda=1$) | $82.25_{0.73}$ | $84.66_{0.50}$ | $87.12_{0.53}$ | $88.13_{0.70}$ | $88.81_{0.97}$ |
| post-hoc | $88.06_{0.46}$ | $89.87_{0.39}$ | $90.95_{0.40}$ | $90.90_{0.56}$ | $91.32_{0.53}$ |
| p-h w losses | $88.19_{0.42}$ | $\mathbf{90.04}_{0.51}$ | $\mathbf{91.10}_{0.47}$ | $\mathbf{91.55}_{0.38}$ | $91.35_{0.60}$ |
| p-h w sampling | $\mathbf{88.70}_{0.47}$ | $89.69_{0.50}$ | $90.75_{0.57}$ | $91.51_{0.48}$ | $\mathbf{91.55}_{0.52}$ |
| **Photo** | | | | | |
| N2V (inductive) | $25.51_{0.46}$ | $24.86_{1.30}$ | $23.66_{1.99}$ | $22.96_{1.95}$ | $23.15_{2.24}$ |
| frozen ($\lambda=1$) | $86.08_{0.97}$ | $87.72_{0.68}$ | $90.01_{0.78}$ | $90.95_{0.56}$ | $91.56_{0.88}$ |
| post-hoc | $90.42_{0.57}$ | $91.71_{0.36}$ | $92.47_{0.50}$ | $\mathbf{93.08}_{0.73}$ | $93.07_{0.68}$ |
| p-h w losses | $\mathbf{90.70}_{0.50}$ | $91.74_{0.36}$ | $92.60_{0.55}$ | $92.86_{0.72}$ | $93.12_{0.87}$ |
| p-h w sampling | $90.38_{0.54}$ | $\mathbf{91.90}_{0.44}$ | $92.47_{0.64}$ | $92.70_{0.79}$ | $93.11_{1.02}$ |
| **WikiCS** | | | | | |
| N2V (inductive) | $22.99_{0.41}$ | $22.96_{0.49}$ | $23.05_{0.51}$ | $23.12_{0.76}$ | $22.82_{1.26}$ |
| frozen ($\lambda=1$) | $70.64_{2.01}$ | $71.89_{1.30}$ | $75.48_{0.98}$ | $77.27_{0.54}$ | $78.62_{1.65}$ |
| post-hoc | $78.92_{0.69}$ | $80.16_{0.78}$ | $81.36_{0.63}$ | $82.03_{0.57}$ | $82.56_{0.79}$ |
| p-h w losses | $78.98_{0.64}$ | $80.22_{0.81}$ | $81.47_{0.52}$ | $82.12_{0.68}$ | $\mathbf{82.76}_{0.83}$ |
| p-h w sampling | $\mathbf{79.21}_{0.59}$ | $\mathbf{80.37}_{0.80}$ | $81.48_{0.66}$ | $82.18_{0.77}$ | $82.60_{0.83}$ |
| **Actor** | | | | | |
| N2V (inductive) | $25.41_{1.39}$ | $25.55_{1.16}$ | $\mathbf{25.69}_{1.08}$ | $\mathbf{26.05}_{0.51}$ | $\mathbf{26.57}_{0.79}$ |
| frozen ($\lambda=1$) | $\mathbf{25.78}_{0.76}$ | $25.45_{0.87}$ | $25.00_{1.35}$ | $24.62_{0.94}$ | $25.03_{0.99}$ |
| post-hoc | $25.43_{1.35}$ | $25.78_{0.78}$ | $24.84_{0.97}$ | $24.93_{1.07}$ | $25.83_{1.50}$ |
| p-h w losses | $25.42_{1.11}$ | $25.76_{0.92}$ | $24.75_{0.87}$ | $25.45_{0.40}$ | $25.51_{2.00}$ |
| p-h w sampling | $24.96_{1.25}$ | $\mathbf{25.82}_{0.83}$ | $25.62_{0.80}$ | $25.14_{0.76}$ | $24.43_{1.97}$ |
| **Amazon-ratings** | | | | | |
| N2V (inductive) | $36.91_{0.30}$ | $36.93_{0.35}$ | $36.74_{0.37}$ | $36.78_{0.52}$ | $36.54_{0.89}$ |
| frozen ($\lambda=1$) | $39.04_{0.65}$ | $41.31_{1.01}$ | $44.44_{1.62}$ | $45.77_{0.44}$ | $46.74_{0.81}$ |
| post-hoc | $41.33_{0.66}$ | $\mathbf{43.89}_{0.50}$ | $46.52_{0.77}$ | $49.44_{0.78}$ | $49.78_{1.70}$ |
| p-h w losses | $\mathbf{41.43}_{0.79}$ | $43.69_{0.57}$ | $\mathbf{46.68}_{0.94}$ | $\mathbf{51.66}_{0.31}$ | $\mathbf{51.93}_{0.78}$ |
| p-h w sampling | $41.41_{0.87}$ | $43.75_{0.54}$ | $46.58_{1.02}$ | $49.28_{0.64}$ | $49.06_{0.67}$ |
| **Roman-empire** | | | | | |
| N2V (inductive) | $13.34_{1.25}$ | $13.55_{0.97}$ | $14.00_{0.28}$ | $13.99_{0.40}$ | $13.27_{1.67}$ |
| frozen ($\lambda=1$) | $13.89_{0.26}$ | $\mathbf{13.90}_{0.27}$ | $13.90_{0.19}$ | $13.74_{0.62}$ | $13.64_{0.76}$ |
| post-hoc | $\mathbf{13.90}_{0.26}$ | $13.85_{0.30}$ | $\mathbf{14.01}_{0.24}$ | $\mathbf{14.23}_{0.69}$ | $\mathbf{14.27}_{0.94}$ |
| p-h w losses | $13.81_{0.36}$ | $13.68_{0.33}$ | $13.79_{0.24}$ | $13.97_{0.40}$ | $14.16_{0.80}$ |
| p-h w sampling | $13.85_{0.29}$ | $13.84_{0.28}$ | $13.88_{0.18}$ | $13.96_{0.55}$ | $13.61_{0.81}$ |

*Table 8.* Comparison of N2V vs different iN2V setups, Graph-SAGE accuracy.

| Dataset | Percentage of training data | | | | |
|---|---|---|---|---|---|
| | 10% | 20% | 40% | 60% | 80% |
| **Cora** | | | | | |
| N2V (inductive) | $42.18_{3.52}$ | $59.91_{4.15}$ | $75.07_{1.92}$ | $81.31_{1.56}$ | $84.50_{1.44}$ |
| frozen ($\lambda=1$) | $74.76_{2.08}$ | $76.86_{1.66}$ | $80.75_{1.17}$ | $82.45_{2.14}$ | $84.13_{1.99}$ |
| post-hoc | $\mathbf{79.20}_{1.80}$ | $79.44_{1.55}$ | $\mathbf{83.45}_{1.08}$ | $83.73_{1.13}$ | $84.46_{2.08}$ |
| p-h w losses | $78.88_{1.45}$ | $80.50_{1.26}$ | $83.30_{1.09}$ | $83.39_{1.74}$ | $\mathbf{84.69}_{1.65}$ |
| p-h w sampling | $79.11_{1.77}$ | $\mathbf{80.94}_{1.58}$ | $82.02_{1.34}$ | $83.43_{1.50}$ | $84.02_{1.88}$ |
| **Citeseer** | | | | | |
| N2V (inductive) | $34.17_{3.43}$ | $42.76_{2.15}$ | $56.12_{3.24}$ | $63.61_{2.57}$ | $68.89_{2.61}$ |
| frozen ($\lambda=1$) | $56.22_{1.72}$ | $59.00_{2.16}$ | $\mathbf{63.63}_{1.47}$ | $\mathbf{67.73}_{1.63}$ | $68.92_{2.26}$ |
| post-hoc | $57.25_{1.27}$ | $60.59_{1.52}$ | $62.88_{1.80}$ | $66.81_{1.42}$ | $\mathbf{69.13}_{2.33}$ |
| p-h w losses | $57.25_{1.55}$ | $\mathbf{60.78}_{1.78}$ | $62.76_{1.90}$ | $67.23_{2.14}$ | $68.92_{2.07}$ |
| p-h w sampling | $\mathbf{57.88}_{0.91}$ | $57.92_{3.47}$ | $63.02_{1.88}$ | $67.23_{1.81}$ | $68.83_{2.37}$ |
| **Pubmed** | | | | | |
| N2V (inductive) | $66.02_{5.57}$ | $74.73_{2.36}$ | $79.84_{1.78}$ | $81.85_{0.61}$ | $82.61_{0.52}$ |
| frozen ($\lambda=1$) | $75.99_{0.52}$ | $78.37_{1.02}$ | $81.31_{0.55}$ | $82.30_{0.81}$ | $82.88_{0.72}$ |
| post-hoc | $\mathbf{79.93}_{0.50}$ | $\mathbf{80.80}_{0.46}$ | $81.80_{0.49}$ | $82.74_{0.29}$ | $82.90_{0.73}$ |
| p-h w losses | $79.75_{0.61}$ | $80.75_{0.61}$ | $81.91_{0.54}$ | $82.71_{0.34}$ | $82.59_{0.63}$ |
| p-h w sampling | $79.68_{0.55}$ | $80.59_{0.54}$ | $\mathbf{82.14}_{0.43}$ | $\mathbf{82.81}_{0.46}$ | $\mathbf{82.94}_{0.60}$ |
| **Computers** | | | | | |
| N2V (inductive) | $77.64_{2.81}$ | $84.44_{0.84}$ | $87.18_{0.77}$ | $89.21_{0.42}$ | $89.35_{0.70}$ |
| frozen ($\lambda=1$) | $85.40_{0.54}$ | $87.45_{0.42}$ | $89.05_{0.56}$ | $90.09_{0.44}$ | $90.24_{0.59}$ |
| post-hoc | $87.94_{0.43}$ | $89.62_{0.43}$ | $90.66_{0.38}$ | $90.71_{0.52}$ | $91.07_{0.64}$ |
| p-h w losses | $87.84_{0.32}$ | $\mathbf{89.67}_{0.40}$ | $\mathbf{90.84}_{0.37}$ | $91.08_{0.36}$ | $91.24_{0.52}$ |
| p-h w sampling | $\mathbf{88.36}_{0.58}$ | $89.26_{0.50}$ | $90.53_{0.62}$ | $\mathbf{91.16}_{0.47}$ | $\mathbf{91.38}_{0.51}$ |
| **Photo** | | | | | |
| N2V (inductive) | $85.48_{1.28}$ | $87.73_{1.40}$ | $90.98_{0.71}$ | $91.92_{0.63}$ | $92.21_{0.96}$ |
| frozen ($\lambda=1$) | $88.64_{0.95}$ | $90.31_{0.50}$ | $91.78_{0.47}$ | $92.48_{0.59}$ | $92.76_{0.78}$ |
| post-hoc | $90.49_{0.67}$ | $91.62_{0.50}$ | $92.40_{0.53}$ | $\mathbf{92.93}_{0.74}$ | $\mathbf{93.08}_{0.77}$ |
| p-h w losses | $\mathbf{90.51}_{0.72}$ | $91.67_{0.30}$ | $92.37_{0.46}$ | $92.74_{0.61}$ | $92.95_{1.00}$ |
| p-h w sampling | $90.42_{0.54}$ | $\mathbf{91.70}_{0.44}$ | $\mathbf{92.50}_{0.50}$ | $92.82_{0.63}$ | $92.88_{0.81}$ |
| **WikiCS** | | | | | |
| N2V (inductive) | $67.78_{2.68}$ | $74.22_{1.65}$ | $78.21_{0.69}$ | $80.40_{0.45}$ | $81.62_{0.86}$ |
| frozen ($\lambda=1$) | $75.62_{0.63}$ | $77.96_{0.56}$ | $79.83_{0.60}$ | $80.85_{0.63}$ | $81.55_{0.58}$ |
| post-hoc | $78.80_{0.74}$ | $80.07_{0.46}$ | $81.01_{0.63}$ | $\mathbf{81.84}_{0.39}$ | $\mathbf{82.55}_{0.72}$ |
| p-h w losses | $78.72_{0.55}$ | $80.06_{0.71}$ | $81.20_{0.48}$ | $81.73_{0.76}$ | $82.43_{0.69}$ |
| p-h w sampling | $\mathbf{78.91}_{0.61}$ | $\mathbf{80.19}_{0.70}$ | $\mathbf{81.28}_{0.61}$ | $81.82_{0.54}$ | $82.37_{1.01}$ |
| **Actor** | | | | | |
| N2V (inductive) | $25.14_{1.09}$ | $25.56_{0.90}$ | $25.54_{1.15}$ | $\mathbf{25.42}_{1.03}$ | $25.68_{1.42}$ |
| frozen ($\lambda=1$) | $25.18_{0.97}$ | $22.94_{0.83}$ | $25.40_{0.99}$ | $24.93_{0.89}$ | $24.97_{1.10}$ |
| post-hoc | $\mathbf{25.40}_{1.08}$ | $\mathbf{25.82}_{0.99}$ | $25.34_{0.85}$ | $24.89_{0.75}$ | $25.18_{1.18}$ |
| p-h w losses | $25.06_{1.06}$ | $24.95_{0.78}$ | $\mathbf{25.85}_{0.79}$ | $24.41_{1.27}$ | $\mathbf{25.79}_{2.27}$ |
| p-h w sampling | $25.39_{1.14}$ | $25.50_{0.70}$ | $25.71_{0.93}$ | $25.16_{0.92}$ | $24.17_{1.69}$ |
| **Amazon-ratings** | | | | | |
| N2V (inductive) | $37.47_{0.47}$ | $40.69_{0.63}$ | $44.71_{0.85}$ | $46.98_{0.68}$ | $49.47_{1.07}$ |
| frozen ($\lambda=1$) | $36.70_{1.51}$ | $41.76_{0.91}$ | $\mathbf{45.48}_{0.57}$ | $48.10_{0.58}$ | $50.19_{1.10}$ |
| post-hoc | $38.68_{1.00}$ | $\mathbf{42.12}_{0.66}$ | $45.20_{0.70}$ | $47.93_{0.29}$ | $48.40_{1.23}$ |
| p-h w losses | $40.02_{0.79}$ | $42.01_{0.48}$ | $45.46_{0.82}$ | $\mathbf{50.39}_{0.48}$ | $\mathbf{50.38}_{2.22}$ |
| p-h w sampling | $\mathbf{40.10}_{0.68}$ | $41.92_{0.85}$ | $45.31_{0.74}$ | $47.44_{0.46}$ | $48.40_{1.30}$ |
| **Roman-empire** | | | | | |
| N2V (inductive) | $\mathbf{13.96}_{0.35}$ | $\mathbf{14.10}_{0.32}$ | $\mathbf{15.77}_{0.76}$ | $\mathbf{15.98}_{1.55}$ | $16.81_{2.94}$ |
| frozen ($\lambda=1$) | $13.64_{0.85}$ | $13.88_{0.29}$ | $14.48_{0.63}$ | $14.82_{1.04}$ | $17.59_{4.35}$ |
| post-hoc | $13.31_{1.26}$ | $13.86_{0.29}$ | $13.86_{0.17}$ | $14.89_{0.64}$ | $14.89_{1.70}$ |
| p-h w losses | $13.17_{1.43}$ | $13.84_{0.27}$ | $13.90_{0.26}$ | $15.05_{1.07}$ | $16.93_{1.31}$ |
| p-h w sampling | $13.79_{0.39}$ | $13.59_{0.75}$ | $14.02_{0.41}$ | $15.33_{1.12}$ | $\mathbf{18.55}_{1.23}$ |

*Table 9.* Comparison of best iN2V variant vs baselines concatenated with the original graph features; MLP accuracy. Gray numbers are not directly comparable as they use additional information (transductive setup).

| Dataset | Percentage of training data | | | | |
| --- | --- | --- | --- | --- | --- |
| | 10% | 20% | 40% | 60% | 80% |
| **Cora** | | | | | |
| N2V (inductive) | $66.48_{1.68}$ | $70.54_{1.20}$ | $75.47_{1.91}$ | $76.09_{2.08}$ | $76.27_{1.68}$ |
| Feature Propagation | $\mathbf{82.24}_{1.06}$ | $83.49_{1.26}$ | $83.83_{1.49}$ | $85.59_{1.24}$ | $85.54_{1.48}$ |
| **iN2V** (own) | $82.01_{1.38}$ | $\mathbf{83.97}_{1.35}$ | $\mathbf{85.83}_{1.45}$ | $85.81_{1.77}$ | $\mathbf{86.49}_{1.62}$ |
| Original features | $66.98_{1.70}$ | $71.34_{1.79}$ | $75.94_{1.38}$ | $78.78_{1.11}$ | $79.23_{1.95}$ |
| N2V (transductive) | $81.94_{1.39}$ | $83.52_{1.13}$ | $85.99_{1.10}$ | $86.99_{1.44}$ | $86.13_{1.69}$ |
| **Citeseer** | | | | | |
| N2V (inductive) | $66.06_{1.31}$ | $70.04_{1.29}$ | $72.33_{1.07}$ | $73.58_{1.14}$ | $74.68_{2.12}$ |
| Feature Propagation | $\mathbf{68.48}_{1.20}$ | $\mathbf{72.13}_{1.08}$ | $\mathbf{74.37}_{1.26}$ | $75.29_{0.97}$ | $76.34_{1.66}$ |
| **iN2V** (own) | $66.98_{1.14}$ | $71.35_{0.83}$ | $74.13_{1.23}$ | $\mathbf{75.85}_{0.87}$ | $\mathbf{77.09}_{2.79}$ |
| Original features | $66.60_{1.56}$ | $70.53_{1.04}$ | $73.23_{1.17}$ | $74.56_{1.04}$ | $75.20_{2.10}$ |
| N2V (transductive) | $69.40_{0.88}$ | $72.58_{1.28}$ | $74.78_{0.93}$ | $75.94_{1.03}$ | $77.27_{2.34}$ |
| **Pubmed** | | | | | |
| N2V (inductive) | $83.94_{0.42}$ | $85.55_{0.42}$ | $86.73_{0.46}$ | $87.21_{0.58}$ | $87.40_{0.92}$ |
| Feature Propagation | $\mathbf{85.59}_{0.26}$ | $87.06_{0.41}$ | $\mathbf{88.71}_{0.49}$ | $89.33_{0.39}$ | $\mathbf{89.95}_{0.60}$ |
| **iN2V** (own) | $85.38_{0.39}$ | $\mathbf{87.07}_{0.39}$ | $88.69_{0.60}$ | $\mathbf{89.46}_{0.51}$ | $89.90_{0.51}$ |
| Original features | $84.11_{0.21}$ | $86.05_{0.40}$ | $87.72_{0.40}$ | $88.63_{0.60}$ | $89.10_{0.63}$ |
| N2V (transductive) | $86.16_{0.28}$ | $87.95_{0.47}$ | $88.73_{0.37}$ | $89.42_{0.47}$ | $89.89_{0.59}$ |
| **Computers** | | | | | |
| N2V (inductive) | $80.41_{0.60}$ | $82.98_{0.65}$ | $84.08_{0.76}$ | $84.34_{0.97}$ | $84.72_{0.87}$ |
| Feature Propagation | $85.39_{0.67}$ | $87.03_{0.33}$ | $89.14_{0.40}$ | $89.91_{0.70}$ | $90.15_{0.78}$ |
| **iN2V** (own) | $\mathbf{88.63}_{0.22}$ | $\mathbf{89.75}_{0.29}$ | $\mathbf{90.67}_{0.24}$ | $\mathbf{90.99}_{0.52}$ | $\mathbf{90.95}_{0.59}$ |
| Original features | $82.15_{0.63}$ | $84.08_{0.65}$ | $85.57_{0.53}$ | $86.00_{0.77}$ | $86.56_{0.58}$ |
| N2V (transductive) | $89.36_{0.39}$ | $90.61_{0.23}$ | $91.19_{0.39}$ | $91.47_{0.40}$ | $91.30_{0.47}$ |
| **Photo** | | | | | |
| N2V (inductive) | $85.73_{2.20}$ | $86.22_{1.22}$ | $90.71_{0.59}$ | $90.90_{0.65}$ | $91.76_{0.70}$ |
| Feature Propagation | $92.75_{0.44}$ | $93.93_{0.28}$ | $94.73_{0.40}$ | $94.75_{0.51}$ | $95.31_{0.62}$ |
| **iN2V** (own) | $\mathbf{93.99}_{0.36}$ | $\mathbf{94.55}_{0.29}$ | $\mathbf{95.33}_{0.32}$ | $\mathbf{95.42}_{0.45}$ | $\mathbf{95.70}_{0.74}$ |
| Original features | $89.32_{0.59}$ | $90.85_{0.55}$ | $91.96_{0.55}$ | $92.41_{0.60}$ | $92.92_{0.87}$ |
| N2V (transductive) | $94.07_{0.46}$ | $94.93_{0.29}$ | $95.36_{0.28}$ | $95.73_{0.51}$ | $95.63_{0.53}$ |
| **WikiCS** | | | | | |
| N2V (inductive) | $73.16_{1.17}$ | $75.60_{0.78}$ | $76.79_{0.51}$ | $77.30_{0.73}$ | $76.71_{0.97}$ |
| Feature Propagation | $78.14_{1.23}$ | $80.58_{0.52}$ | $82.91_{0.53}$ | $83.98_{1.01}$ | $84.78_{1.15}$ |
| **iN2V** (own) | $\mathbf{81.78}_{0.50}$ | $\mathbf{83.47}_{0.47}$ | $\mathbf{84.77}_{0.55}$ | $\mathbf{85.41}_{0.76}$ | $\mathbf{86.26}_{0.87}$ |
| Original features | $76.85_{0.64}$ | $78.76_{0.66}$ | $80.43_{0.64}$ | $81.56_{0.78}$ | $82.38_{1.06}$ |
| N2V (transductive) | $82.26_{0.55}$ | $83.73_{0.65}$ | $84.70_{0.41}$ | $85.44_{0.49}$ | $85.50_{0.77}$ |
| **Actor** | | | | | |
| N2V (inductive) | $33.99_{0.93}$ | $35.54_{1.01}$ | $36.64_{0.85}$ | $38.22_{0.70}$ | $\mathbf{38.08}_{1.62}$ |
| Feature Propagation | $33.91_{0.72}$ | $35.94_{0.44}$ | $36.63_{0.76}$ | $38.22_{0.72}$ | $37.43_{1.16}$ |
| **iN2V** (own) | $\mathbf{35.15}_{1.06}$ | $35.92_{0.56}$ | $37.66_{0.96}$ | $37.72_{0.89}$ | $38.07_{1.12}$ |
| Original features | $35.13_{0.51}$ | $\mathbf{36.20}_{0.67}$ | $\mathbf{37.79}_{0.62}$ | $38.75_{0.86}$ | $37.87_{1.53}$ |
| N2V (transductive) | $32.35_{0.94}$ | $34.75_{0.83}$ | $36.35_{1.01}$ | $37.30_{1.52}$ | $37.83_{1.43}$ |
| **Amazon-ratings** | | | | | |
| N2V (inductive) | $37.36_{0.39}$ | $37.93_{0.59}$ | $38.30_{0.50}$ | $38.51_{0.69}$ | $38.57_{1.67}$ |
| Feature Propagation | $39.86_{0.72}$ | $42.92_{1.07}$ | $47.89_{0.95}$ | $50.90_{0.70}$ | $53.07_{1.18}$ |
| **iN2V** (own) | $\mathbf{41.24}_{0.74}$ | $\mathbf{45.64}_{0.67}$ | $\mathbf{51.91}_{0.40}$ | $\mathbf{55.96}_{0.81}$ | $\mathbf{58.44}_{0.67}$ |
| Original features | $37.80_{0.51}$ | $41.77_{0.61}$ | $47.20_{0.56}$ | $50.82_{0.77}$ | $54.00_{0.69}$ |
| N2V (transductive) | $41.73_{0.66}$ | $45.42_{0.53}$ | $50.92_{0.58}$ | $53.93_{0.52}$ | $56.78_{1.45}$ |
| **Roman-empire** | | | | | |
| N2V (inductive) | $62.37_{0.51}$ | $64.35_{0.38}$ | $65.67_{0.18}$ | $66.48_{0.70}$ | $66.54_{1.07}$ |
| Feature Propagation | $62.02_{0.41}$ | $64.16_{0.39}$ | $65.81_{0.33}$ | $65.96_{0.51}$ | $66.34_{1.11}$ |
| **iN2V** (own) | $60.57_{0.39}$ | $63.57_{0.30}$ | $65.36_{0.35}$ | $65.83_{0.80}$ | $65.99_{1.12}$ |
| Original features | $\mathbf{63.46}_{0.43}$ | $\mathbf{65.05}_{0.31}$ | $\mathbf{66.47}_{0.41}$ | $\mathbf{66.83}_{0.71}$ | $\mathbf{66.55}_{1.02}$ |
| N2V (transductive) | $60.28_{0.27}$ | $63.54_{0.28}$ | $65.09_{0.37}$ | $65.99_{0.62}$ | $66.32_{1.23}$ |

*Table 10.* Comparison of best iN2V variant vs baselines concatenated with the original graph features; GraphSAGE accuracy. Gray numbers are not directly comparable as they use additional information (transductive setup).

| Dataset | Percentage of training data | | | | |
| --- | --- | --- | --- | --- | --- |
| | 10% | 20% | 40% | 60% | 80% |
| **Cora** | | | | | |
| N2V (inductive) | $73.97_{3.35}$ | $81.92_{1.84}$ | $85.97_{1.67}$ | $86.97_{0.96}$ | $87.38_{0.79}$ |
| Feature Propagation | $80.92_{1.29}$ | $\mathbf{83.49}_{1.54}$ | $85.33_{1.44}$ | $\mathbf{87.32}_{1.07}$ | $\mathbf{87.75}_{1.12}$ |
| **iN2V** (own) | $\mathbf{81.01}_{1.42}$ | $83.28_{1.41}$ | $85.92_{1.17}$ | $86.77_{1.42}$ | $87.60_{1.81}$ |
| Original features | $75.27_{2.63}$ | $83.37_{1.17}$ | $\mathbf{86.23}_{1.77}$ | $86.99_{0.86}$ | $87.05_{1.20}$ |
| N2V (transductive) | $82.43_{0.77}$ | $84.45_{1.23}$ | $86.50_{1.04}$ | $86.85_{1.16}$ | $88.30_{1.47}$ |
| **Citeseer** | | | | | |
| N2V (inductive) | $68.52_{2.47}$ | $72.49_{1.14}$ | $74.53_{1.28}$ | $75.85_{1.40}$ | $76.76_{1.85}$ |
| Feature Propagation | $68.00_{1.58}$ | $71.72_{1.15}$ | $74.77_{0.97}$ | $\mathbf{76.39}_{0.99}$ | $76.46_{1.73}$ |
| **iN2V** (own) | $67.11_{1.47}$ | $71.14_{0.60}$ | $73.99_{1.47}$ | $75.77_{1.29}$ | $75.50_{1.63}$ |
| Original features | $\mathbf{69.85}_{1.42}$ | $\mathbf{72.86}_{0.98}$ | $\mathbf{74.93}_{1.56}$ | $76.39_{1.05}$ | $\mathbf{76.82}_{1.77}$ |
| N2V (transductive) | $71.08_{1.04}$ | $73.21_{1.12}$ | $74.76_{0.81}$ | $76.14_{1.43}$ | $76.82_{1.77}$ |
| **Pubmed** | | | | | |
| N2V (inductive) | $84.33_{0.46}$ | $86.06_{0.59}$ | $87.32_{0.38}$ | $87.32_{0.39}$ | $88.01_{0.78}$ |
| Feature Propagation | $82.99_{0.64}$ | $85.46_{0.46}$ | $86.74_{0.55}$ | $87.97_{0.40}$ | $88.96_{0.63}$ |
| **iN2V** (own) | $83.02_{0.72}$ | $85.09_{0.37}$ | $87.51_{0.51}$ | $88.59_{0.35}$ | $89.39_{0.72}$ |
| Original features | $\mathbf{85.95}_{0.47}$ | $\mathbf{86.99}_{0.28}$ | $\mathbf{88.32}_{0.48}$ | $\mathbf{89.25}_{0.44}$ | $\mathbf{89.85}_{0.56}$ |
| N2V (transductive) | $85.36_{0.45}$ | $87.28_{0.46}$ | $87.12_{0.44}$ | $88.42_{0.36}$ | $89.02_{0.57}$ |
| **Computers** | | | | | |
| N2V (inductive) | $86.64_{1.11}$ | $89.05_{0.63}$ | $90.72_{0.36}$ | $91.33_{0.56}$ | $91.70_{0.55}$ |
| Feature Propagation | $87.77_{0.57}$ | $89.93_{0.39}$ | $91.30_{0.34}$ | $91.85_{0.52}$ | $\mathbf{92.13}_{0.45}$ |
| **iN2V** (own) | $\mathbf{88.90}_{0.36}$ | $\mathbf{90.50}_{0.22}$ | $\mathbf{91.43}_{0.36}$ | $\mathbf{91.91}_{0.49}$ | $91.72_{0.23}$ |
| Original features | $87.52_{0.48}$ | $89.76_{0.40}$ | $91.12_{0.20}$ | $91.49_{0.58}$ | $91.50_{0.48}$ |
| N2V (transductive) | $90.48_{0.37}$ | $91.59_{0.36}$ | $92.09_{0.36}$ | $92.35_{0.36}$ | $92.20_{0.37}$ |
| **Photo** | | | | | |
| N2V (inductive) | $91.97_{1.73}$ | $93.87_{0.70}$ | $94.94_{0.44}$ | $95.37_{0.60}$ | $95.56_{0.81}$ |
| Feature Propagation | $93.25_{0.54}$ | $94.40_{0.41}$ | $95.11_{0.41}$ | $95.54_{0.54}$ | $\mathbf{95.73}_{0.77}$ |
| **iN2V** (own) | $\mathbf{93.87}_{0.52}$ | $\mathbf{94.68}_{0.33}$ | $95.26_{0.28}$ | $\mathbf{95.59}_{0.42}$ | $95.63_{0.74}$ |
| Original features | $93.74_{0.42}$ | $94.59_{0.37}$ | $\mathbf{95.27}_{0.38}$ | $95.52_{0.52}$ | $95.59_{0.77}$ |
| N2V (transductive) | $94.49_{0.46}$ | $94.97_{0.28}$ | $95.50_{0.22}$ | $95.69_{0.32}$ | $95.79_{1.00}$ |
| **WikiCS** | | | | | |
| N2V (inductive) | $76.50_{1.34}$ | $79.99_{0.65}$ | $82.23_{0.75}$ | $83.76_{0.63}$ | $84.79_{0.87}$ |
| Feature Propagation | $79.58_{0.88}$ | $82.09_{0.86}$ | $84.10_{0.63}$ | $84.95_{0.77}$ | $85.77_{0.76}$ |
| **iN2V** (own) | $\mathbf{81.50}_{0.64}$ | $\mathbf{83.38}_{0.58}$ | $\mathbf{84.62}_{0.47}$ | $\mathbf{85.25}_{0.47}$ | $85.73_{0.70}$ |
| Original features | $80.75_{0.64}$ | $82.56_{0.81}$ | $84.28_{0.55}$ | $85.21_{0.63}$ | $\mathbf{85.88}_{0.70}$ |
| N2V (transductive) | $81.91_{0.52}$ | $83.37_{0.67}$ | $84.66_{0.64}$ | $85.56_{0.61}$ | $86.10_{0.70}$ |
| **Actor** | | | | | |
| N2V (inductive) | $32.04_{0.94}$ | $33.44_{0.73}$ | $35.14_{0.93}$ | $36.13_{1.52}$ | $36.05_{2.46}$ |
| Feature Propagation | $\mathbf{32.51}_{0.51}$ | $33.08_{0.73}$ | $34.89_{1.40}$ | $36.19_{0.88}$ | $36.25_{1.13}$ |
| **iN2V** (own) | $32.12_{0.73}$ | $33.53_{0.63}$ | $35.38_{1.24}$ | $36.03_{0.64}$ | $36.66_{2.24}$ |
| Original features | $31.77_{0.71}$ | $\mathbf{33.84}_{0.91}$ | $\mathbf{36.48}_{0.54}$ | $\mathbf{37.38}_{0.73}$ | $\mathbf{36.71}_{1.23}$ |
| N2V (transductive) | $30.82_{1.12}$ | $32.52_{0.71}$ | $34.46_{0.75}$ | $35.45_{1.18}$ | $36.54_{2.27}$ |
| **Amazon-ratings** | | | | | |
| N2V (inductive) | $39.14_{0.57}$ | $41.43_{0.65}$ | $44.99_{1.04}$ | $48.97_{1.16}$ | $50.98_{1.35}$ |
| Feature Propagation | $39.50_{0.98}$ | $42.93_{0.82}$ | $48.31_{0.67}$ | $51.28_{0.63}$ | $53.90_{1.14}$ |
| **iN2V** (own) | $\mathbf{40.29}_{0.67}$ | $\mathbf{43.72}_{0.70}$ | $\mathbf{50.99}_{0.84}$ | $\mathbf{55.42}_{0.83}$ | $\mathbf{58.70}_{0.84}$ |
| Original features | $39.20_{1.06}$ | $41.66_{0.70}$ | $48.07_{0.58}$ | $53.09_{0.69}$ | $57.34_{0.97}$ |
| N2V (transductive) | $42.38_{0.53}$ | $46.31_{0.72}$ | $51.60_{0.76}$ | $54.84_{1.11}$ | $57.05_{1.31}$ |
| **Roman-empire** | | | | | |
| N2V (inductive) | $64.71_{0.66}$ | $68.52_{0.65}$ | $72.78_{0.31}$ | $76.68_{0.64}$ | $81.16_{1.65}$ |
| Feature Propagation | $62.25_{0.46}$ | $67.03_{0.44}$ | $71.74_{0.25}$ | $76.02_{0.75}$ | $80.31_{0.97}$ |
| **iN2V** (own) | $59.29_{0.87}$ | $66.19_{0.59}$ | $71.50_{0.43}$ | $76.07_{0.49}$ | $80.48_{0.75}$ |
| Original features | $\mathbf{66.09}_{0.75}$ | $\mathbf{70.28}_{0.63}$ | $\mathbf{74.41}_{0.47}$ | $\mathbf{78.56}_{0.50}$ | $\mathbf{82.90}_{1.09}$ |
| N2V (transductive) | $66.95_{0.77}$ | $71.66_{0.47}$ | $78.39_{0.46}$ | $82.41_{0.48}$ | $84.67_{0.95}$ |

*Table 11.* MLP on different iN2V setups concatenated with the original graph features.

| Dataset | 10% | 20% | 40% | 60% | 80% |
|---|---|---|---|---|---|
| Cora | | | | | |
| N2V (inductive) | $66.48_{1.68}$ | $70.54_{1.20}$ | $75.47_{1.91}$ | $76.09_{2.08}$ | $76.27_{1.68}$ |
| frozen ($\lambda=1$) | $76.83_{1.66}$ | $79.70_{1.44}$ | $82.76_{1.87}$ | $85.31_{1.85}$ | $85.06_{2.20}$ |
| post-hoc | $\mathbf{82.01}_{1.38}$ | $82.53_{1.58}$ | $85.81_{1.38}$ | $85.63_{1.73}$ | $\mathbf{87.12}_{2.08}$ |
| p-h w losses | $81.85_{1.42}$ | $83.27_{1.21}$ | $\mathbf{85.83}_{1.45}$ | $\mathbf{86.27}_{1.97}$ | $86.68_{1.66}$ |
| p-h w sampling | $81.72_{1.26}$ | $\mathbf{83.97}_{1.35}$ | $85.37_{1.35}$ | $85.81_{1.77}$ | $86.49_{1.62}$ |
| Citeseer | | | | | |
| N2V (inductive) | $66.06_{1.31}$ | $70.04_{1.29}$ | $72.33_{1.07}$ | $73.58_{1.14}$ | $74.68_{2.12}$ |
| frozen ($\lambda=1$) | $65.59_{1.52}$ | $70.19_{0.97}$ | $73.53_{1.23}$ | $74.35_{0.96}$ | $76.49_{1.76}$ |
| post-hoc | $66.75_{1.01}$ | $71.24_{1.00}$ | $74.13_{1.23}$ | $75.85_{0.87}$ | $76.40_{2.23}$ |
| p-h w losses | $66.98_{1.14}$ | $\mathbf{71.35}_{0.83}$ | $\mathbf{74.37}_{1.05}$ | $75.70_{1.53}$ | $76.49_{2.44}$ |
| p-h w sampling | $\mathbf{67.33}_{1.33}$ | $70.35_{1.11}$ | $74.30_{1.03}$ | $\mathbf{75.92}_{1.10}$ | $\mathbf{77.09}_{2.79}$ |
| Pubmed | | | | | |
| N2V (inductive) | $83.94_{0.42}$ | $85.55_{0.42}$ | $86.73_{0.46}$ | $87.21_{0.58}$ | $87.40_{0.92}$ |
| frozen ($\lambda=1$) | $83.40_{0.62}$ | $86.17_{0.42}$ | $88.08_{0.52}$ | $88.86_{0.63}$ | $88.97_{0.79}$ |
| post-hoc | $85.34_{0.40}$ | $87.05_{0.45}$ | $88.01_{0.45}$ | $89.44_{0.51}$ | $\mathbf{90.01}_{0.71}$ |
| p-h w losses | $85.37_{0.40}$ | $\mathbf{87.07}_{0.39}$ | $88.15_{0.49}$ | $89.29_{0.44}$ | $89.62_{0.68}$ |
| p-h w sampling | $\mathbf{85.38}_{0.39}$ | $86.85_{0.51}$ | $\mathbf{88.69}_{0.60}$ | $\mathbf{89.46}_{0.51}$ | $89.90_{0.51}$ |
| Computers | | | | | |
| N2V (inductive) | $80.41_{0.60}$ | $82.98_{0.65}$ | $84.08_{0.76}$ | $84.34_{0.97}$ | $84.72_{0.87}$ |
| frozen ($\lambda=1$) | $86.09_{0.53}$ | $87.99_{0.36}$ | $89.47_{0.38}$ | $90.21_{0.43}$ | $90.19_{0.64}$ |
| post-hoc | $88.32_{0.29}$ | $89.57_{0.36}$ | $90.46_{0.44}$ | $90.37_{0.56}$ | $90.64_{0.82}$ |
| p-h w losses | $88.34_{0.16}$ | $89.63_{0.34}$ | $90.40_{0.39}$ | $90.68_{0.53}$ | $90.89_{0.74}$ |
| p-h w sampling | $\mathbf{88.63}_{0.22}$ | $\mathbf{89.75}_{0.22}$ | $\mathbf{90.67}_{0.24}$ | $\mathbf{90.99}_{0.52}$ | $\mathbf{90.95}_{0.59}$ |
| Photo | | | | | |
| N2V (inductive) | $85.73_{2.20}$ | $86.22_{1.22}$ | $90.71_{0.59}$ | $90.90_{0.65}$ | $91.76_{0.70}$ |
| frozen ($\lambda=1$) | $92.13_{0.39}$ | $93.66_{0.64}$ | $94.59_{0.37}$ | $95.10_{0.49}$ | $95.32_{0.74}$ |
| post-hoc | $93.86_{0.39}$ | $94.54_{0.37}$ | $95.24_{0.40}$ | $95.31_{0.45}$ | $95.48_{0.69}$ |
| p-h w losses | $93.85_{0.46}$ | $\mathbf{94.59}_{0.30}$ | $95.21_{0.38}$ | $95.28_{0.38}$ | $95.69_{0.64}$ |
| p-h w sampling | $\mathbf{93.99}_{0.36}$ | $94.55_{0.29}$ | $\mathbf{95.33}_{0.32}$ | $\mathbf{95.42}_{0.45}$ | $\mathbf{95.70}_{0.74}$ |
| WikiCS | | | | | |
| N2V (inductive) | $73.16_{1.17}$ | $75.60_{0.78}$ | $76.79_{0.51}$ | $77.30_{0.73}$ | $76.71_{0.97}$ |
| frozen ($\lambda=1$) | $77.52_{0.81}$ | $79.41_{0.76}$ | $81.46_{0.82}$ | $82.94_{0.67}$ | $83.87_{1.02}$ |
| post-hoc | $81.82_{0.69}$ | $83.34_{0.63}$ | $84.37_{0.56}$ | $85.27_{0.45}$ | $85.62_{0.69}$ |
| p-h w losses | $\mathbf{81.88}_{0.50}$ | $83.37_{0.59}$ | $\mathbf{84.77}_{0.55}$ | $85.41_{0.76}$ | $86.05_{0.93}$ |
| p-h w sampling | $81.78_{0.50}$ | $\mathbf{83.47}_{0.47}$ | $84.54_{0.48}$ | $\mathbf{85.45}_{0.54}$ | $\mathbf{86.26}_{0.87}$ |
| Actor | | | | | |
| N2V (inductive) | $33.99_{0.93}$ | $35.54_{1.01}$ | $36.64_{0.85}$ | $\mathbf{38.22}_{0.70}$ | $38.08_{1.62}$ |
| frozen ($\lambda=1$) | $33.07_{1.00}$ | $34.32_{0.97}$ | $36.57_{1.18}$ | $37.91_{0.76}$ | $37.80_{1.56}$ |
| post-hoc | $34.96_{0.78}$ | $\mathbf{36.16}_{0.53}$ | $37.13_{1.08}$ | $37.75_{1.10}$ | $\mathbf{38.11}_{1.18}$ |
| p-h w losses | $34.97_{0.54}$ | $35.95_{0.39}$ | $37.40_{1.19}$ | $37.53_{1.02}$ | $37.32_{1.22}$ |
| p-h w sampling | $\mathbf{35.15}_{1.06}$ | $35.92_{0.56}$ | $\mathbf{37.66}_{0.96}$ | $37.72_{0.89}$ | $38.07_{1.12}$ |
| Amazon-ratings | | | | | |
| N2V (inductive) | $37.36_{0.39}$ | $37.93_{0.59}$ | $38.30_{0.50}$ | $38.51_{0.69}$ | $38.57_{1.67}$ |
| frozen ($\lambda=1$) | $39.49_{0.78}$ | $41.98_{1.16}$ | $48.63_{2.84}$ | $49.49_{0.80}$ | $51.99_{1.25}$ |
| post-hoc | $41.24_{0.74}$ | $45.64_{0.67}$ | $51.89_{0.53}$ | $55.90_{0.77}$ | $\mathbf{58.69}_{0.44}$ |
| p-h w losses | $41.11_{0.66}$ | $45.56_{0.96}$ | $\mathbf{52.08}_{0.49}$ | $\mathbf{55.96}_{0.81}$ | $58.44_{0.67}$ |
| p-h w sampling | $\mathbf{41.26}_{0.69}$ | $\mathbf{45.73}_{0.67}$ | $51.91_{0.40}$ | $55.93_{0.68}$ | $51.69_{1.60}$ |
| Roman-empire | | | | | |
| N2V (inductive) | $\mathbf{62.37}_{0.51}$ | $\mathbf{64.35}_{0.38}$ | $\mathbf{65.67}_{0.18}$ | $\mathbf{66.48}_{0.70}$ | $\mathbf{66.54}_{1.07}$ |
| frozen ($\lambda=1$) | $60.57_{0.41}$ | $63.64_{0.31}$ | $65.36_{0.35}$ | $65.98_{0.67}$ | $66.11_{1.17}$ |
| post-hoc | $60.33_{0.33}$ | $63.49_{0.33}$ | $65.07_{0.39}$ | $65.83_{0.80}$ | $65.97_{1.23}$ |
| p-h w losses | $60.57_{0.39}$ | $63.59_{0.32}$ | $65.28_{0.34}$ | $65.81_{0.56}$ | $65.97_{1.32}$ |
| p-h w sampling | $60.04_{0.37}$ | $63.57_{0.30}$ | $65.16_{0.36}$ | $65.86_{0.59}$ | $65.99_{1.12}$ |

*Table 12.* GraphSAGE on different iN2V setups concatenated with the original graph features.

| Dataset | 10% | 20% | 40% | 60% | 80% |
|---|---|---|---|---|---|
| Cora | | | | | |
| N2V (inductive) | $73.97_{3.35}$ | $81.92_{1.84}$ | $85.97_{1.67}$ | $\mathbf{86.97}_{0.96}$ | $87.38_{0.79}$ |
| frozen ($\lambda=1$) | $76.56_{1.72}$ | $81.51_{1.77}$ | $85.12_{1.38}$ | $86.62_{1.87}$ | $87.60_{1.81}$ |
| post-hoc | $81.01_{1.42}$ | $81.98_{1.75}$ | $\mathbf{86.13}_{1.08}$ | $86.55_{1.35}$ | $86.64_{1.26}$ |
| p-h w losses | $\mathbf{81.45}_{1.39}$ | $82.82_{1.20}$ | $85.92_{1.17}$ | $86.35_{1.56}$ | $87.45_{1.39}$ |
| p-h w sampling | $80.97_{1.47}$ | $\mathbf{83.28}_{1.41}$ | $85.71_{1.44}$ | $86.77_{1.42}$ | $\mathbf{87.64}_{1.27}$ |
| Citeseer | | | | | |
| N2V (inductive) | $\mathbf{68.52}_{2.47}$ | $\mathbf{72.49}_{1.14}$ | $\mathbf{74.53}_{1.28}$ | $75.85_{1.40}$ | $76.76_{1.85}$ |
| frozen ($\lambda=1$) | $65.62_{1.85}$ | $70.69_{1.10}$ | $73.99_{1.47}$ | $75.77_{1.29}$ | $\mathbf{77.09}_{1.77}$ |
| post-hoc | $66.79_{1.51}$ | $71.14_{0.60}$ | $73.91_{1.38}$ | $\mathbf{76.35}_{1.42}$ | $75.50_{1.63}$ |
| p-h w losses | $67.11_{1.47}$ | $71.14_{0.98}$ | $73.51_{1.81}$ | $75.94_{1.41}$ | $76.55_{2.79}$ |
| p-h w sampling | $66.69_{1.34}$ | $69.10_{1.60}$ | $73.47_{1.54}$ | $75.73_{1.27}$ | $76.13_{2.14}$ |
| Pubmed | | | | | |
| N2V (inductive) | $\mathbf{84.33}_{0.46}$ | $\mathbf{86.06}_{0.59}$ | $\mathbf{87.32}_{0.38}$ | $87.32_{0.39}$ | $88.01_{0.78}$ |
| frozen ($\lambda=1$) | $81.23_{0.69}$ | $84.86_{0.58}$ | $86.74_{0.60}$ | $87.92_{0.47}$ | $88.38_{0.88}$ |
| post-hoc | $82.83_{0.77}$ | $84.76_{0.56}$ | $86.78_{0.56}$ | $\mathbf{88.69}_{0.49}$ | $89.23_{0.69}$ |
| p-h w losses | $82.75_{0.71}$ | $85.09_{0.37}$ | $86.68_{0.48}$ | $88.36_{0.47}$ | $89.07_{0.69}$ |
| p-h w sampling | $83.02_{0.72}$ | $84.82_{0.46}$ | $87.51_{0.51}$ | $88.59_{0.35}$ | $\mathbf{89.39}_{0.72}$ |
| Computers | | | | | |
| N2V (inductive) | $86.64_{1.11}$ | $89.05_{0.63}$ | $90.72_{0.36}$ | $91.33_{0.56}$ | $91.70_{0.55}$ |
| frozen ($\lambda=1$) | $88.52_{0.51}$ | $90.17_{0.31}$ | $91.16_{0.35}$ | $91.63_{0.30}$ | $91.60_{0.52}$ |
| post-hoc | $88.83_{0.51}$ | $\mathbf{90.50}_{0.24}$ | $91.47_{0.39}$ | $91.80_{0.59}$ | $\mathbf{92.01}_{0.35}$ |
| p-h w losses | $88.83_{0.34}$ | $90.46_{0.19}$ | $91.43_{0.36}$ | $91.90_{0.38}$ | $91.85_{0.61}$ |
| p-h w sampling | $\mathbf{88.90}_{0.36}$ | $90.50_{0.22}$ | $\mathbf{91.47}_{0.30}$ | $\mathbf{91.91}_{0.49}$ | $91.72_{0.37}$ |
| Photo | | | | | |
| N2V (inductive) | $91.97_{1.73}$ | $93.87_{0.70}$ | $94.94_{0.44}$ | $95.37_{0.60}$ | $95.56_{0.81}$ |
| frozen ($\lambda=1$) | $93.04_{0.51}$ | $94.29_{0.47}$ | $95.25_{0.30}$ | $95.54_{0.50}$ | $95.63_{0.75}$ |
| post-hoc | $93.74_{0.41}$ | $94.53_{0.45}$ | $95.23_{0.31}$ | $95.54_{0.61}$ | $95.73_{0.61}$ |
| p-h w losses | $93.54_{0.49}$ | $94.56_{0.33}$ | $95.14_{0.42}$ | $95.44_{0.49}$ | $\mathbf{95.74}_{0.85}$ |
| p-h w sampling | $\mathbf{93.87}_{0.52}$ | $\mathbf{94.68}_{0.33}$ | $\mathbf{95.26}_{0.28}$ | $\mathbf{95.59}_{0.42}$ | $95.63_{0.74}$ |
| WikiCS | | | | | |
| N2V (inductive) | $76.50_{1.34}$ | $79.99_{0.65}$ | $82.23_{0.75}$ | $83.76_{0.63}$ | $84.79_{0.87}$ |
| frozen ($\lambda=1$) | $79.29_{0.98}$ | $81.62_{0.51}$ | $83.29_{0.71}$ | $84.44_{0.65}$ | $85.38_{0.45}$ |
| post-hoc | $81.50_{0.64}$ | $83.30_{0.62}$ | $84.41_{0.51}$ | $\mathbf{85.43}_{0.48}$ | $85.97_{0.78}$ |
| p-h w losses | $\mathbf{81.54}_{0.68}$ | $83.30_{0.59}$ | $84.47_{0.64}$ | $85.25_{0.47}$ | $\mathbf{86.00}_{0.90}$ |
| p-h w sampling | $81.36_{0.52}$ | $\mathbf{83.38}_{0.58}$ | $\mathbf{84.62}_{0.47}$ | $85.29_{0.66}$ | $85.73_{0.70}$ |
| Actor | | | | | |
| N2V (inductive) | $32.04_{0.94}$ | $33.44_{0.73}$ | $35.14_{0.93}$ | $36.13_{1.52}$ | $36.05_{2.46}$ |
| frozen ($\lambda=1$) | $31.29_{0.58}$ | $32.75_{0.46}$ | $34.78_{0.77}$ | $\mathbf{36.23}_{1.14}$ | $36.46_{1.84}$ |
| post-hoc | $32.12_{0.73}$ | $33.18_{0.45}$ | $34.89_{0.75}$ | $36.04_{1.06}$ | $36.66_{2.24}$ |
| p-h w losses | $\mathbf{32.82}_{0.74}$ | $\mathbf{33.83}_{0.91}$ | $\mathbf{35.82}_{0.86}$ | $35.21_{0.75}$ | $\mathbf{36.76}_{2.21}$ |
| p-h w sampling | $32.79_{1.03}$ | $33.53_{0.63}$ | $35.38_{1.24}$ | $36.03_{0.64}$ | $36.58_{1.50}$ |
| Amazon-ratings | | | | | |
| N2V (inductive) | $39.14_{0.57}$ | $41.43_{0.65}$ | $44.99_{1.04}$ | $48.97_{1.16}$ | $50.98_{1.35}$ |
| frozen ($\lambda=1$) | $38.76_{0.69}$ | $42.28_{0.54}$ | $48.16_{1.51}$ | $50.02_{0.59}$ | $54.05_{1.04}$ |
| post-hoc | $39.93_{0.48}$ | $43.45_{0.94}$ | $50.70_{0.51}$ | $55.12_{0.80}$ | $58.70_{0.84}$ |
| p-h w losses | $\mathbf{40.29}_{0.67}$ | $43.72_{0.70}$ | $50.63_{0.90}$ | $55.29_{0.65}$ | $\mathbf{58.94}_{0.89}$ |
| p-h w sampling | $40.13_{0.60}$ | $\mathbf{43.91}_{1.02}$ | $\mathbf{50.99}_{0.84}$ | $\mathbf{55.42}_{0.83}$ | $49.54_{1.30}$ |
| Roman-empire | | | | | |
| N2V (inductive) | $\mathbf{64.71}_{0.66}$ | $\mathbf{68.52}_{0.65}$ | $\mathbf{72.78}_{0.31}$ | $\mathbf{76.68}_{0.64}$ | $\mathbf{81.16}_{1.65}$ |
| frozen ($\lambda=1$) | $57.86_{1.21}$ | $65.40_{0.54}$ | $70.93_{0.39}$ | $75.48_{0.57}$ | $80.58_{0.71}$ |
| post-hoc | $57.94_{1.32}$ | $65.57_{0.44}$ | $71.50_{0.43}$ | $76.07_{0.49}$ | $80.48_{0.75}$ |
| p-h w losses | $59.29_{0.87}$ | $66.10_{0.31}$ | $70.45_{0.38}$ | $75.62_{0.64}$ | $79.92_{0.96}$ |
| p-h w sampling | $55.56_{1.30}$ | $66.19_{0.59}$ | $70.88_{0.38}$ | $75.15_{0.84}$ | $79.91_{0.71}$ |

*Table 13.* Comparison of best iN2V variant vs baselines; GAT accuracy. Gray numbers are not directly comparable as they use additional information (graph features/transductive setup).

| Dataset | Percentage of training data | | | | |
| --- | --- | --- | --- | --- | --- |
|  | 10% | 20% | 40% | 60% | 80% |
| **Cora** | | | | | |
| N2V (inductive) | $69.16_{5.02}$ | $77.14_{1.54}$ | $81.97_{1.67}$ | $83.47_{1.42}$ | $85.20_{1.53}$ |
| Feature Propagation | $79.10_{1.02}$ | $\mathbf{81.90_{1.53}}$ | $83.02_{1.75}$ | $84.30_{1.22}$ | $85.31_{2.43}$ |
| iN2V (own) | $\mathbf{79.70_{1.29}}$ | $81.74_{1.27}$ | $\mathbf{83.90_{1.14}}$ | $\mathbf{84.58_{1.05}}$ | $\mathbf{85.68_{2.04}}$ |
| Original features | $81.35_{1.83}$ | $84.38_{1.13}$ | $86.51_{1.19}$ | $88.21_{0.84}$ | $88.15_{1.13}$ |
| N2V (transductive) | $79.20_{1.53}$ | $81.89_{1.17}$ | $84.16_{1.38}$ | $84.98_{0.87}$ | $85.94_{1.67}$ |
| **Citeseer** | | | | | |
| N2V (inductive) | $51.54_{3.24}$ | $58.89_{2.60}$ | $65.81_{1.83}$ | $70.12_{1.51}$ | $73.03_{2.83}$ |
| Feature Propagation | $56.73_{2.03}$ | $\mathbf{62.29_{1.30}}$ | $66.42_{1.40}$ | $70.47_{1.10}$ | $72.79_{2.02}$ |
| iN2V (own) | $\mathbf{58.13_{0.99}}$ | $61.74_{1.53}$ | $\mathbf{67.12_{1.30}}$ | $\mathbf{71.02_{0.90}}$ | $\mathbf{73.42_{1.91}}$ |
| Original features | $71.64_{1.30}$ | $73.71_{1.24}$ | $75.94_{1.18}$ | $76.78_{1.34}$ | $77.36_{2.04}$ |
| N2V (transductive) | $57.80_{1.27}$ | $60.98_{1.03}$ | $66.76_{1.33}$ | $69.98_{1.18}$ | $72.61_{2.47}$ |
| **Pubmed** | | | | | |
| N2V (inductive) | $72.71_{5.53}$ | $77.35_{2.18}$ | $81.74_{0.49}$ | $82.83_{0.55}$ | $83.02_{0.74}$ |
| Feature Propagation | $79.30_{0.47}$ | $80.31_{0.56}$ | $81.87_{0.37}$ | $82.93_{0.59}$ | $83.06_{0.79}$ |
| iN2V (own) | $\mathbf{80.65_{0.49}}$ | $\mathbf{81.80_{0.51}}$ | $\mathbf{82.77_{0.53}}$ | $\mathbf{83.17_{0.43}}$ | $\mathbf{83.34_{0.72}}$ |
| Original features | $85.02_{0.42}$ | $85.79_{0.42}$ | $87.47_{0.45}$ | $88.21_{0.42}$ | $88.85_{0.42}$ |
| N2V (transductive) | $81.32_{0.53}$ | $82.09_{0.49}$ | $83.21_{0.41}$ | $83.46_{0.47}$ | $83.50_{0.81}$ |
| **Computers** | | | | | |
| N2V (inductive) | $83.44_{2.61}$ | $87.80_{0.68}$ | $90.40_{0.49}$ | $90.92_{0.50}$ | $91.18_{0.68}$ |
| Feature Propagation | $83.09_{1.19}$ | $86.66_{0.66}$ | $89.85_{0.51}$ | $90.75_{0.53}$ | $\mathbf{91.28_{0.70}}$ |
| iN2V (own) | $\mathbf{87.80_{0.41}}$ | $\mathbf{89.19_{0.48}}$ | $\mathbf{90.60_{0.55}}$ | $\mathbf{91.06_{0.59}}$ | $91.08_{0.60}$ |
| Original features | $87.44_{0.58}$ | $90.17_{0.20}$ | $91.38_{0.29}$ | $91.89_{0.22}$ | $92.47_{0.56}$ |
| N2V (transductive) | $89.29_{0.32}$ | $90.35_{0.31}$ | $91.01_{0.32}$ | $91.33_{0.40}$ | $91.33_{0.68}$ |
| **Photo** | | | | | |
| N2V (inductive) | $86.95_{1.78}$ | $88.80_{2.49}$ | $92.52_{0.67}$ | $93.09_{0.36}$ | $\mathbf{93.54_{0.87}}$ |
| Feature Propagation | $87.18_{1.28}$ | $90.21_{0.61}$ | $92.10_{0.42}$ | $93.05_{0.50}$ | $93.40_{0.77}$ |
| iN2V (own) | $\mathbf{90.46_{0.58}}$ | $\mathbf{91.26_{0.50}}$ | $\mathbf{92.71_{0.50}}$ | $\mathbf{93.31_{0.88}}$ | $93.18_{0.81}$ |
| Original features | $93.47_{0.60}$ | $94.42_{0.59}$ | $95.02_{0.58}$ | $95.37_{0.26}$ | $95.57_{0.86}$ |
| N2V (transductive) | $91.37_{0.49}$ | $92.50_{0.27}$ | $93.18_{0.44}$ | $93.69_{0.49}$ | $93.36_{0.49}$ |
| **WikiCS** | | | | | |
| N2V (inductive) | $71.58_{2.60}$ | $76.91_{1.26}$ | $79.96_{0.73}$ | $80.87_{1.08}$ | $82.08_{0.87}$ |
| Feature Propagation | $73.42_{3.49}$ | $77.89_{1.12}$ | $80.11_{0.69}$ | $81.27_{0.75}$ | $\mathbf{82.22_{1.20}}$ |
| iN2V (own) | $\mathbf{78.21_{0.57}}$ | $\mathbf{79.63_{0.82}}$ | $\mathbf{80.37_{0.63}}$ | $\mathbf{81.56_{0.80}}$ | $82.11_{0.67}$ |
| Original features | $80.95_{0.82}$ | $82.64_{0.85}$ | $84.10_{0.46}$ | $84.61_{0.79}$ | $84.95_{0.62}$ |
| N2V (transductive) | $79.51_{0.66}$ | $80.65_{0.53}$ | $81.96_{0.59}$ | $82.25_{0.71}$ | $82.86_{0.96}$ |
| **Actor** | | | | | |
| N2V (inductive) | $25.38_{1.11}$ | $25.21_{0.85}$ | $\mathbf{26.09_{0.96}}$ | $\mathbf{25.66_{0.63}}$ | $\mathbf{25.28_{1.86}}$ |
| Feature Propagation | $24.97_{1.38}$ | $25.63_{0.85}$ | $25.18_{0.96}$ | $25.53_{0.62}$ | $25.07_{1.62}$ |
| iN2V (own) | $\mathbf{25.53_{0.95}}$ | $\mathbf{25.89_{0.94}}$ | $24.80_{0.93}$ | $24.98_{0.84}$ | $25.20_{1.62}$ |
| Original features | $30.82_{0.86}$ | $32.23_{0.87}$ | $32.96_{0.96}$ | $33.72_{1.03}$ | $34.39_{1.51}$ |
| N2V (transductive) | $25.20_{0.81}$ | $25.69_{0.61}$ | $24.50_{1.37}$ | $24.30_{1.53}$ | $24.50_{0.71}$ |
| **Amazon-ratings** | | | | | |
| N2V (inductive) | $39.33_{0.71}$ | $43.06_{0.58}$ | $45.92_{0.68}$ | $48.58_{0.43}$ | $50.27_{0.62}$ |
| Feature Propagation | $40.97_{0.84}$ | $42.92_{0.46}$ | $\mathbf{46.44_{0.77}}$ | $48.87_{0.54}$ | $51.21_{1.12}$ |
| iN2V (own) | $\mathbf{41.49_{0.76}}$ | $\mathbf{43.58_{0.51}}$ | $46.35_{0.59}$ | $\mathbf{49.22_{0.81}}$ | $\mathbf{51.56_{1.48}}$ |
| Original features | $40.67_{0.56}$ | $42.48_{0.66}$ | $46.06_{0.53}$ | $48.75_{0.60}$ | $50.64_{1.70}$ |
| N2V (transductive) | $41.78_{0.64}$ | $43.92_{0.89}$ | $47.67_{0.79}$ | $49.85_{0.81}$ | $51.57_{1.21}$ |
| **Roman-empire** | | | | | |
| N2V (inductive) | $13.86_{0.26}$ | $13.93_{0.24}$ | $\mathbf{14.30_{0.40}}$ | $14.48_{1.15}$ | $15.90_{1.45}$ |
| Feature Propagation | $13.88_{0.25}$ | $13.94_{0.25}$ | $13.94_{0.22}$ | $14.32_{0.75}$ | $16.04_{0.85}$ |
| iN2V (own) | $\mathbf{13.91_{0.19}}$ | $\mathbf{13.94_{0.22}}$ | $14.21_{0.37}$ | $\mathbf{16.05_{0.53}}$ | $\mathbf{17.49_{1.05}}$ |
| Original features | $52.23_{0.82}$ | $59.46_{0.80}$ | $68.98_{0.60}$ | $74.95_{0.43}$ | $78.89_{1.37}$ |
| N2V (transductive) | $13.75_{0.19}$ | $13.63_{0.43}$ | $14.01_{0.32}$ | $14.13_{0.72}$ | $13.72_{0.66}$ |

*Table 14.* Comparison of best iN2V variant vs baselines; GIN accuracy. Gray numbers are not directly comparable as they use additional information (graph features/transductive setup).

| Dataset | Percentage of training data | | | | |
| --- | --- | --- | --- | --- | --- |
|  | 10% | 20% | 40% | 60% | 80% |
| **Cora** | | | | | |
| N2V (inductive) | $68.38_{2.63}$ | $74.48_{2.55}$ | $81.76_{1.10}$ | $83.97_{1.39}$ | $\mathbf{85.20_{1.92}}$ |
| Feature Propagation | $78.27_{1.37}$ | $81.02_{1.56}$ | $82.40_{1.72}$ | $84.32_{1.10}$ | $85.13_{1.98}$ |
| iN2V (own) | $\mathbf{79.58_{1.16}}$ | $\mathbf{81.21_{1.24}}$ | $\mathbf{84.03_{1.25}}$ | $\mathbf{84.91_{1.10}}$ | $85.06_{1.67}$ |
| Original features | $82.40_{1.08}$ | $84.05_{0.79}$ | $86.70_{1.39}$ | $88.08_{1.05}$ | $87.34_{0.95}$ |
| N2V (transductive) | $79.09_{1.35}$ | $81.56_{1.35}$ | $83.23_{1.26}$ | $85.31_{0.85}$ | $86.09_{1.85}$ |
| **Citeseer** | | | | | |
| N2V (inductive) | $50.40_{2.60}$ | $58.06_{1.75}$ | $64.83_{2.19}$ | $69.31_{2.08}$ | $72.91_{1.88}$ |
| Feature Propagation | $56.45_{2.41}$ | $60.72_{2.14}$ | $65.97_{1.61}$ | $70.48_{1.31}$ | $\mathbf{73.30_{1.73}}$ |
| iN2V (own) | $\mathbf{57.69_{1.25}}$ | $\mathbf{61.24_{2.15}}$ | $\mathbf{67.01_{1.65}}$ | $\mathbf{70.74_{1.48}}$ | $72.64_{2.21}$ |
| Original features | $71.66_{0.86}$ | $73.74_{1.19}$ | $75.44_{1.08}$ | $76.39_{0.77}$ | $77.18_{1.80}$ |
| N2V (transductive) | $57.31_{1.58}$ | $61.86_{1.11}$ | $66.49_{1.17}$ | $70.48_{1.05}$ | $73.24_{2.39}$ |
| **Pubmed** | | | | | |
| N2V (inductive) | $76.22_{1.28}$ | $79.78_{1.02}$ | $81.91_{0.52}$ | $82.73_{0.47}$ | $83.07_{0.82}$ |
| Feature Propagation | $78.45_{0.55}$ | $79.19_{0.51}$ | $81.53_{0.59}$ | $82.71_{0.42}$ | $\mathbf{83.09_{0.89}}$ |
| iN2V (own) | $\mathbf{80.49_{0.51}}$ | $\mathbf{81.60_{0.53}}$ | $\mathbf{82.64_{0.57}}$ | $\mathbf{83.28_{0.55}}$ | $83.07_{0.61}$ |
| Original features | $84.94_{0.27}$ | $86.03_{0.43}$ | $87.31_{0.38}$ | $88.08_{0.50}$ | $88.47_{0.59}$ |
| N2V (transductive) | $81.31_{0.48}$ | $82.21_{0.45}$ | $83.23_{0.41}$ | $83.47_{0.57}$ | $83.46_{0.69}$ |
| **Computers** | | | | | |
| N2V (inductive) | $86.06_{1.24}$ | $88.86_{0.57}$ | $90.22_{0.44}$ | $91.00_{0.52}$ | $91.36_{0.70}$ |
| Feature Propagation | $84.34_{0.97}$ | $87.82_{0.47}$ | $89.61_{0.61}$ | $90.94_{0.39}$ | $91.08_{0.36}$ |
| iN2V (own) | $\mathbf{88.43_{0.57}}$ | $\mathbf{89.63_{0.42}}$ | $\mathbf{90.71_{0.58}}$ | $\mathbf{91.32_{0.54}}$ | $\mathbf{91.45_{0.36}}$ |
| Original features | $87.57_{0.34}$ | $89.47_{0.26}$ | $90.97_{0.33}$ | $91.43_{0.41}$ | $91.85_{0.45}$ |
| N2V (transductive) | $88.97_{0.31}$ | $89.95_{0.71}$ | $90.71_{0.41}$ | $91.22_{0.46}$ | $91.39_{0.55}$ |
| **Photo** | | | | | |
| N2V (inductive) | $88.95_{0.76}$ | $90.50_{0.64}$ | $92.19_{0.48}$ | $\mathbf{93.13_{0.67}}$ | $\mathbf{93.56_{0.44}}$ |
| Feature Propagation | $87.68_{1.15}$ | $90.35_{0.29}$ | $92.07_{0.55}$ | $92.86_{0.41}$ | $93.20_{0.98}$ |
| iN2V (own) | $\mathbf{90.60_{0.55}}$ | $\mathbf{91.53_{0.44}}$ | $\mathbf{92.27_{0.45}}$ | $92.81_{0.34}$ | $93.45_{0.75}$ |
| Original features | $93.42_{0.62}$ | $94.30_{0.54}$ | $95.05_{0.30}$ | $95.34_{0.51}$ | $95.70_{0.58}$ |
| N2V (transductive) | $91.15_{0.50}$ | $92.22_{0.38}$ | $92.93_{0.48}$ | $93.46_{0.49}$ | $93.59_{1.00}$ |
| **WikiCS** | | | | | |
| N2V (inductive) | $76.68_{1.36}$ | $78.50_{0.84}$ | $79.85_{0.65}$ | $80.92_{0.80}$ | $81.94_{0.96}$ |
| Feature Propagation | $75.73_{2.67}$ | $78.43_{1.18}$ | $80.34_{0.78}$ | $80.98_{0.86}$ | $81.88_{1.40}$ |
| iN2V (own) | $\mathbf{78.51_{0.61}}$ | $\mathbf{80.03_{0.78}}$ | $\mathbf{80.96_{0.53}}$ | $\mathbf{81.56_{0.68}}$ | $\mathbf{82.69_{0.86}}$ |
| Original features | $80.82_{0.89}$ | $82.40_{0.81}$ | $83.89_{0.55}$ | $84.73_{0.82}$ | $85.45_{0.54}$ |
| N2V (transductive) | $79.23_{0.56}$ | $80.56_{0.73}$ | $81.66_{0.63}$ | $82.30_{0.63}$ | $82.55_{0.78}$ |
| **Actor** | | | | | |
| N2V (inductive) | $\mathbf{25.33_{0.97}}$ | $\mathbf{25.84_{1.07}}$ | $\mathbf{25.70_{0.76}}$ | $24.91_{1.03}$ | $24.62_{1.70}$ |
| Feature Propagation | $25.20_{0.92}$ | $25.72_{0.88}$ | $25.59_{0.57}$ | $\mathbf{25.66_{0.57}}$ | $25.29_{1.17}$ |
| iN2V (own) | $24.23_{1.46}$ | $25.48_{1.15}$ | $25.64_{0.90}$ | $25.26_{0.76}$ | $\mathbf{25.53_{2.13}}$ |
| Original features | $30.24_{0.72}$ | $31.80_{0.88}$ | $32.97_{0.65}$ | $33.70_{0.78}$ | $35.50_{1.91}$ |
| N2V (transductive) | $25.28_{1.21}$ | $25.37_{1.14}$ | $24.15_{0.65}$ | $24.86_{1.14}$ | $24.99_{0.71}$ |
| **Amazon-ratings** | | | | | |
| N2V (inductive) | $40.58_{0.77}$ | $\mathbf{43.56_{0.32}}$ | $46.29_{0.94}$ | $49.84_{0.49}$ | $53.49_{0.68}$ |
| Feature Propagation | $40.74_{0.73}$ | $43.24_{0.69}$ | $46.85_{0.92}$ | $50.56_{0.91}$ | $\mathbf{53.83_{0.93}}$ |
| iN2V (own) | $\mathbf{41.17_{0.71}}$ | $43.32_{0.63}$ | $\mathbf{47.35_{0.84}}$ | $\mathbf{50.93_{0.79}}$ | $53.59_{1.06}$ |
| Original features | $39.87_{0.67}$ | $41.48_{0.52}$ | $45.30_{0.67}$ | $48.18_{1.17}$ | $52.18_{1.28}$ |
| N2V (transductive) | $41.77_{0.52}$ | $44.47_{0.45}$ | $48.34_{0.60}$ | $50.38_{0.50}$ | $52.45_{0.79}$ |
| **Roman-empire** | | | | | |
| N2V (inductive) | $13.89_{0.24}$ | $13.82_{0.80}$ | $17.24_{2.64}$ | $\mathbf{22.57_{1.56}}$ | $\mathbf{26.66_{1.06}}$ |
| Feature Propagation | $13.89_{0.24}$ | $13.83_{0.55}$ | $16.58_{2.01}$ | $19.74_{2.99}$ | $24.87_{1.90}$ |
| iN2V (own) | $\mathbf{13.92_{0.27}}$ | $\mathbf{13.98_{0.32}}$ | $\mathbf{18.34_{2.28}}$ | $20.32_{2.67}$ | $24.94_{1.41}$ |
| Original features | $58.01_{0.33}$ | $62.58_{0.74}$ | $69.23_{0.44}$ | $73.74_{0.55}$ | $76.29_{0.69}$ |
| N2V (transductive) | $19.61_{1.67}$ | $21.11_{0.46}$ | $24.47_{1.17}$ | $27.82_{1.36}$ | $31.55_{2.46}$ |

*Table 15.* Comparison of N2V vs different iN2V setups, GAT accuracy.

| Dataset | 10% | 20% | 40% | 60% | 80% |
|---|---|---|---|---|---|
| **Cora** | | | | | |
| N2V (inductive) | $69.16_{5.02}$ | $77.14_{1.54}$ | $81.97_{1.67}$ | $83.47_{1.42}$ | $85.20_{1.53}$ |
| frozen ($\lambda = 1$) | $75.96_{1.82}$ | $80.49_{1.61}$ | $82.91_{1.18}$ | $84.45_{1.08}$ | $85.06_{1.62}$ |
| post-hoc | $79.52_{1.25}$ | $81.15_{1.59}$ | $83.85_{0.99}$ | $84.21_{1.31}$ | $85.42_{1.91}$ |
| p-h w losses | $79.16_{1.16}$ | $81.26_{1.26}$ | $\mathbf{83.90}_{1.14}$ | $84.35_{1.54}$ | $\mathbf{85.68}_{2.04}$ |
| p-h w sampling | $\mathbf{79.70}_{1.29}$ | $\mathbf{81.74}_{1.27}$ | $83.44_{0.69}$ | $\mathbf{84.58}_{1.05}$ | $85.35_{1.65}$ |
| **Citeseer** | | | | | |
| N2V (inductive) | $51.54_{3.24}$ | $58.89_{2.60}$ | $65.81_{1.83}$ | $70.12_{1.51}$ | $73.03_{2.83}$ |
| frozen ($\lambda = 1$) | $57.13_{1.22}$ | $61.47_{1.69}$ | $66.18_{1.82}$ | $70.09_{1.14}$ | $73.24_{2.15}$ |
| post-hoc | $57.87_{1.19}$ | $61.38_{1.35}$ | $67.09_{1.00}$ | $70.72_{1.63}$ | $72.67_{1.67}$ |
| p-h w losses | $58.13_{0.99}$ | $61.74_{1.53}$ | $\mathbf{67.23}_{1.45}$ | $70.18_{1.41}$ | $\mathbf{73.42}_{1.91}$ |
| p-h w sampling | $\mathbf{58.28}_{0.98}$ | $\mathbf{61.96}_{1.47}$ | $67.12_{1.30}$ | $\mathbf{71.02}_{0.90}$ | $72.97_{1.74}$ |
| **Pubmed** | | | | | |
| N2V (inductive) | $72.71_{5.53}$ | $77.35_{2.18}$ | $81.74_{0.49}$ | $82.83_{0.55}$ | $83.02_{0.74}$ |
| frozen ($\lambda = 1$) | $79.25_{0.64}$ | $80.45_{0.63}$ | $82.11_{0.49}$ | $82.87_{0.66}$ | $83.21_{0.68}$ |
| post-hoc | $80.57_{0.33}$ | $\mathbf{81.87}_{0.58}$ | $82.66_{0.45}$ | $\mathbf{83.27}_{0.39}$ | $83.34_{0.72}$ |
| p-h w losses | $80.65_{0.49}$ | $81.80_{0.51}$ | $82.63_{0.47}$ | $83.17_{0.43}$ | $\mathbf{83.35}_{0.72}$ |
| p-h w sampling | $\mathbf{80.71}_{0.56}$ | $81.79_{0.53}$ | $82.77_{0.53}$ | $83.20_{0.51}$ | $83.09_{0.84}$ |
| **Computers** | | | | | |
| N2V (inductive) | $83.44_{2.61}$ | $87.80_{0.68}$ | $90.40_{0.49}$ | $90.92_{0.50}$ | $\mathbf{91.18}_{0.68}$ |
| frozen ($\lambda = 1$) | $86.02_{0.79}$ | $88.23_{0.55}$ | $90.00_{0.44}$ | $90.55_{0.47}$ | $91.05_{0.56}$ |
| post-hoc | $87.55_{0.36}$ | $89.20_{0.34}$ | $90.28_{0.39}$ | $90.89_{0.46}$ | $90.74_{0.74}$ |
| p-h w losses | $87.77_{0.50}$ | $89.19_{0.48}$ | $90.31_{0.40}$ | $90.97_{0.41}$ | $91.13_{0.67}$ |
| p-h w sampling | $\mathbf{87.80}_{0.41}$ | $\mathbf{89.32}_{0.34}$ | $\mathbf{90.60}_{0.55}$ | $\mathbf{91.06}_{0.59}$ | $91.08_{0.60}$ |
| **Photo** | | | | | |
| N2V (inductive) | $86.95_{1.78}$ | $88.80_{2.49}$ | $92.52_{0.67}$ | $93.09_{0.36}$ | $93.54_{0.87}$ |
| frozen ($\lambda = 1$) | $89.24_{0.76}$ | $90.73_{0.55}$ | $92.41_{0.51}$ | $93.14_{0.39}$ | $\mathbf{93.62}_{0.72}$ |
| post-hoc | $90.41_{0.45}$ | $91.26_{0.50}$ | $92.51_{0.51}$ | $93.11_{0.79}$ | $93.25_{0.86}$ |
| p-h w losses | $\mathbf{90.46}_{0.58}$ | $91.53_{0.49}$ | $92.37_{0.32}$ | $93.06_{0.56}$ | $93.52_{0.58}$ |
| p-h w sampling | $90.19_{0.61}$ | $\mathbf{91.71}_{0.42}$ | $\mathbf{92.71}_{0.50}$ | $\mathbf{93.31}_{0.88}$ | $93.18_{0.81}$ |
| **WikiCS** | | | | | |
| N2V (inductive) | $71.58_{2.60}$ | $76.91_{1.26}$ | $79.96_{0.73}$ | $80.87_{1.08}$ | $82.08_{0.87}$ |
| frozen ($\lambda = 1$) | $76.51_{1.08}$ | $78.09_{1.17}$ | $79.56_{0.56}$ | $81.29_{0.74}$ | $82.04_{0.77}$ |
| post-hoc | $78.11_{0.97}$ | $79.45_{0.89}$ | $80.33_{0.80}$ | $81.09_{0.69}$ | $\mathbf{82.32}_{0.59}$ |
| p-h w losses | $78.00_{0.59}$ | $79.55_{0.93}$ | $80.37_{0.63}$ | $81.12_{0.82}$ | $82.13_{0.79}$ |
| p-h w sampling | $\mathbf{78.21}_{0.57}$ | $\mathbf{79.63}_{0.82}$ | $\mathbf{80.59}_{0.45}$ | $\mathbf{81.56}_{0.80}$ | $82.11_{0.67}$ |
| **Actor** | | | | | |
| N2V (inductive) | $25.38_{1.11}$ | $25.21_{0.85}$ | $\mathbf{26.09}_{0.96}$ | $\mathbf{25.66}_{0.63}$ | $25.28_{1.86}$ |
| frozen ($\lambda = 1$) | $25.23_{0.91}$ | $25.79_{0.68}$ | $24.80_{0.93}$ | $24.98_{0.84}$ | $25.20_{1.62}$ |
| post-hoc | $\mathbf{26.01}_{0.65}$ | $\mathbf{26.03}_{0.56}$ | $25.46_{0.68}$ | $24.97_{0.63}$ | $25.01_{1.37}$ |
| p-h w losses | $25.41_{1.06}$ | $25.89_{0.94}$ | $25.70_{1.01}$ | $24.91_{0.61}$ | $\mathbf{25.50}_{1.97}$ |
| p-h w sampling | $25.53_{0.95}$ | $25.66_{1.14}$ | $25.53_{1.19}$ | $24.91_{0.69}$ | $23.99_{1.74}$ |
| **Amazon-ratings** | | | | | |
| N2V (inductive) | $39.33_{0.71}$ | $43.06_{0.58}$ | $45.92_{0.68}$ | $48.58_{0.43}$ | $50.27_{0.62}$ |
| frozen ($\lambda = 1$) | $40.22_{0.89}$ | $43.30_{0.40}$ | $\mathbf{46.35}_{0.59}$ | $\mathbf{49.22}_{0.81}$ | $\mathbf{51.56}_{1.48}$ |
| post-hoc | $41.48_{0.75}$ | $\mathbf{43.58}_{0.51}$ | $45.85_{0.54}$ | $47.45_{0.57}$ | $48.00_{1.23}$ |
| p-h w losses | $41.49_{0.76}$ | $43.38_{0.61}$ | $45.75_{0.67}$ | $48.15_{0.55}$ | $49.23_{1.25}$ |
| p-h w sampling | $\mathbf{41.63}_{0.82}$ | $43.49_{0.66}$ | $45.77_{0.62}$ | $47.19_{0.74}$ | $48.89_{1.00}$ |
| **Roman-empire** | | | | | |
| N2V (inductive) | $13.86_{0.26}$ | $13.93_{0.24}$ | $\mathbf{14.30}_{0.40}$ | $14.48_{1.15}$ | $15.90_{1.45}$ |
| frozen ($\lambda = 1$) | $\mathbf{13.91}_{0.19}$ | $\mathbf{13.94}_{0.22}$ | $13.88_{0.31}$ | $14.42_{0.69}$ | $16.86_{0.75}$ |
| post-hoc | $13.84_{0.25}$ | $13.82_{0.29}$ | $14.14_{0.75}$ | $\mathbf{16.05}_{0.53}$ | $\mathbf{17.49}_{1.05}$ |
| p-h w losses | $13.84_{0.28}$ | $13.86_{0.23}$ | $14.21_{0.37}$ | $14.16_{0.38}$ | $15.63_{1.05}$ |
| p-h w sampling | $13.89_{0.20}$ | $13.78_{0.34}$ | $13.90_{0.29}$ | $14.00_{0.56}$ | $15.65_{0.89}$ |

*Table 16.* Comparison of N2V vs different iN2V setups, GIN accuracy.

| Dataset | 10% | 20% | 40% | 60% | 80% |
|---|---|---|---|---|---|
| **Cora** | | | | | |
| N2V (inductive) | $68.38_{2.63}$ | $74.48_{2.55}$ | $81.76_{1.10}$ | $83.97_{1.39}$ | $85.20_{1.92}$ |
| frozen ($\lambda = 1$) | $75.57_{2.08}$ | $79.45_{1.36}$ | $82.23_{1.26}$ | $83.69_{1.19}$ | $84.58_{2.36}$ |
| post-hoc | $79.42_{1.37}$ | $80.66_{1.64}$ | $\mathbf{84.03}_{1.25}$ | $84.10_{1.52}$ | $85.09_{2.21}$ |
| p-h w losses | $79.51_{1.10}$ | $\mathbf{81.21}_{1.46}$ | $83.53_{1.30}$ | $83.71_{1.70}$ | $\mathbf{85.72}_{1.86}$ |
| p-h w sampling | $\mathbf{79.58}_{1.16}$ | $81.21_{1.24}$ | $83.15_{0.74}$ | $\mathbf{84.91}_{1.10}$ | $85.06_{1.67}$ |
| **Citeseer** | | | | | |
| N2V (inductive) | $50.40_{2.60}$ | $58.06_{1.75}$ | $64.83_{2.19}$ | $69.31_{2.08}$ | $72.91_{1.88}$ |
| frozen ($\lambda = 1$) | $56.23_{1.39}$ | $59.00_{2.13}$ | $65.57_{1.56}$ | $69.91_{1.64}$ | $72.85_{2.20}$ |
| post-hoc | $\mathbf{57.78}_{1.30}$ | $61.42_{1.65}$ | $66.89_{1.82}$ | $70.74_{1.48}$ | $\mathbf{73.12}_{1.86}$ |
| p-h w losses | $57.69_{1.25}$ | $61.24_{2.15}$ | $66.77_{2.36}$ | $\mathbf{70.87}_{0.80}$ | $72.97_{1.69}$ |
| p-h w sampling | $57.75_{1.23}$ | $\mathbf{61.55}_{1.07}$ | $\mathbf{67.01}_{1.65}$ | $70.24_{1.62}$ | $72.64_{2.21}$ |
| **Pubmed** | | | | | |
| N2V (inductive) | $76.22_{1.28}$ | $79.78_{1.02}$ | $81.91_{0.52}$ | $82.73_{0.47}$ | $83.07_{0.82}$ |
| frozen ($\lambda = 1$) | $78.15_{0.54}$ | $80.23_{0.63}$ | $81.73_{0.32}$ | $82.57_{0.50}$ | $82.79_{0.81}$ |
| post-hoc | $80.45_{0.34}$ | $\mathbf{81.60}_{0.53}$ | $82.53_{0.38}$ | $\mathbf{83.39}_{0.38}$ | $83.22_{0.75}$ |
| p-h w losses | $\mathbf{80.49}_{0.51}$ | $81.53_{0.46}$ | $82.77_{0.46}$ | $83.21_{0.43}$ | $83.07_{0.61}$ |
| p-h w sampling | $80.42_{0.45}$ | $81.52_{0.45}$ | $82.64_{0.57}$ | $83.28_{0.55}$ | $83.10_{0.59}$ |
| **Computers** | | | | | |
| N2V (inductive) | $86.06_{1.24}$ | $88.86_{0.57}$ | $90.22_{0.44}$ | $91.00_{0.52}$ | $91.36_{0.70}$ |
| frozen ($\lambda = 1$) | $86.62_{0.44}$ | $88.36_{0.71}$ | $89.96_{0.62}$ | $90.76_{0.34}$ | $91.18_{0.75}$ |
| post-hoc | $88.21_{0.42}$ | $\mathbf{89.78}_{0.34}$ | $90.54_{0.44}$ | $90.88_{0.52}$ | $91.27_{0.63}$ |
| p-h w losses | $88.10_{0.52}$ | $89.63_{0.42}$ | $\mathbf{90.74}_{0.43}$ | $91.12_{0.44}$ | $\mathbf{91.45}_{0.36}$ |
| p-h w sampling | $\mathbf{88.43}_{0.57}$ | $89.60_{0.45}$ | $90.71_{0.58}$ | $\mathbf{91.32}_{0.54}$ | $91.45_{0.72}$ |
| **Photo** | | | | | |
| N2V (inductive) | $88.95_{0.76}$ | $90.50_{0.64}$ | $92.19_{0.48}$ | $\mathbf{93.13}_{0.67}$ | $\mathbf{93.56}_{0.44}$ |
| frozen ($\lambda = 1$) | $89.45_{0.76}$ | $91.04_{0.47}$ | $\mathbf{92.40}_{0.46}$ | $93.11_{0.54}$ | $93.45_{0.75}$ |
| post-hoc | $\mathbf{90.60}_{0.54}$ | $\mathbf{91.60}_{0.40}$ | $92.34_{0.65}$ | $92.75_{0.71}$ | $93.45_{0.64}$ |
| p-h w losses | $90.60_{0.55}$ | $91.58_{0.38}$ | $92.36_{0.44}$ | $92.81_{0.34}$ | $93.23_{0.84}$ |
| p-h w sampling | $90.59_{0.50}$ | $91.53_{0.44}$ | $92.27_{0.45}$ | $92.69_{0.64}$ | $93.02_{0.78}$ |
| **WikiCS** | | | | | |
| N2V (inductive) | $76.68_{1.36}$ | $78.50_{0.84}$ | $79.85_{0.65}$ | $80.92_{0.80}$ | $81.94_{0.96}$ |
| frozen ($\lambda = 1$) | $77.17_{0.86}$ | $78.41_{0.98}$ | $80.14_{0.69}$ | $80.76_{0.96}$ | $82.26_{0.78}$ |
| post-hoc | $78.59_{0.45}$ | $79.98_{0.84}$ | $80.89_{0.51}$ | $81.91_{0.81}$ | $\mathbf{82.69}_{0.86}$ |
| p-h w losses | $78.51_{0.61}$ | $79.91_{0.78}$ | $80.95_{0.50}$ | $81.56_{0.68}$ | $82.26_{0.91}$ |
| p-h w sampling | $\mathbf{78.62}_{0.50}$ | $\mathbf{80.03}_{0.78}$ | $\mathbf{80.96}_{0.53}$ | $\mathbf{81.97}_{0.68}$ | $82.67_{0.78}$ |
| **Actor** | | | | | |
| N2V (inductive) | $25.33_{0.97}$ | $\mathbf{25.84}_{1.07}$ | $25.70_{0.76}$ | $24.91_{1.03}$ | $24.62_{1.70}$ |
| frozen ($\lambda = 1$) | $\mathbf{25.71}_{0.78}$ | $25.48_{1.15}$ | $25.64_{0.90}$ | $25.26_{0.76}$ | $25.32_{1.21}$ |
| post-hoc | $25.10_{1.09}$ | $25.53_{1.03}$ | $25.06_{1.34}$ | $25.18_{1.34}$ | $25.96_{0.74}$ |
| p-h w losses | $25.30_{1.06}$ | $25.76_{1.01}$ | $\mathbf{25.94}_{0.69}$ | $\mathbf{25.40}_{1.15}$ | $25.53_{2.13}$ |
| p-h w sampling | $24.23_{1.46}$ | $25.52_{1.00}$ | $25.72_{0.90}$ | $25.12_{1.01}$ | $\mathbf{26.49}_{0.74}$ |
| **Amazon-ratings** | | | | | |
| N2V (inductive) | $40.58_{0.77}$ | $\mathbf{43.56}_{0.32}$ | $46.29_{0.94}$ | $49.84_{0.49}$ | $53.49_{0.68}$ |
| frozen ($\lambda = 1$) | $40.15_{0.81}$ | $43.42_{0.57}$ | $\mathbf{47.35}_{0.84}$ | $\mathbf{50.93}_{0.79}$ | $\mathbf{53.59}_{1.06}$ |
| post-hoc | $\mathbf{41.26}_{0.58}$ | $43.18_{0.59}$ | $45.72_{0.71}$ | $48.02_{0.42}$ | $48.33_{0.99}$ |
| p-h w losses | $41.17_{0.71}$ | $42.94_{0.78}$ | $45.84_{0.35}$ | $48.79_{1.12}$ | $50.19_{1.28}$ |
| p-h w sampling | $41.14_{0.77}$ | $43.32_{0.63}$ | $45.75_{0.75}$ | $47.18_{0.48}$ | $49.67_{1.14}$ |
| **Roman-empire** | | | | | |
| N2V (inductive) | $13.89_{0.24}$ | $13.82_{0.80}$ | $17.24_{2.64}$ | $\mathbf{22.57}_{1.56}$ | $\mathbf{26.66}_{1.06}$ |
| frozen ($\lambda = 1$) | $13.89_{0.24}$ | $13.98_{0.32}$ | $17.36_{2.30}$ | $19.72_{1.74}$ | $24.94_{1.41}$ |
| post-hoc | $13.89_{0.24}$ | $13.96_{0.38}$ | $\mathbf{18.34}_{2.28}$ | $19.22_{2.72}$ | $24.47_{1.94}$ |
| p-h w losses | $13.89_{0.24}$ | $\mathbf{14.17}_{0.55}$ | $17.00_{2.05}$ | $20.32_{2.67}$ | $24.88_{0.90}$ |
| p-h w sampling | $\mathbf{13.92}_{0.27}$ | $13.85_{0.50}$ | $18.07_{1.61}$ | $20.49_{3.18}$ | $24.72_{1.79}$ |

*Table 17.* Comparison of best iN2V variant vs baselines concatenated with the original graph features; GAT accuracy. Gray numbers are not directly comparable as they use additional information (transductive setup).

| | Percentage of training data | | | | |
|---|---|---|---|---|---|
| Dataset | 10% | 20% | 40% | 60% | 80% |
| **Cora** | | | | | |
| N2V (inductive) | $80.65_{2.14}$ | $83.12_{1.22}$ | $86.08_{1.52}$ | $87.25_{1.09}$ | $\mathbf{88.27}_{1.48}$ |
| Feature Propagation | $81.56_{1.14}$ | $84.08_{1.37}$ | $86.49_{1.33}$ | $\mathbf{88.12}_{1.36}$ | $87.56_{1.35}$ |
| **iN2V** (own) | $\mathbf{81.68}_{1.23}$ | $\mathbf{84.11}_{1.12}$ | $\mathbf{86.70}_{1.36}$ | $87.99_{0.85}$ | $87.93_{1.53}$ |
| Original features | $81.35_{1.83}$ | $84.38_{1.13}$ | $86.51_{1.19}$ | $88.21_{0.84}$ | $88.15_{1.13}$ |
| N2V (transductive) | $82.69_{1.21}$ | $84.90_{1.08}$ | $86.93_{1.38}$ | $87.68_{1.09}$ | $88.27_{1.29}$ |
| **Citeseer** | | | | | |
| N2V (inductive) | $\mathbf{71.10}_{1.37}$ | $\mathbf{73.02}_{1.25}$ | $74.79_{1.40}$ | $\mathbf{76.93}_{1.05}$ | $\mathbf{77.63}_{1.99}$ |
| Feature Propagation | $69.46_{1.06}$ | $72.03_{1.46}$ | $\mathbf{75.81}_{0.65}$ | $76.30_{1.23}$ | $77.15_{1.61}$ |
| **iN2V** (own) | $68.04_{1.39}$ | $71.48_{0.89}$ | $75.11_{0.91}$ | $76.69_{1.42}$ | $77.15_{2.36}$ |
| Original features | $71.64_{1.30}$ | $73.71_{1.24}$ | $75.94_{1.18}$ | $76.78_{1.34}$ | $77.36_{2.04}$ |
| N2V (transductive) | $70.91_{1.13}$ | $73.12_{0.99}$ | $75.69_{0.79}$ | $76.71_{1.34}$ | $77.18_{1.54}$ |
| **Pubmed** | | | | | |
| N2V (inductive) | $\mathbf{84.12}_{0.50}$ | $\mathbf{85.41}_{0.39}$ | $86.42_{0.54}$ | $87.24_{0.45}$ | $87.92_{0.63}$ |
| Feature Propagation | $83.93_{0.44}$ | $85.30_{0.43}$ | $\mathbf{86.61}_{0.39}$ | $87.18_{0.36}$ | $87.63_{0.54}$ |
| **iN2V** (own) | $83.51_{0.49}$ | $84.86_{0.46}$ | $86.44_{0.53}$ | $\mathbf{87.57}_{0.43}$ | $\mathbf{87.97}_{0.81}$ |
| Original features | $85.02_{0.42}$ | $85.79_{0.42}$ | $87.47_{0.45}$ | $88.21_{0.42}$ | $88.85_{0.42}$ |
| N2V (transductive) | $85.20_{0.50}$ | $87.16_{0.44}$ | $87.20_{0.52}$ | $88.23_{0.56}$ | $88.80_{0.51}$ |
| **Computers** | | | | | |
| N2V (inductive) | $85.44_{1.32}$ | $89.01_{0.77}$ | $90.83_{0.54}$ | $91.64_{0.52}$ | $92.29_{0.52}$ |
| Feature Propagation | $87.25_{1.02}$ | $89.95_{0.53}$ | $91.25_{0.49}$ | $91.93_{0.35}$ | $\mathbf{92.31}_{0.58}$ |
| **iN2V** (own) | $\mathbf{88.72}_{0.48}$ | $\mathbf{90.28}_{0.27}$ | $\mathbf{91.51}_{0.52}$ | $\mathbf{91.94}_{0.52}$ | $92.12_{0.37}$ |
| Original features | $87.44_{0.58}$ | $90.17_{0.20}$ | $91.38_{0.29}$ | $91.89_{0.22}$ | $92.47_{0.56}$ |
| N2V (transductive) | $90.54_{0.29}$ | $91.46_{0.33}$ | $92.19_{0.37}$ | $92.53_{0.31}$ | $92.74_{0.31}$ |
| **Photo** | | | | | |
| N2V (inductive) | $91.32_{1.81}$ | $93.17_{0.77}$ | $94.93_{0.48}$ | $95.01_{0.45}$ | $95.57_{0.51}$ |
| Feature Propagation | $93.17_{0.61}$ | $94.39_{0.34}$ | $95.09_{0.39}$ | $95.25_{0.44}$ | $95.61_{0.73}$ |
| **iN2V** (own) | $\mathbf{93.82}_{0.54}$ | $\mathbf{94.71}_{0.37}$ | $\mathbf{95.23}_{0.41}$ | $\mathbf{95.61}_{0.47}$ | $\mathbf{95.75}_{0.68}$ |
| Original features | $93.47_{0.60}$ | $94.42_{0.59}$ | $95.02_{0.58}$ | $95.37_{0.26}$ | $95.57_{0.86}$ |
| N2V (transductive) | $94.33_{0.33}$ | $94.95_{0.24}$ | $95.42_{0.42}$ | $95.63_{0.36}$ | $95.92_{0.67}$ |
| **WikiCS** | | | | | |
| N2V (inductive) | $77.94_{0.97}$ | $80.22_{0.85}$ | $82.03_{0.85}$ | $83.06_{0.82}$ | $84.37_{1.17}$ |
| Feature Propagation | $79.69_{0.96}$ | $81.51_{0.90}$ | $83.26_{0.75}$ | $84.43_{0.73}$ | $84.98_{0.97}$ |
| **iN2V** (own) | $\mathbf{81.21}_{0.54}$ | $\mathbf{82.70}_{0.81}$ | $\mathbf{83.75}_{0.71}$ | $\mathbf{84.63}_{0.63}$ | $\mathbf{85.11}_{0.77}$ |
| Original features | $80.95_{0.82}$ | $82.64_{0.85}$ | $84.10_{0.46}$ | $84.61_{0.79}$ | $84.95_{0.62}$ |
| N2V (transductive) | $81.57_{0.63}$ | $83.04_{0.59}$ | $84.00_{0.50}$ | $84.83_{0.59}$ | $85.22_{1.05}$ |
| **Actor** | | | | | |
| N2V (inductive) | $30.70_{0.73}$ | $31.71_{0.75}$ | $33.10_{0.83}$ | $\mathbf{33.64}_{0.79}$ | $34.01_{2.26}$ |
| Feature Propagation | $30.51_{0.72}$ | $31.83_{1.04}$ | $33.10_{1.21}$ | $33.35_{1.05}$ | $33.99_{1.54}$ |
| **iN2V** (own) | $\mathbf{30.81}_{0.81}$ | $\mathbf{32.35}_{0.72}$ | $\mathbf{33.18}_{1.17}$ | $33.34_{1.31}$ | $34.21_{1.92}$ |
| Original features | $30.82_{0.86}$ | $32.23_{0.87}$ | $32.96_{0.96}$ | $33.72_{1.03}$ | $34.39_{1.51}$ |
| N2V (transductive) | $29.45_{0.71}$ | $31.13_{0.96}$ | $32.50_{1.00}$ | $33.35_{1.34}$ | $33.67_{2.04}$ |
| **Amazon-ratings** | | | | | |
| N2V (inductive) | $40.23_{0.78}$ | $43.04_{0.69}$ | $45.80_{0.75}$ | $48.15_{0.74}$ | $51.23_{1.01}$ |
| Feature Propagation | $\mathbf{41.74}_{0.66}$ | $\mathbf{43.78}_{0.55}$ | $46.85_{0.75}$ | $49.00_{0.58}$ | $\mathbf{51.49}_{0.88}$ |
| **iN2V** (own) | $41.50_{0.72}$ | $43.62_{0.81}$ | $\mathbf{47.16}_{0.72}$ | $\mathbf{49.51}_{0.53}$ | $51.19_{0.94}$ |
| Original features | $40.67_{0.56}$ | $42.48_{0.66}$ | $46.06_{0.53}$ | $48.75_{0.60}$ | $50.64_{1.70}$ |
| N2V (transductive) | $42.02_{0.53}$ | $44.65_{0.61}$ | $48.96_{0.25}$ | $51.48_{0.67}$ | $53.07_{0.91}$ |
| **Roman-empire** | | | | | |
| N2V (inductive) | $49.85_{1.92}$ | $57.09_{0.95}$ | $63.17_{0.97}$ | $71.36_{0.57}$ | $76.53_{2.07}$ |
| Feature Propagation | $48.42_{1.11}$ | $54.87_{0.95}$ | $62.28_{0.55}$ | $69.30_{0.84}$ | $74.91_{0.91}$ |
| **iN2V** (own) | $43.34_{1.12}$ | $55.04_{0.59}$ | $64.40_{0.90}$ | $69.46_{0.51}$ | $74.14_{1.04}$ |
| Original features | $52.23_{0.82}$ | $59.46_{0.80}$ | $68.98_{0.60}$ | $74.95_{0.43}$ | $78.89_{1.37}$ |
| N2V (transductive) | $55.55_{0.52}$ | $62.07_{0.78}$ | $68.74_{0.63}$ | $73.50_{0.93}$ | $76.51_{1.16}$ |

*Table 18.* Comparison of best iN2V variant vs baselines concatenated with the original graph features; GIN accuracy. Gray numbers are not directly comparable as they use additional information (transductive setup).

| | Percentage of training data | | | | |
|---|---|---|---|---|---|
| Dataset | 10% | 20% | 40% | 60% | 80% |
| **Cora** | | | | | |
| N2V (inductive) | $81.53_{1.44}$ | $\mathbf{84.26}_{1.18}$ | $\mathbf{86.53}_{1.10}$ | $87.40_{2.26}$ | $87.38_{1.40}$ |
| Feature Propagation | $\mathbf{82.54}_{1.40}$ | $84.16_{1.34}$ | $85.83_{1.44}$ | $\mathbf{87.44}_{1.34}$ | $87.38_{1.42}$ |
| **iN2V** (own) | $81.68_{1.23}$ | $83.80_{1.02}$ | $86.51_{1.12}$ | $87.32_{1.01}$ | $\mathbf{87.79}_{1.43}$ |
| Original features | $82.40_{1.08}$ | $84.05_{0.79}$ | $86.70_{1.39}$ | $88.08_{1.05}$ | $87.34_{0.95}$ |
| N2V (transductive) | $82.42_{1.11}$ | $84.28_{0.89}$ | $86.43_{1.21}$ | $87.21_{1.06}$ | $88.34_{1.33}$ |
| **Citeseer** | | | | | |
| N2V (inductive) | $71.04_{1.07}$ | $\mathbf{73.64}_{0.99}$ | $74.56_{0.97}$ | $76.68_{1.26}$ | $76.64_{2.60}$ |
| Feature Propagation | $69.51_{0.65}$ | $72.20_{1.22}$ | $\mathbf{75.12}_{1.17}$ | $\mathbf{76.87}_{1.01}$ | $\mathbf{77.60}_{2.02}$ |
| **iN2V** (own) | $68.47_{0.80}$ | $71.62_{1.14}$ | $74.31_{1.00}$ | $76.75_{1.49}$ | $77.36_{2.20}$ |
| Original features | $71.66_{0.86}$ | $73.74_{1.19}$ | $75.44_{1.08}$ | $76.39_{0.77}$ | $77.18_{1.80}$ |
| N2V (transductive) | $70.30_{0.99}$ | $72.96_{0.79}$ | $74.68_{1.22}$ | $75.91_{1.44}$ | $77.21_{2.40}$ |
| **Pubmed** | | | | | |
| N2V (inductive) | $84.01_{0.37}$ | $85.53_{0.35}$ | $86.65_{0.42}$ | $87.29_{0.47}$ | $\mathbf{88.05}_{0.35}$ |
| Feature Propagation | $\mathbf{84.16}_{0.22}$ | $\mathbf{85.55}_{0.30}$ | $\mathbf{86.81}_{0.54}$ | $87.51_{0.39}$ | $88.04_{0.53}$ |
| **iN2V** (own) | $83.75_{0.49}$ | $84.95_{0.33}$ | $86.79_{0.53}$ | $\mathbf{87.65}_{0.31}$ | $88.00_{0.58}$ |
| Original features | $84.94_{0.27}$ | $86.03_{0.43}$ | $87.31_{0.38}$ | $88.08_{0.50}$ | $88.47_{0.59}$ |
| N2V (transductive) | $84.68_{0.46}$ | $86.00_{0.39}$ | $86.63_{0.57}$ | $87.47_{0.39}$ | $87.74_{0.43}$ |
| **Computers** | | | | | |
| N2V (inductive) | $85.46_{1.30}$ | $88.31_{0.84}$ | $90.29_{0.35}$ | $91.30_{0.69}$ | $91.54_{0.45}$ |
| Feature Propagation | $87.07_{0.62}$ | $89.52_{0.38}$ | $90.95_{0.37}$ | $91.49_{0.40}$ | $\mathbf{91.88}_{0.64}$ |
| **iN2V** (own) | $\mathbf{88.60}_{0.17}$ | $\mathbf{90.09}_{0.41}$ | $\mathbf{91.38}_{0.36}$ | $\mathbf{91.76}_{0.33}$ | $91.80_{0.52}$ |
| Original features | $87.57_{0.34}$ | $89.47_{0.26}$ | $90.97_{0.33}$ | $91.43_{0.41}$ | $91.85_{0.45}$ |
| N2V (transductive) | $90.04_{0.43}$ | $91.09_{0.31}$ | $91.71_{0.43}$ | $92.04_{0.39}$ | $91.93_{0.57}$ |
| **Photo** | | | | | |
| N2V (inductive) | $91.26_{1.30}$ | $92.96_{0.75}$ | $94.83_{0.42}$ | $95.05_{0.42}$ | $95.35_{0.56}$ |
| Feature Propagation | $92.87_{0.58}$ | $94.24_{0.56}$ | $94.92_{0.43}$ | $95.20_{0.55}$ | $95.41_{0.52}$ |
| **iN2V** (own) | $\mathbf{93.81}_{0.38}$ | $\mathbf{94.75}_{0.37}$ | $\mathbf{95.25}_{0.38}$ | $\mathbf{95.56}_{0.48}$ | $\mathbf{95.76}_{0.73}$ |
| Original features | $93.42_{0.62}$ | $94.30_{0.54}$ | $95.05_{0.30}$ | $95.34_{0.51}$ | $95.70_{0.58}$ |
| N2V (transductive) | $94.20_{0.34}$ | $94.93_{0.34}$ | $95.60_{0.15}$ | $95.65_{0.35}$ | $95.92_{0.56}$ |
| **WikiCS** | | | | | |
| N2V (inductive) | $77.69_{1.46}$ | $80.48_{0.43}$ | $82.43_{0.77}$ | $83.80_{0.54}$ | $84.94_{0.88}$ |
| Feature Propagation | $79.43_{1.15}$ | $81.61_{0.71}$ | $82.96_{0.83}$ | $84.29_{0.57}$ | $84.74_{1.02}$ |
| **iN2V** (own) | $\mathbf{81.26}_{0.67}$ | $\mathbf{82.98}_{0.72}$ | $\mathbf{84.26}_{0.53}$ | $\mathbf{84.80}_{0.67}$ | $\mathbf{85.49}_{0.93}$ |
| Original features | $80.82_{0.89}$ | $82.40_{0.81}$ | $83.89_{0.55}$ | $84.73_{0.82}$ | $85.45_{0.54}$ |
| N2V (transductive) | $81.70_{0.61}$ | $83.16_{0.67}$ | $84.13_{0.73}$ | $84.78_{0.53}$ | $85.22_{0.73}$ |
| **Actor** | | | | | |
| N2V (inductive) | $30.39_{1.03}$ | $32.11_{0.75}$ | $33.20_{1.12}$ | $\mathbf{34.32}_{0.85}$ | $34.80_{1.63}$ |
| Feature Propagation | $30.28_{0.97}$ | $31.65_{0.91}$ | $33.08_{0.59}$ | $33.85_{0.95}$ | $34.05_{1.40}$ |
| **iN2V** (own) | $\mathbf{30.57}_{0.65}$ | $\mathbf{32.62}_{0.84}$ | $\mathbf{33.92}_{0.93}$ | $34.08_{1.17}$ | $\mathbf{34.87}_{1.11}$ |
| Original features | $30.24_{0.72}$ | $31.80_{0.88}$ | $32.97_{0.65}$ | $33.70_{0.78}$ | $35.50_{1.91}$ |
| N2V (transductive) | $28.95_{0.71}$ | $30.56_{0.93}$ | $32.48_{1.04}$ | $33.38_{1.13}$ | $33.86_{1.53}$ |
| **Amazon-ratings** | | | | | |
| N2V (inductive) | $40.16_{0.59}$ | $42.10_{0.59}$ | $45.36_{0.74}$ | $48.74_{0.84}$ | $53.10_{0.77}$ |
| Feature Propagation | $41.30_{0.73}$ | $\mathbf{43.90}_{0.56}$ | $47.44_{0.68}$ | $50.39_{0.88}$ | $53.59_{0.83}$ |
| **iN2V** (own) | $\mathbf{41.32}_{0.80}$ | $43.58_{0.71}$ | $\mathbf{47.54}_{0.68}$ | $\mathbf{50.99}_{0.76}$ | $\mathbf{54.23}_{1.12}$ |
| Original features | $39.87_{0.67}$ | $41.48_{0.52}$ | $45.30_{0.67}$ | $48.18_{1.17}$ | $52.18_{1.28}$ |
| N2V (transductive) | $42.21_{0.47}$ | $45.17_{0.63}$ | $49.53_{0.51}$ | $51.99_{0.74}$ | $54.00_{1.05}$ |
| **Roman-empire** | | | | | |
| N2V (inductive) | $\mathbf{57.60}_{0.97}$ | $\mathbf{62.77}_{0.49}$ | $\mathbf{68.75}_{0.40}$ | $\mathbf{73.50}_{0.46}$ | $\mathbf{76.17}_{0.69}$ |
| Feature Propagation | $56.49_{1.17}$ | $61.36_{0.60}$ | $67.70_{0.70}$ | $72.28_{0.53}$ | $75.40_{1.06}$ |
| **iN2V** (own) | $52.69_{0.76}$ | $59.17_{0.56}$ | $66.86_{0.46}$ | $71.91_{0.54}$ | $75.46_{0.84}$ |
| Original features | $58.01_{0.33}$ | $62.58_{0.74}$ | $69.23_{0.44}$ | $73.74_{0.55}$ | $76.29_{0.69}$ |
| N2V (transductive) | $61.38_{0.68}$ | $67.92_{0.64}$ | $73.50_{0.50}$ | $75.71_{0.50}$ | $76.59_{0.89}$ |

*Table 19.* GAT on different iN2V setups concatenated with the original graph features.

| Dataset | Percentage of training data | | | | |
| --- | --- | --- | --- | --- | --- |
| | 10% | 20% | 40% | 60% | 80% |
| **Cora** | | | | | |
| N2V (inductive) | $80.65_{2.14}$ | $83.12_{1.22}$ | $86.08_{1.52}$ | $87.25_{1.09}$ | $\mathbf{88.27}_{1.48}$ |
| frozen ($\lambda=1$) | $79.19_{1.56}$ | $82.24_{1.62}$ | $85.43_{1.39}$ | $86.99_{1.34}$ | $87.93_{1.53}$ |
| post-hoc | $81.75_{1.27}$ | $83.06_{1.67}$ | $\mathbf{86.70}_{1.36}$ | $86.85_{1.46}$ | $87.75_{1.87}$ |
| p-h w losses | $\mathbf{81.78}_{1.67}$ | $83.55_{1.46}$ | $86.53_{0.93}$ | $87.45_{1.18}$ | $87.12_{1.99}$ |
| p-h w sampling | $81.68_{1.23}$ | $\mathbf{84.11}_{1.12}$ | $86.16_{1.14}$ | $\mathbf{87.99}_{0.85}$ | $88.08_{1.58}$ |
| **Citeseer** | | | | | |
| N2V (inductive) | $\mathbf{71.10}_{1.37}$ | $\mathbf{73.02}_{1.25}$ | $74.79_{1.40}$ | $\mathbf{76.93}_{1.05}$ | $\mathbf{77.63}_{1.99}$ |
| frozen ($\lambda=1$) | $67.90_{1.20}$ | $71.18_{1.15}$ | $\mathbf{75.11}_{0.91}$ | $75.71_{1.30}$ | $76.85_{2.04}$ |
| post-hoc | $68.10_{1.23}$ | $71.35_{1.16}$ | $74.06_{1.39}$ | $76.69_{1.42}$ | $77.15_{2.36}$ |
| p-h w losses | $68.04_{1.39}$ | $71.48_{0.89}$ | $73.71_{1.68}$ | $75.80_{1.89}$ | $76.61_{2.00}$ |
| p-h w sampling | $68.06_{1.24}$ | $71.14_{1.11}$ | $74.42_{1.17}$ | $75.64_{1.39}$ | $77.06_{2.45}$ |
| **Pubmed** | | | | | |
| N2V (inductive) | $\mathbf{84.12}_{0.50}$ | $\mathbf{85.41}_{0.39}$ | $86.42_{0.54}$ | $87.24_{0.45}$ | $87.92_{0.63}$ |
| frozen ($\lambda=1$) | $82.83_{0.64}$ | $84.93_{0.54}$ | $86.35_{0.57}$ | $86.89_{0.52}$ | $87.17_{0.54}$ |
| post-hoc | $83.40_{0.31}$ | $84.70_{0.40}$ | $86.44_{0.53}$ | $87.57_{0.43}$ | $87.97_{0.81}$ |
| p-h w losses | $83.51_{0.49}$ | $84.86_{0.46}$ | $86.13_{0.54}$ | $87.51_{0.30}$ | $87.88_{0.69}$ |
| p-h w sampling | $83.41_{0.47}$ | $84.65_{0.43}$ | $\mathbf{86.61}_{0.42}$ | $\mathbf{87.62}_{0.37}$ | $\mathbf{88.15}_{0.51}$ |
| **Computers** | | | | | |
| N2V (inductive) | $85.44_{1.32}$ | $89.01_{0.77}$ | $90.83_{0.54}$ | $91.64_{0.52}$ | $\mathbf{92.29}_{0.52}$ |
| frozen ($\lambda=1$) | $87.93_{0.92}$ | $90.06_{0.35}$ | $91.30_{0.36}$ | $92.03_{0.52}$ | $92.20_{0.54}$ |
| post-hoc | $88.37_{0.50}$ | $\mathbf{90.28}_{0.27}$ | $91.37_{0.26}$ | $91.94_{0.52}$ | $92.23_{0.44}$ |
| p-h w losses | $88.39_{0.61}$ | $90.25_{0.35}$ | $91.47_{0.51}$ | $91.89_{0.43}$ | $92.12_{0.59}$ |
| p-h w sampling | $\mathbf{88.72}_{0.48}$ | $90.13_{0.27}$ | $\mathbf{91.51}_{0.37}$ | $\mathbf{92.13}_{0.39}$ | $92.12_{0.37}$ |
| **Photo** | | | | | |
| N2V (inductive) | $91.32_{1.81}$ | $93.17_{0.77}$ | $94.93_{0.48}$ | $95.01_{0.45}$ | $95.57_{0.51}$ |
| frozen ($\lambda=1$) | $92.61_{0.50}$ | $94.16_{0.59}$ | $95.01_{0.27}$ | $95.18_{0.49}$ | $95.48_{0.59}$ |
| post-hoc | $93.76_{0.47}$ | $94.68_{0.55}$ | $95.16_{0.38}$ | $95.35_{0.44}$ | $\mathbf{95.78}_{0.86}$ |
| p-h w losses | $93.66_{0.52}$ | $94.53_{0.35}$ | $95.10_{0.43}$ | $95.16_{0.49}$ | $95.42_{0.82}$ |
| p-h w sampling | $\mathbf{93.82}_{0.54}$ | $\mathbf{94.71}_{0.37}$ | $\mathbf{95.23}_{0.41}$ | $\mathbf{95.61}_{0.47}$ | $95.75_{0.68}$ |
| **WikiCS** | | | | | |
| N2V (inductive) | $77.94_{0.97}$ | $80.22_{0.85}$ | $82.03_{0.85}$ | $83.06_{0.82}$ | $84.37_{1.17}$ |
| frozen ($\lambda=1$) | $79.10_{0.92}$ | $81.17_{0.69}$ | $82.72_{0.66}$ | $83.16_{0.70}$ | $83.90_{0.82}$ |
| post-hoc | $\mathbf{81.21}_{0.54}$ | $82.60_{0.72}$ | $83.55_{0.52}$ | $84.56_{0.60}$ | $85.11_{0.77}$ |
| p-h w losses | $81.02_{0.59}$ | $82.62_{0.71}$ | $83.61_{0.76}$ | $\mathbf{84.71}_{0.66}$ | $84.87_{0.88}$ |
| p-h w sampling | $81.02_{0.78}$ | $\mathbf{82.70}_{0.81}$ | $\mathbf{83.75}_{0.71}$ | $84.63_{0.63}$ | $\mathbf{85.29}_{1.16}$ |
| **Actor** | | | | | |
| N2V (inductive) | $30.70_{0.73}$ | $31.71_{0.75}$ | $33.10_{0.83}$ | $33.64_{0.79}$ | $34.01_{2.26}$ |
| frozen ($\lambda=1$) | $29.92_{0.87}$ | $30.97_{0.61}$ | $32.68_{0.77}$ | $33.28_{1.33}$ | $33.20_{2.10}$ |
| post-hoc | $30.42_{1.55}$ | $32.00_{0.79}$ | $32.83_{0.85}$ | $33.68_{0.80}$ | $\mathbf{34.24}_{1.32}$ |
| p-h w losses | $30.66_{0.59}$ | $\mathbf{32.35}_{0.72}$ | $33.18_{1.17}$ | $\mathbf{33.84}_{0.92}$ | $33.11_{1.50}$ |
| p-h w sampling | $\mathbf{30.81}_{0.81}$ | $31.80_{0.92}$ | $\mathbf{33.20}_{0.87}$ | $33.34_{1.31}$ | $34.21_{1.92}$ |
| **Amazon-ratings** | | | | | |
| N2V (inductive) | $40.23_{0.78}$ | $43.04_{0.69}$ | $45.80_{0.75}$ | $48.15_{0.74}$ | $51.23_{1.01}$ |
| frozen ($\lambda=1$) | $40.96_{0.69}$ | $43.35_{0.41}$ | $\mathbf{47.16}_{0.72}$ | $\mathbf{49.51}_{0.53}$ | $\mathbf{51.76}_{1.38}$ |
| post-hoc | $41.30_{0.45}$ | $43.44_{0.55}$ | $46.38_{0.62}$ | $48.88_{0.51}$ | $51.48_{1.32}$ |
| p-h w losses | $\mathbf{41.50}_{0.72}$ | $43.42_{0.56}$ | $46.63_{0.76}$ | $49.08_{0.79}$ | $51.19_{0.94}$ |
| p-h w sampling | $41.31_{0.70}$ | $\mathbf{43.62}_{0.81}$ | $46.65_{0.63}$ | $48.32_{0.69}$ | $48.97_{1.41}$ |
| **Roman-empire** | | | | | |
| N2V (inductive) | $\mathbf{49.85}_{1.92}$ | $\mathbf{57.09}_{0.95}$ | $63.17_{0.97}$ | $\mathbf{71.36}_{0.57}$ | $\mathbf{76.53}_{2.07}$ |
| frozen ($\lambda=1$) | $43.20_{1.36}$ | $54.02_{0.85}$ | $61.61_{0.83}$ | $68.65_{0.62}$ | $74.19_{1.65}$ |
| post-hoc | $43.34_{1.12}$ | $54.62_{1.04}$ | $\mathbf{64.40}_{0.90}$ | $69.46_{0.51}$ | $74.14_{1.04}$ |
| p-h w losses | $43.26_{1.28}$ | $55.04_{0.59}$ | $63.81_{0.65}$ | $66.72_{0.95}$ | $73.95_{1.33}$ |
| p-h w sampling | $41.55_{0.88}$ | $53.68_{0.71}$ | $60.86_{0.94}$ | $67.16_{1.00}$ | $73.66_{0.99}$ |

*Table 20.* GIN on different iN2V setups concatenated with the original graph features.

| Dataset | Percentage of training data | | | | |
| --- | --- | --- | --- | --- | --- |
| | 10% | 20% | 40% | 60% | 80% |
| **Cora** | | | | | |
| N2V (inductive) | $81.53_{1.44}$ | $\mathbf{84.26}_{1.18}$ | $86.53_{1.10}$ | $\mathbf{87.40}_{2.26}$ | $87.38_{1.40}$ |
| frozen ($\lambda=1$) | $79.73_{1.69}$ | $82.98_{1.49}$ | $85.44_{1.28}$ | $86.42_{1.78}$ | $\mathbf{87.79}_{1.43}$ |
| post-hoc | $\mathbf{81.68}_{1.23}$ | $82.83_{1.68}$ | $86.51_{1.12}$ | $86.92_{1.42}$ | $87.49_{1.74}$ |
| p-h w losses | $81.58_{1.34}$ | $83.34_{1.56}$ | $\mathbf{86.64}_{1.07}$ | $86.62_{1.57}$ | $87.60_{1.59}$ |
| p-h w sampling | $81.66_{1.41}$ | $83.80_{1.02}$ | $85.95_{0.74}$ | $87.32_{1.01}$ | $87.79_{1.60}$ |
| **Citeseer** | | | | | |
| N2V (inductive) | $\mathbf{71.04}_{1.07}$ | $\mathbf{73.64}_{0.99}$ | $74.56_{0.97}$ | $76.68_{1.26}$ | $76.64_{2.60}$ |
| frozen ($\lambda=1$) | $67.49_{1.02}$ | $71.32_{1.29}$ | $74.31_{1.00}$ | $76.12_{1.63}$ | $77.03_{2.22}$ |
| post-hoc | $68.47_{0.80}$ | $71.68_{1.24}$ | $74.37_{1.71}$ | $\mathbf{76.75}_{1.49}$ | $\mathbf{77.36}_{2.20}$ |
| p-h w losses | $68.20_{1.00}$ | $71.62_{1.14}$ | $74.22_{1.47}$ | $75.62_{1.46}$ | $76.94_{1.59}$ |
| p-h w sampling | $68.59_{0.88}$ | $71.62_{1.14}$ | $\mathbf{74.62}_{1.42}$ | $75.94_{1.33}$ | $76.88_{2.03}$ |
| **Pubmed** | | | | | |
| N2V (inductive) | $\mathbf{84.01}_{0.37}$ | $\mathbf{85.53}_{0.35}$ | $86.65_{0.42}$ | $87.29_{0.47}$ | $88.05_{0.35}$ |
| frozen ($\lambda=1$) | $82.46_{0.75}$ | $85.09_{0.42}$ | $86.60_{0.49}$ | $87.27_{0.51}$ | $87.65_{0.43}$ |
| post-hoc | $83.72_{0.37}$ | $84.95_{0.33}$ | $\mathbf{86.79}_{0.53}$ | $\mathbf{87.65}_{0.31}$ | $\mathbf{88.09}_{0.79}$ |
| p-h w losses | $83.64_{0.39}$ | $84.99_{0.45}$ | $86.39_{0.54}$ | $87.43_{0.48}$ | $88.02_{0.66}$ |
| p-h w sampling | $83.75_{0.49}$ | $84.99_{0.39}$ | $86.46_{0.48}$ | $87.55_{0.39}$ | $88.00_{0.58}$ |
| **Computers** | | | | | |
| N2V (inductive) | $85.46_{1.30}$ | $88.31_{0.84}$ | $90.29_{0.35}$ | $91.30_{0.69}$ | $91.54_{0.45}$ |
| frozen ($\lambda=1$) | $88.12_{0.64}$ | $89.78_{0.29}$ | $91.01_{0.36}$ | $91.68_{0.52}$ | $91.66_{0.61}$ |
| post-hoc | $88.52_{0.29}$ | $90.10_{0.24}$ | $91.38_{0.38}$ | $91.68_{0.38}$ | $91.78_{0.70}$ |
| p-h w losses | $\mathbf{88.62}_{0.31}$ | $\mathbf{90.27}_{0.34}$ | $\mathbf{91.43}_{0.36}$ | $91.76_{0.33}$ | $\mathbf{91.94}_{0.55}$ |
| p-h w sampling | $88.60_{0.17}$ | $90.09_{0.41}$ | $91.32_{0.34}$ | $\mathbf{91.84}_{0.44}$ | $91.80_{0.52}$ |
| **Photo** | | | | | |
| N2V (inductive) | $91.26_{1.30}$ | $92.96_{0.75}$ | $94.83_{0.42}$ | $95.05_{0.42}$ | $95.35_{0.56}$ |
| frozen ($\lambda=1$) | $92.64_{0.52}$ | $94.11_{0.65}$ | $94.67_{0.37}$ | $95.23_{0.46}$ | $95.25_{0.74}$ |
| post-hoc | $93.64_{0.57}$ | $94.62_{0.30}$ | $\mathbf{95.31}_{0.49}$ | $95.39_{0.49}$ | $95.73_{0.62}$ |
| p-h w losses | $93.52_{0.46}$ | $94.51_{0.49}$ | $95.14_{0.35}$ | $95.35_{0.67}$ | $95.70_{0.68}$ |
| p-h w sampling | $\mathbf{93.81}_{0.38}$ | $\mathbf{94.75}_{0.37}$ | $95.25_{0.38}$ | $\mathbf{95.56}_{0.48}$ | $\mathbf{95.76}_{0.73}$ |
| **WikiCS** | | | | | |
| N2V (inductive) | $77.69_{1.46}$ | $80.48_{0.43}$ | $82.43_{0.77}$ | $83.80_{0.54}$ | $84.94_{0.88}$ |
| frozen ($\lambda=1$) | $79.26_{0.98}$ | $81.15_{0.59}$ | $82.96_{0.74}$ | $84.02_{0.68}$ | $84.65_{1.07}$ |
| post-hoc | $81.33_{0.70}$ | $82.98_{0.72}$ | $84.12_{0.47}$ | $84.80_{0.67}$ | $85.27_{0.84}$ |
| p-h w losses | $81.26_{0.67}$ | $82.97_{0.70}$ | $\mathbf{84.26}_{0.53}$ | $\mathbf{84.91}_{0.65}$ | $85.21_{0.78}$ |
| p-h w sampling | $\mathbf{81.37}_{0.77}$ | $\mathbf{83.05}_{0.64}$ | $84.13_{0.50}$ | $84.83_{0.61}$ | $\mathbf{85.49}_{0.93}$ |
| **Actor** | | | | | |
| N2V (inductive) | $30.39_{1.03}$ | $32.11_{0.75}$ | $33.20_{1.12}$ | $\mathbf{34.32}_{0.85}$ | $34.80_{1.63}$ |
| frozen ($\lambda=1$) | $29.89_{1.05}$ | $31.46_{0.99}$ | $32.68_{0.96}$ | $33.44_{1.15}$ | $34.37_{2.04}$ |
| post-hoc | $\mathbf{30.69}_{0.76}$ | $\mathbf{32.67}_{0.87}$ | $33.46_{1.11}$ | $34.08_{1.17}$ | $\mathbf{34.87}_{1.11}$ |
| p-h w losses | $30.57_{0.65}$ | $32.62_{0.84}$ | $\mathbf{33.92}_{0.93}$ | $33.63_{0.98}$ | $34.13_{1.99}$ |
| p-h w sampling | $30.60_{0.72}$ | $32.33_{0.65}$ | $33.29_{0.89}$ | $33.72_{1.19}$ | $34.42_{1.85}$ |
| **Amazon-ratings** | | | | | |
| N2V (inductive) | $40.16_{0.59}$ | $42.10_{0.59}$ | $45.36_{0.74}$ | $48.74_{0.84}$ | $53.10_{0.77}$ |
| frozen ($\lambda=1$) | $40.96_{0.68}$ | $\mathbf{43.82}_{0.52}$ | $\mathbf{47.54}_{0.68}$ | $\mathbf{50.99}_{0.76}$ | $\mathbf{54.23}_{1.12}$ |
| post-hoc | $\mathbf{41.40}_{0.66}$ | $43.53_{0.68}$ | $46.87_{0.57}$ | $50.27_{0.56}$ | $52.74_{1.31}$ |
| p-h w losses | $41.32_{0.80}$ | $43.50_{0.71}$ | $46.74_{0.69}$ | $50.53_{0.69}$ | $53.21_{1.21}$ |
| p-h w sampling | $41.26_{0.67}$ | $43.58_{0.71}$ | $46.79_{0.58}$ | $50.04_{0.92}$ | $50.22_{1.22}$ |
| **Roman-empire** | | | | | |
| N2V (inductive) | $\mathbf{57.60}_{0.97}$ | $\mathbf{62.77}_{0.49}$ | $\mathbf{68.75}_{0.40}$ | $\mathbf{73.50}_{0.46}$ | $\mathbf{76.17}_{0.69}$ |
| frozen ($\lambda=1$) | $52.23_{0.91}$ | $59.19_{0.86}$ | $66.86_{0.46}$ | $71.91_{0.54}$ | $75.46_{0.84}$ |
| post-hoc | $51.27_{1.34}$ | $58.78_{0.92}$ | $66.50_{0.57}$ | $71.89_{0.47}$ | $74.81_{0.67}$ |
| p-h w losses | $52.69_{0.76}$ | $59.35_{0.51}$ | $66.27_{0.61}$ | $71.22_{0.56}$ | $74.97_{1.02}$ |
| p-h w sampling | $50.42_{1.05}$ | $59.17_{0.56}$ | $66.83_{0.61}$ | $71.84_{0.63}$ | $74.79_{0.80}$ |

