# OpenReview forum: "iN2V: Bringing Transductive Node Embeddings to Inductive Graphs"
_ICML.cc/2025/Conference — ICML 2025 poster_

### Official Review · Reviewer_qGa8 · 2025-02-25

**Overall Recommendation:** 3

**Summary:**

This paper proposes an inductive way to learn node features from the graph structure. Specifically, it proposes to to extend node2vec to the inductive setting with the key idea being to set the embeddings of the testing nodes to be the average embeddings of their training neighbors.

**Claims And Evidence:**

- A question I have is about the induced subgraph of a training node as mentioned in section 3.1.
- Even if the testing nodes are masked, their neighborhood are still observed to some extent right (especially given figure 1)? since testing nodes may be connected to training nodes? if so then is this really inductive? I think inductive means for example splitting the graph in two, training on one side and testing on the other? Like the PPI set which has different graphs amongst which training is done on a few graphs and testing on the other graphs?

**Essential References Not Discussed:**

No recomendations

**Experimental Designs Or Analyses:**

See section named "Methods And Evaluation Criteria"

**Methods And Evaluation Criteria:**

- See section named "Claims And Evidence"
- If I am correct for the question above, then can the proposed algorithm adapt to PPI for example? How does it perform? And what about sectioning a graph (e.g., Cora) in two or three (train, validation, test) and then reporting the performance too?
- Moreover is it also possible to see the performance when the number of training nodes is low say 10-5% and the validation set is low as well say 5-10% (since in the eperiments when the training set was small 10% the  validation set was still large 45%)

**Other Comments Or Suggestions:**

Typos:
- "This is repeated for multiple iterations to also deal with test nodes which longer distances to training nodes" should be "This is repeated for multiple iterations to also deal with test nodes with longer distances to training nodes" ?
- In section 3.4 under the heading Sampling-based, did you mean probability "r" as in the equation instead of "p" (see line 190).

**Other Strengths And Weaknesses:**

See section named "Methods And Evaluation Criteria"

**Questions For Authors:**

See section named "Methods And Evaluation Criteria"

**Relation To Broader Scientific Literature:**

See section named "Methods And Evaluation Criteria"

**Theoretical Claims:**

See section named "Methods And Evaluation Criteria"

---

> ### Author Rebuttal · Authors · 2025-03-31
>
> Question:
> about the induced subgraph of a training node as mentioned in section 3.1.
> Even if the testing nodes are masked, their neighborhood are still observed to some extent right (especially given figure 1)? since testing nodes may be connected to training nodes? if so then is this really inductive? I think inductive means for example splitting the graph in two, training on one side and testing on the other? Like the PPI set which has different graphs amongst which training is done on a few graphs and testing on the other graphs?
>
> Answer:
> During training the validation and test nodes with all edges that contain at least one of them are removed. As you correctly state, some (how many depends on the train/val/test split) test nodes have neighbors in the training set.
> Inductive means that the test nodes (features, labels, and edges) are unseen during training.
> So splitting the graph in two, training on one side and testing on the other is inductive (for testing the whole graph is used as input for the models, the metrics are calculated on the test nodes).
> PPI is a classic inductive dataset with 24 components, where the train/val/test split is based on component.
>
>
> Question:
> If I am correct for the question above, then can the proposed algorithm adapt to PPI for example? How does it perform? And what about sectioning a graph (e.g., Cora) in two or three (train, validation, test) and then reporting the performance too?
>
> Answer:
> As explained in the Limitation section, “our method can only provide embeddings to test nodes that have a path to at least one train node” (also applies to FP).
> So that a node gets an embedding during the extension to the test set, it needs to have a path (that can contain nodes from both the test and val set) to at least one train node.
> This does not have to be the case for all test nodes, as e.g. Citeseer consists of 428 components, most of which have only very few nodes. With our random splits it will happen that for some components all nodes are in the validation or test set and therefore do not get an embedding during testing.
> But for PPI, where no nodes in the val and test set have a path to nodes from the training set, our method would not produce useful embeddings.
> To apply it to PPI one would either have to switch the embedding model , e.g. from N2V to Struct2Vec (and use the internal fully connected Struc2Vec graph for the extension) or add additional edges to the PPI dataset, like adding edges that connect the same protein in the different components (such that the test nodes have connections to nodes in the 20 train graphs).
> The same methods could be tried for other graphs that are split component wise like ZINC.
> Though for very good performance on these kinds of datasets (protein/molecules) GNNs have to have some additional properties like being able to distinguish and efficiently aggregate information over cycles.
>
>
> Question:
> Moreover is it also possible to see the performance when the number of training nodes is low say 10-5% and the validation set is low as well say 5-10% (since in the eperiments when the training set was small 10% the validation set was still large 45%)
>
> Answer:
> For the experiments in the paper our focus was a wide range of training sizes (some papers claim good inductive performance but then only use over 80% of the data for training) with a balanced val and test size such that the validation performance is indicative of test performance. We expect e.g. a 10-10-80 split to perform similarly to our 10-45-45 split as the amount of training data is the same, but that the hyperparameters and early stopping generalize worse from val to test set.
> Nevertheless, we implemented a 10-10-80 split and are currently training (i)N2V embeddings on it and will add this as an additional section in the Appendix.
>
>
> Typos:
> "This is repeated for multiple iterations to also deal with test nodes which longer distances to training nodes" should be "This is repeated for multiple iterations to also deal with test nodes with longer distances to training nodes" ?
> In section 3.4 under the heading Sampling-based, did you mean probability "r" as in the equation instead of "p" (see line 190).
>
> Answer: Implemented the suggested changes.
> Yes, r/p is a typo (we switched from p to r to distinguish it from the N2V p and q hyperparameters)

---

### Official Review · Reviewer_9oof · 2025-03-09

**Overall Recommendation:** 2

**Summary:**

The paper proposes inductive node2vec (iN2V), a procedure which updates node embeddings generated by methods like node2vec which form node embeddings based on graph topology, to account for updates to the graph topology like new nodes and edges. The core idea of the update is similar to feature propagation, but crucially, embeddings of nodes seen at training time can also be updated. Additionally, modifications to N2V are proposed, including modifications to the loss function, to make it more suitable for iN2V. On a node classification benchmark of 10 network datasets, iN2V shows overall improvement relative to N2V in an inductive evaluation.

**Claims And Evidence:**

I feel that the authors could make a more targeted claim than the broader claim made in the abstract and introduction that iN2V provides a lift relative to N2V in the inductive setting, as this seems unsurprising and is already achieved by the prior method of feature propagation, which can serve as a baseline. In particular, I think a claim of gain relative to feature propagation should be specified upfront in terms of: 1) homophilic vs heterophilic dataset; 2) use of MLP vs SAGE as the classifier; and 3) whether provided features are used (concatenated). The results seem to indicate a significant lift relative to N2V in most cases, and some smaller significant lift relative to feature propagation in some cases, but this should be specified upfront so it is easier to evaluate the claim.

**Essential References Not Discussed:**

The area of graph signal processing contains many works on embedding / feature propagation that are core to the paper's thrust but not discussed here. A prime example is "A Unifying Generative Model for Graph Learning Algorithms: Label Propagation, Graph Convolutions, and Combinations" (Jia and Benson 2021), which also discusses other works in this area. This Unifying Model paper overlaps significantly with the core of this work's proposed algorithm, though that work provides more theoretical justification for their model and algorithms. The primary difference is the focus on node classification here vs node regression in that work. In particular, this prior work also effectively allows for the embeddings of nodes seen with given embeddings to be updated at test time, so is is unclear that the core distinction of this reviewed work from feature propagation is indeed original.

**Experimental Designs Or Analyses:**

The experimental design seems valid overall, and different results are provided for the cases of using provided node features or not, or testing the N2V modifications on feature propagation to separate the lift of these modifications from the core iN2V idea. Enough details in Section 4 that the experiments seem reproducible.
As the authors mention, one weakness is that random splits are used to test the inductive setting, which does not provide a realistic evaluation. Perhaps temporal network datasets could be used instead.

**Methods And Evaluation Criteria:**

The benchmark datasets used are reasonable and span a good variety of both homophilic and heterophilic networks. The authors commendably also include results where their proposed method, like the original N2V, performs poorly. Ideally, confidence intervals would be included for Table 3, since these results represent the performance lift in the more common setting with nodes have feature vectors.

**Other Comments Or Suggestions:**

- The "RDF" acronym should be defined in the text.
- Line 231, missing space after "GraphSAGE"
- Line 190, probability $p$ becomes probability $r$ in line 192

**Other Strengths And Weaknesses:**

- The related work contains broad descriptions of prior methods on node embedding methods and GNNs. It could be more focused on prior work relevant to the paper, like work involving the inductive setting and embedding propagation. The inductive setting is also related to the temporal setting, which could also be discussed.

**Questions For Authors:**

1. Please address the concerns in "Claims And Evidence": under which of these cases specifically is a significant gain claimed relative to feature propagation.
2. Please also address the concern in "Essential References Not Discussed" regarding originality vs prior work in graph signal processing. Relatedly, is there any possible theoretical validation of the proposed method vs feature propagation?

**Relation To Broader Scientific Literature:**

The authors discuss how their method can be used to extend node embedding methods like N2V to the inductive setting, presenting it is an overall superior alternative to feature propagation. As discussed below, there is arguably significant overlap with some prior work in the area of graph signal processing.

**Theoretical Claims:**

There are no theoretical claims or results.

---

> ### Author Rebuttal · Authors · 2025-04-01
>
> Question: [...]  make a more targeted claim than the broader claim made in the abstract and introduction that iN2V provides a lift relative to N2V in the inductive setting,
> [...]
> In particular, a claim of gain relative to feature propagation should be specified upfront in terms of: 1) homophilic vs heterophilic dataset; 2) use of MLP vs SAGE as the classifier; and 3) whether provided features are used (concatenated).
> [...] this should be specified upfront so it is easier to evaluate the claim.
>
> Answer:
> Averaged over the other parameters (dataset, split, model, embedding/cat) iN2V outperforms FP by the following amounts:
> +1 point on homophilic vs +0.7 on heterophilic datasets
> +1.3 when using MLP vs +0.6 when using Sage
> +1.3 when using the trained embeddings only vs +0.6 when concatenating them with the graph features
>
> Another interesting comparison in the same direction is
> +1.2 when using few (10, 20%) training data vs +0.8 when using more (60, 80%)
>
> We will update the claims in the Abstract and give some of these details in the Introduction.
>
>
> Question: [...] confidence intervals would be included for Table 3, since these results represent the performance lift in the more common setting with nodes have feature vectors.
>
> Answer: Table 3 aggregates the performance over all datasets and splits to show a compact comparison between using only the trained embeddings and concatenating them with the existing features.
> Details on the concatenation results including standard deviation can be found in tables 9-12 in the appendix.
>
>
> Question:
> 1) As the authors mention, one weakness is that random splits are used to test the inductive setting, which does not provide a realistic evaluation.
> 2) Perhaps temporal network datasets could be used instead.
>
> Answer:
> 1) Unless the domain really requires it, it is important to use random splits as similarly performing methods can be arbitrarily reranked by choosing a fixed split, see Pitfalls of Graph Neural Network Evaluation https://arxiv.org/abs/1811.05868
> 2) We specifically focus on the inductive setting.
> When proposing a method for temporal datasets and claiming benefits there, one should do a thorough investigation there and deal with things like feature and class shifts, unknown classes appearing during testing that were not part of the train set, how to incorporate the data from the intermediate timesteps and catastrophic forgetting (e.g. if a class disappears for some timesteps).
> As we want to keep our work focused on the classical inductive case, we chose not to use temporal datasets and baselines.
>
> Question: A prime example is "A Unifying Generative Model for Graph Learning Algorithms: Label Propagation, Graph Convolutions, and Combinations" (Jia and Benson 2021), which also discusses other works in this area. This Unifying Model paper overlaps significantly with the core of this work's proposed algorithm, though that work provides more theoretical justification for their model and algorithms. [...]
>
> Answer: The Unifying Model paper takes an attributed (features+labels) graph and fits their MRF to it. The focus of the paper is to show that this is similar to label propagation when using only labels as attributes and similar to a linear GCN when using the features.
> This is a different setting than what we tackle which is to give embeddings to unseen test nodes in graphs with no features or labels.
> Their method could be used on top of N2V to treat the N2V embeddings as attributes of the training nodes. Though this combination would be a lot more complex than FP or our proposed iN2V.
>
> We add this paper to our related work and explain the difference in setting.
>
>
> The related work contains broad descriptions of prior methods on node embedding methods and GNNs. It could be more focused on prior work relevant to the paper, like work involving the inductive setting and embedding propagation. The inductive setting is also related to the temporal setting, which could also be discussed.
>
> Answer: We will adapt the related work to focus more on inductive methods, as reviewer VRfA also proposed some in the inductive knowledge graph direction. Additionally, we will add a short paragraph discussing the connection between the inductive and temporal settings.
>
>
> Question: Relatedly, is there any possible theoretical validation of the proposed method vs feature propagation?
>
> Answer: We currently do not have a theoretical comparison of FP vs iN2V, but in the motivating for our method we demonstrate examples where FP is unable to adapt to the changed graph structure when new edges appear at test time vs our iN2V which can adapt to this situation. As more changes to edge structure happen with fewer training data, the bigger gain of iN2V vs FP when using less training data (+1.2 for (10, 20%) vs +0.8 for (60, 80%)) also supports this example empirically.
>
> We incorporate the Other Comments Or Suggestions:
> RDF acronym definition; Line 231 missing space; Line 190/192 r/p

---

> > ### Comment · Reviewer_9oof · 2025-04-05
> >
> > Thank you for the response.
> > Regarding the point about lift, I think it would be beneficial to highlight the settings (dataset type, classifier, ratio) where there is consistent statistically significant lift across datasets, as opposed to an aggregated percentage lift, since the latter can overweight a few datasets.
> > More importantly, I think the paper would benefit from a revision incorporating the reviewers' suggestions about better positioning the idea here among other related work, so I will maintain the score.

---

> > > ### Author Response · Authors · 2025-04-08
> > >
> > > First, we have created the revision. But please note that providing an updated paper in the ICML procedure is not possible.
> > >
> > > Second, we will check the details of the significance tests, as it requires careful analysis in terms of chosing a Bonferroni correction and likely will require a more advanced post-hoc test to provide insights of significance.

---

### Official Review · Reviewer_VRfA · 2025-03-14

**Overall Recommendation:** 4

**Summary:**

This paper introduces an inductive approach to adapt shallow node embeddings for predicting unseen nodes that have at least one edge connected to the training graph. The proposed inductive node2vec method integrates post-hoc processing of node embeddings alongside corresponding adjustments to the training process to accommodate this post-hoc refinement. Extensive experiments are conducted on both homophilous and heterophilous graphs. Results demonstrate that the proposed method improves node classification performance by five percentage points.

**Claims And Evidence:**

1. Lack of comparison with other GNNs. The submission lacks some baseline comparisons with Graph Neural Networks (GNNs), which are well-known for their inductive reasoning capabilities, particularly since they can leverage node features. While these models may not necessarily outperform the proposed method, including them in the results table would provide valuable context and better ground the method's performance. I would strongly recommend incorporating additional GNN baselines such as GIN and GAT, rather than relying solely on GraphSAGE.

2. The submission overlooks key foundational works in the literature. Notably, there have been prior efforts to inductivize node embeddings. For instance, [RefactorGNNs](https://arxiv.org/abs/2207.09980) (NeurIPS 2022) specifically focus on making shallow knowledge graph embeddings inductive by establishing connections between node embeddings and message-passing frameworks. Drawing connections between the proposed method and RefactorGNNs could offer useful insights and strengthen the theoretical grounding of the work.

**Essential References Not Discussed:**

The task of making transductive models inductive has been explored in the literature. On one dimension, the authors discuss the limitations of GNNs for inductive reasoning due to the lack of node features, which I appreciate. On the other dimension, there is deeper connection between transductive embeddings and inductive embeddings [1], which can benefit from more discussion in the revision.

For related works on knowledge graphs, Sec 2.1 touches some of them. Other popular methods like [2,3,4] are also prominent in knowledge graph embeddings and should be referenced in Section 2.1.

[1] [RefactorGNNs](https://arxiv.org/abs/2207.09980) (NeurIPS 2022)

[2] [Canonical Tensor Decomposition for Knowledge Base Completion](https://arxiv.org/pdf/1806.07297.pdf) ICML 2018

[3] [Convolutional 2D Knowledge Graph Embeddings](https://arxiv.org/abs/1707.01476) AAAI 2018

[4] [Relation Prediction as an Auxiliary Training Objective for Improving Multi-Relational Graph Representations](https://openreview.net/forum?id=Qa3uS3H7-Le) (AKBC 2021)

**Experimental Designs Or Analyses:**

Yes, I reviewed the datasets used, the dataset splitting strategy for train/validation/test sets, and the hyperparameter selection process. The experimental design and analysis appear sound and appropriate.

**Methods And Evaluation Criteria:**

Yes, the proposed method is simple and intuitive for inductivizing node embeddings. One concern is the evaluation task only consists of node classification while it might be different for inductivising node embeddings for tasks like link prediction.

**Other Comments Or Suggestions:**

- These two sentences convey similar meanings and could be merged for conciseness:
"We expand on this idea and propose a general, simple post-hoc approach to turn transductive embeddings into effective inductive embeddings. We introduce iN2V, an approach to using trained embeddings to induce embeddings for nodes appearing in the inductive test set."

- Line 246: This sentence is dense with information and may benefit from being split into two for improved clarity:
"Two baselines that are not directly comparable as they use more information but are nevertheless useful for perspective are using the original graph features and training N2V embeddings in a transductive setup."

**Other Strengths And Weaknesses:**

Strengths:

- The proposed method is simple yet effective in making node2vec inductive.

Weaknesses:

- see above in Claims and Evidence box.
- The absence of key baselines, such as Graph Neural Networks (GNNs), limits the completeness of the evaluation. Including models like GIN and GAT would offer better insights into the method's relative performance.
- There is little discussion on potential connections between the proposed method and established message-passing frameworks in GNNs.  For example, can the proposed method rewrite into some form of message-passing?

**Questions For Authors:**

Q1: Can you explain line 134: v17 = v7/4^4? How is this derived?

Q2: How could your method be extended to knowledge graphs?

**Relation To Broader Scientific Literature:**

The paper presents a simple yet intuitive approach for predicting unseen nodes at test time. The proposed method combines post-hoc processing with modifications to the standard training procedure, offering a meaningful contribution to the field of inductivizing shallow embeddings.

However, the paper overlooks some key related works in this area. For instance, periodically resetting a subset of node embeddings could serve as a useful baseline [1,2]. Additionally, exploring extensions to knowledge graphs could provide valuable insights, as efforts to inductivize transductive models have been investigated in the context of knowledge graph embeddings [1,3].

**References:**

[1] [RefactorGNNs](https://arxiv.org/abs/2207.09980) (NeurIPS 2022)

[2] [Resetting Embedding Layer](https://arxiv.org/abs/2307.01163) (NeurIPS 2023)

[3] [NBFNet](https://arxiv.org/abs/2106.06935) (NeurIPS 2021)

**Theoretical Claims:**

NA

---

> ### Author Rebuttal · Authors · 2025-03-31
>
> Question:
> I would strongly recommend incorporating additional GNN baselines such as GIN and GAT, rather than relying solely on GraphSAGE.
>
> Answer:
> The focus of the experiments is on evaluating the trained embeddings. For this, we chose MLP and GraphSage as representative models. Other models might reach higher absolute performance numbers, but the focus was always on evaluating the trained embeddings. Nevertheless, we agree that using only one MP-GNN is limited and are currently training GIN and GAT, as suggested.
>
> First results suggest that for already well-performing settings (homophilic FP/iN2V), there is a 0-2% gain compared to GraphSage.
> For very challenging settings (e.g. Actor dataset) there is no change to a small drop in performance. The biggest impact we have seen so far is that the N2V (inductive) baseline in the embedding only setting improves from 42(GraphSAGE) to 68 (GIN) / 69 (GAT) on the Cora  10% split.
>
> Full results will be added to the Appendix in the style of tables 5-12.
>
>
> Question:
> Relation to RefactorGNNs (NeurIPS 2022)
>
> Answer:
> RefactorGNN tries to combine the benefits of FMs and GNNs (applicable to inductive tasks, easy feature integration) into a single model that is used for KGC.
> Our goal differs as we modify N2V to generate embeddings in the inductive case that can be used with any other GNN afterwards instead of building a singular model.
> Also, we do not focus on KGC.
> Nonetheless, this is a methodologically interesting paper that is somewhat related as it also tackles an inductive problem, and we add it to our related work.
>
>
> Question:
> [...] key related works in this area. For instance, periodically resetting a subset of node embeddings could serve as a useful baseline [1,2]. Additionally, exploring extensions to knowledge graphs could provide valuable insights, as efforts to inductivize transductive models have been investigated in the context of knowledge graph embeddings [1,3].
> References:
> [1] RefactorGNNs (NeurIPS 2022)
> [2] Resetting Embedding Layer (NeurIPS 2023)
> [3] NBFNet (NeurIPS 2021)
>
> Answer:
> [2] occasionally resets the embedding layer of a PLM during training to improve adaptation to new languages. N2V trains an embedding for each node (in the training set) but does not have any layers after the embedding like PLMs that could be coerced to prioritize adaptability to changing embeddings. The closest connection we see to our work is to the sampling-based modification of N2V training, which during training replaces some of the trained node embeddings with the mean neighbour embedding for a single iteration.
>
> Regarding work that extends KG methods to the inductive case. We will add [1] to our RW and discuss it there, together with the suggestions from your next point.
>
>
> Question:
> [...] Other popular methods like [2,3,4] are also prominent in knowledge graph embeddings and should be referenced in Section 2.1.
> [1] RefactorGNNs (NeurIPS 2022)
> [2] Canonical Tensor Decomposition … ICML 2018
> [3] Convolutional 2D Knowledge Graph Embeddings AAAI 2018
> [4] Relation Prediction as an Auxiliary … (AKBC 2021)
>
> Answer: We had to cut parts of the KG-embedding related work for the given page limit. We will add the proposed papers to the RW and slightly compact the related work.
>
>
> Question:
> Can the proposed method rewrite into some form of message-passing?
>
> Answer:
> Depends on how strict one sees message-passing. The Aggregate function is a simple convolution modified to ignore neighbors without embeddings. Equation 1 defines the Update function, which takes an entity's own embedding and the aggregated neighbor embeddings to compute its updated embedding.
>
>
> Questions For Authors:
>
> Q1: Can you explain line 134: v17 = v7/4^4? How is this derived?
> Q2: How could your method be extended to knowledge graphs?
>
> Answer:
> Q1: This is a typo, and it should be v7/4^3. Does this answer the question, or should we explain the example in more detail?
> Q2: First, one needs to think about what exactly the task and setting are. Our method focuses on the inductive and feature-free setup. For the inductive extension, we rely on having the edges of the test set. A common task for KGs is Link Prediction, which predicts information that our method uses to generate the embeddings. Therefore, it would be a challenge to apply our method here.
> For other tasks where the test edges/relations are available, it could be used as-is or by replacing N2V with a more KG-focused embedding method (like TransE) that embeds relations.
> When doing so, one would also need to adapt equations 1b and 1c not to use the average neighbor embedding, but to reflect how the relation embeddings are trained, e.g. the average over r_vw+w (+ from TransE, depending on the chosen KG embedding).

---

> > ### Comment · Reviewer_VRfA · 2025-04-04
> >
> > Thanks for addressing my concerns and questions. The additional results look good to me. The promised expansion on the related work also makes sense.
> >
> > Q1: yes, the typo explains my confusion.
> >
> > Q2: It would be great if you could add this discussion about how the method can or can not be used in inductivising  in multi-relational graphs (e.g. knowledge graph). For example is the method limited to simple graphs?

---

> > > ### Author Response · Authors · 2025-04-07
> > >
> > > Yes, we will add this point about multi-relational graphs to the future work section.
> > >
> > >
> > > iN2V is not limited to simple graphs.
> > > Self loops are no problem, it just means that v \in N(v) and that h_v has a slightly higher (depending on how many other neighbors of v have an embedding) weight in equation 1b than just \lambda as it also appears in the mean neighbor embedding m_{N_s(v)}.
> > > iN2V also works for graphs with multi-edges, in the equations/notation the set of edges and set of neighbors would just change to multi-sets.
> > > Both of these things should already be possible with our implementation, though we did not explicitly test it.
> > >
> > > Edge weights could be incorporated into the mean neighbor embedding m_{N_s(v)} by calculating a weighted sum using normalized edge weights instead of the simple mean.

---

### Decision · Program_Chairs · 2025-05-01

**Decision:**

Accept (poster)

**Comment:**

The paper proposes iN2V, an inductive version of node2vec for computing node embeddings. The paper has merits that the reviewers have highlighted. These include the simplicity of the approach and its effectiveness on downstream tasks. Nevertheless, the reviewers have also indicated limitations mainly related to the following points: (i) A comparison to GNN models (apart from GraphSAGE) is missing; (ii) The discussion of the related work seems insufficient. Considering the authors' responses during the rebuttal period, I'm in favor of accepting the paper. I encourage the authors to take these comments into consideration (especially the comparison to other GNNs) and revise the manuscript accordingly.